# Provably Efficient Algorithm for Nonstationary Low-Rank MDPs

**Yuan Cheng**
National University of Singapore
`yuan.cheng@u.nus.edu`

**Jing Yang**
The Pennsylvania State University
`yangjing@psu.edu`

**Yingbin Liang**
The Ohio State University
`liang.889@osu.edu`

## Abstract

Reinforcement learning (RL) under changing environment models many real-world applications via nonstationary Markov Decision Processes (MDPs), and hence gains considerable interest. However, theoretical studies on nonstationary MDPs in the literature have mainly focused on tabular and linear (mixture) MDPs, which do not capture the nature of unknown representation in deep RL. In this paper, we make the first effort to investigate nonstationary RL under episodic low-rank MDPs, where both transition kernels and rewards may vary over time, and the low-rank model contains unknown representation in addition to the linear state embedding function. We first propose a parameter-dependent policy optimization algorithm called PORTAL, and further improve PORTAL to its parameter-free version of Ada-PORTAL, which is able to tune its hyper-parameters adaptively without any prior knowledge of nonstationarity. For both algorithms, we provide upper bounds on the average dynamic suboptimality gap, which show that as long as the nonstationarity is not significantly large, PORTAL and Ada-PORTAL are sample-efficient and can achieve arbitrarily small average dynamic suboptimality gap with polynomial sample complexity.

## 1 Introduction

Reinforcement learning (RL) has gained significant success in real-world applications such as board games of Go and chess (Silver et al., 2016, 2017, 2018), robotics (Levine et al., 2016; Gu et al., 2017), recommendation systems (Zhao et al., 2021) and autonomous driving (Bojarski et al., 2016; Ma et al., 2021). Most theoretical studies on RL have been focused on a stationary environment and evaluated the performance of an algorithm by comparing against only one best fixed policy (i.e., *static regret*). However, in practice, the environment is typically time-varying and *nonstationary*. As a result, the transition dynamics, rewards and consequently the optimal policy change over time.

There has been a line of research studies that investigated *nonstationary* RL. Specifically, Gajane et al. (2018); Cheung et al. (2020); Mao et al. (2021) studied nonstationary tabular MDPs. To further overcome the curse of dimensionality, Fei et al. (2020); Zhou et al. (2020) proposed algorithms for nonstationary linear (mixture) MDPs and established upper bounds on the dynamic regret.

In this paper, we significantly advance this line of research by investigating nonstationary RL under *low-rank* MDPs (Agarwal et al., 2020b), where the transition kernel of each MDP admits a decomposition into a representation function and a state-embedding function that map to low dimensional spaces. Compared with linear MDPs where the representation is known, the low-rank MDP model contains unknown representation, and is hence much more powerful to capture

37th Conference on Neural Information Processing Systems (NeurIPS 2023).

representation learning that occurs often in deep RL. Although there have been several recent studies on static low-rank MDPs (Agarwal et al., 2020b; Uehara et al., 2022; Modi et al., 2021), nonstationary low-rank MDPs remain unexlored, and are the focus of this paper.

To investigate nonstationary low-rank MDPs, several challenges arise. (a) All previous studies of nonstationary MDPs took on-policy exploration, such a strategy will have difficulty in providing sufficiently accurate model (as well as representation) learning for nonstationary low-rank MDPs. (b) Under low-rank MDPs, since both representation and state-embedding function change over time, it is more challenging to use history data collected under previous transition kernels for current use.

The **main contribution** of this paper lies in addressing above challenges and designing a provably efficient algorithm for nonstationary low-rank MDPs. We summarize our contributions as follows.

- We propose a novel policy optimization algorithm with representation learning called PORTAL for nonstationary low-rank MDPs. PORTAL features new components, including off-policy exploration, data-transfer model learning, and target policy update with periodic restart.

- We theoretically characterize the average dynamic suboptimality gap ($\mathrm{Gap_{Ave}}$) of PORTAL, where $\mathrm{Gap_{Ave}}$ serves as a new metric that captures the performance of target policies with respect to the best policies at each instance in the nonstationary MDPs under off-policy exploration. We further show that with prior knowledge on the degree of nonstationarity, PORTAL can select hyper-parameters that minimize $\mathrm{Gap_{Ave}}$. If the nonstationarity is not significantly large, PORTAL enjoys a diminishing $\mathrm{Gap_{Ave}}$ with respect to the number of iterations $K$, indicating that PORTAL can achieve arbitrarily small $\mathrm{Gap_{Ave}}$ with polynomial sample complexity.

  Our analysis features a few new developments. (a) We provide a new MLE guarantee under nonstationary transition kernels that captures errors of using history data collected under different transition kernels for benefiting current model estimation. (b) We establish trajectory-wise uncertainty bound for estimation errors via a square-root $\ell_\infty$-norm of variation budgets. (c) We develop an error tracking technique via auxiliary anchor representation for convergence analysis.

- Finally, we improve PORTAL to a *parameter-free* algorithm called Ada-PORTAL, which does not require prior knowledge on nonstationarity and is able to tune the hyper-parameters adaptively. We further characterize $\mathrm{Gap_{Ave}}$ of Ada-PORTAL as $\tilde{O}(K^{-\frac{1}{6}}(\Delta+1)^{\frac{1}{6}})$, where $\Delta$ captures the variation of the environment. Notably, based on PORTAL, we can also use the black-box method called MASTER in Wei & Luo (2021) to turn PORTAL into a parameter-free algorithm (called MASTER+PORTAL) with $\mathrm{Gap_{Ave}}$ of $\tilde{O}(K^{-\frac{1}{6}}\Delta^{\frac{1}{3}})$. Clearly, Ada-PORTAL performs better than MASTER+PORTAL when nonstationarity is not significantly small, i.e. $\Delta \geq \tilde{O}(1)$.

To our best knowledge, this is the first study of nonstationary RL under low-rank MDPs.

## 2   Related Works

Various works have studied nonstationary RL under tabular and linear MDPs, most of which can be divided into two lines: policy optimization methods and value-based methods.

**Nonstationary RL: Policy Optimization Methods.** As a vast body of existing literature (Cai et al., 2020; Shani et al., 2020; Agarwal et al., 2020a; Xu et al., 2021) has proposed policy optimization methods attaining computational efficiency and sample efficiency simultaneously in *stationary* RL under various scenarios, only several papers investigated policy optimization algorithm in *nonstationary* environment. Assuming time-varying rewards and time-invariant transition kernels, Fei et al. (2020) studied nonstationary RL under tabular MDPs. Zhong et al. (2021) assumed both transition kernels and rewards change over episodes and studied nonstationary linear mixture MDPs. These policy optimization methods all assumed prior knowledge on nonstationarity.

**Nonstationary RL: Value-based Methods.** Assuming that both transition kernels and rewards are time-varying, several works have studied nonstationary RL under tabular and linear MDPs, most of which adopted Upper Confidence Bound (UCB) based algrithms. Cheung et al. (2020) investigated tabular MDPs with infinite-horizon and proposed algorithms with both known variation budgets and unknown variation budgets. In addition, this work also proposed a Bandit-over-Reinforcement Learning (BORL) technique to deal with unknown variation budgets. Mao et al. (2021) proposed a model-free algorithm with sublinear dynamic regret bound. They then proved a lower bound for nonstationary tabular MDPs and showed that their regret bound is near min-max optimal. Touati

& Vincent (2020); Zhou et al. (2020) considered nonstationary RL in linear MDPs and proposed algorithms achieving sublinear regret bounds with unknown variation budgets.

Besides these two lines of researches, Wei & Luo (2021) proposed a black-box method that turns a RL algorithm with optimal regret in a (near-)stationary environment into another algorithm that can work in a nonstationary environment with sublinear dynamic regret without prior knowledge on nonstationarity. In this paper, we show that our algorithm Ada-PORTAL outperforms such a type of black-box method (taking PORTAL as subroutine) if nonstationarity is not significantly small.

**Stationary RL under Low-rank MDPs.** Low-rank MDPs were first studied by Agarwal et al. (2020b) in a reward-free regime, and then Uehara et al. (2022) studied low-rank MDPs for both online and offline RL with known rewards. Cheng et al. (2023) studied reward-free RL under low-rank MDPs and improved the sample complexity of previous works. Modi et al. (2021) proposed a model-free algorithm MOFFLE under low-nonnegative-rank MDPs. Cheng et al. (2022); Agarwal et al. (2022) studied multitask representation learning under low-rank MDPs, and further showed the benefit of representation learning to downstream RL tasks.

## 3 Formulation

**Notations:** We use $[K]$ to denote set $\{1, \ldots, K\}$ for any $K \in \mathbb{N}$, use $\|x\|_2$ to denote the $\ell_2$ norm of vector $x$, use $\triangle(\mathcal{A})$ to denote the probability simplex over set $\mathcal{A}$, use $\mathcal{U}(\mathcal{A})$ to denote uniform sampling over $\mathcal{A}$, given $|\mathcal{A}| < \infty$, and use $\triangle(\mathcal{S})$ to denote the set of all possible density distributions over set $\mathcal{S}$. Furthermore, for any symmetric positive definite matrix $\Sigma$, we let $\|x\|_\Sigma := \sqrt{x^\top \Sigma x}$. For distributions $p_1$ and $p_2$, we use $D_{KL}(p_1(\cdot)\|p_2(\cdot))$ to denote the KL divergence between $p_1$ and $p_2$.

### 3.1 Episodic MDPs and Low-rank Approximation

An episodic MDP is denoted by a tuple $\mathcal{M} := \left(\mathcal{S}, \mathcal{A}, H, P := \{P_h\}_{h=1}^H, r := \{r_h\}_{h=1}^H\right)$, where $\mathcal{S}$ is a possibly infinite state space, $\mathcal{A}$ is a finite action space with cardinality $A$, $H$ is the time horizon of each episode, $P_h(\cdot|\cdot,\cdot) : \mathcal{S} \times \mathcal{A} \to \triangle(\mathcal{S})$ denotes the transition kernel at each step $h$, and $r_h(\cdot,\cdot) : \mathcal{S} \times \mathcal{A} \to [0,1]$ denotes the deterministic reward function at each step $h$. We further normalize the reward as $\sum_{h=1}^H r_h \leq 1$. A policy $\pi = \{\pi_h\}_{h\in[H]}$ is a set of mappings where $\pi_h : \mathcal{S} \to \triangle(\mathcal{A})$. For any $(s,a) \in \mathcal{S} \times \mathcal{A}$, $\pi_h(a|s)$ denotes the probability of selecting action $a$ at state $s$ at step $h$. For any $(s,a) \in \mathcal{S} \times \mathcal{A}$, let $(s_h, a_h) \sim (P, \pi)$ denote that the state $s_h$ is sampled by executing policy $\pi$ to step $h$ under transition kernel $P$ and then action $a_h$ is sampled by $\pi_h(\cdot|s_h)$.

Given any state $s \in \mathcal{S}$, the value function for a policy $\pi$ at step $h$ under an MDP $\mathcal{M}$ is defined as the expected value of the accumulative rewards as: $V_{h,P,r}^\pi(s) = \sum_{h'=h}^H \mathbb{E}_{(s_{h'},a_{h'})\sim(P,\pi)}[r_{h'}(s_{h'}, a_{h'})|s_h = s]$. Similarly, given any state-action pair $(s,a) \in \mathcal{S} \times \mathcal{A}$, the action-value function (Q-function) for a policy $\pi$ at step $h$ under an MDP $\mathcal{M}$ is defined as $Q_{h,P,r}^\pi(s,a) = r_h(s,a) + \sum_{h'=h+1}^H \mathbb{E}_{(s_{h'},a_{h'})\sim(P,\pi)}[r_{h'}(s_{h'}, a_{h'})|s_h = s, a_h = a]$. Denote $(P_h f)(s,a) := \mathbb{E}_{s'\sim P_h(\cdot|s,a)}[f(s')]$ for any function $f : \mathcal{S} \to \mathbb{R}$. Then we can write the action-value function as $Q_{h,P,r}^\pi(s,a) = r_h(s,a) + (P_h V_{h+1,P,r}^\pi)(s,a)$. For any $k \in [K]$, without loss of generality, we assume the initial state $s_1$ to be fixed and identical, and we use $V_{P,r}^\pi$ to denote $V_{1,P,r}^\pi(s_1)$ for simplicity.

This paper focuses on low-rank MDPs (Jiang et al., 2017; Agarwal et al., 2020b) defined as follows.

**Definition 3.1** (Low-rank MDPs). A transition kernel $P_h^* : \mathcal{S} \times \mathcal{A} \to \triangle(\mathcal{S})$ admits a low-rank decomposition with dimension $d \in \mathbb{N}$ if there exist a representation function $\phi_h^\star : \mathcal{S} \times \mathcal{A} \to \mathbb{R}^d$ and a state-embedding function $\mu_h^\star : \mathcal{S} \to \mathbb{R}^d$ such that

$$P_h^\star(s'|s,a) = \langle \phi_h^\star(s,a), \mu_h^\star(s') \rangle, \quad \forall s, s' \in \mathcal{S}, a \in \mathcal{A}.$$

Without loss of generality, we assume $\|\phi_h^*(s,a)\|_2 \leq 1$ for all $(s,a) \in \mathcal{S} \times \mathcal{A}$ and for any function $g : \mathcal{S} \mapsto [0,1]$, $\left\|\int \mu_h^\star(s)g(s)ds\right\|_2 \leq \sqrt{d}$. An MDP is a low-rank MDP with dimension $d$ if for any $h \in [H]$, its transition kernel $P_h^*$ admits a low-rank decomposition with dimension $d$. Let $\phi^\star = \{\phi_h^\star\}_{h\in[H]}$ and $\mu^\star = \{\mu_h^\star\}_{h\in[H]}$ be the true representation and state-embedding functions.

## 3.2 Nonstationary Transition Kernels with Adversarial Rewards

In this paper, we consider an episodic RL setting under changing environment, where both transition kernels and rewards vary over time and possibly in an adversarial fashion.

Specifically, suppose the RL system goes by *rounds*, where each round have a fixed number of episodes, and the transition kernel and the reward remain the same in each round, and can change adversarially across rounds. For each round, say round $k$, we denote the MDP as $\mathcal{M}^k = (\mathcal{S}, \mathcal{A}, H, P^k := \{P_h^{\star,k}\}_{h=1}^H, r^k := \{r_h^k\}_{h=1}^H)$, where $P^{\star,k}$ and $r^k$ denote the true transition kernel and the reward of round $k$. Further, $P^{\star,k}$ takes the low-rank decomposition as $P^{\star,k} = \langle \phi^{\star,k}, \mu^{\star,k} \rangle$. Both the representation function $\phi^{\star,k}$ and the state embedding function $\mu^{\star,k}$ can change across rounds. Given the reward function $r^k$, there always exists an optimal policy $\pi^{\star,k}$ that yields the optimal value function $V_{P^{\star,k},r^k}^{\pi^{\star,k}} = \sup_\pi V_{P^{\star,k},r^k}^\pi$, abbreviated as $V_{P^{\star,k},r^k}^\star$. Clearly, the optimal policy also changes across rounds.

We assume the agent interacts with the nonstationary environment (i.e., the time-varying MDPs) over $K$ rounds in total without the knowledge of transition kernels $\{P^{k,\star}\}_{k=1}^K$. At the beginning of each round $k$, the environment changes to a possibly adversarial transition kernel unknown to the agent, picks a reward function $r^k$, which is revealed to the agent only at the end of round $k$, and outputs a fixed initial state $s_1$ for the agent to start the exploration of the environment for each episode. The agent is allowed to interact with MDPs via a few episodes with one or multiple *exploration* policies at her choice to take samples from the environment and then should output an target policy to be executed during the next round. Note that in our setting, the agent needs to decide exploration and target policies only based on the information in previous rounds, and hence exploration samples and the reward information of the current round help only towards future rounds.

## 3.3 Learning Goal and Evaluation Metric

In our setting, the agent seeks to find the optimal policy at each round $k$ (with only the information of previous rounds), where both transition kernels and rewards can change over rounds. Hence we define the following notion of *average dynamic suboptimality gap* to measure the convergence of the target policy series to the optimal policy series.

**Definition 3.2** (Average Dynamic Suboptimality Gap). For $K$ rounds, and any policy set $\{\pi^k\}_{k \in [K]}$, the average dynamic suboptimality gap ($\mathrm{Gap}_{\mathrm{Ave}}$) of the value functions over $K$ rounds is given as $\mathrm{Gap}_{\mathrm{Ave}}(K) = \frac{1}{K}\sum_{k=1}^K [V_{P^{\star,k},r^k}^\star - V_{P^{\star,k},r^k}^{\pi^k}]$. For any $\epsilon$, we say an algorithm is $\epsilon$-average suboptimal, if it outputs a policy set $\{\pi^k\}_{k \in [K]}$ satisfying $\mathrm{Gap}_{\mathrm{Ave}}(K) \leq \epsilon$.

$\mathrm{Gap}_{\mathrm{Ave}}$ compares the agent's target policy to the optimal policy of each individual round in hindsight, which captures the dynamic nature of the environment. This is in stark contrast to the stationary setting where the comparison policy is a single fixed best policy over all rounds. This notion is similar to *dynamic regret* used for nonstationary RL (Fei et al., 2020; Gajane et al., 2018), where the only difference is that $\mathrm{Gap}_{\mathrm{Ave}}$ evaluates the performance of target policies rather than the exploration policies. Hence, given any target accuracy $\epsilon \geq 0$, the agent is further interested in the statistical efficiency of the algorithm, i.e., using as few trajectories as possible to achieve $\epsilon$-average suboptimal.

## 4 Policy Optimization Algorithm and Theoretical Guarantee

### 4.1 Base Algorithm: PORTAL

We propose a novel algorithm called PORTAL (Algorithm 1), which features three main steps. Below we first summarize our main design ideas and then explain reasons behind these ideas as we further elaborate main steps of PORTAL.

**Summary of New Design Ideas:** PORTAL features the following main design ideas beyond previous studies on nonstationary RL under tabular and linear MDPs. (a) PORTAL features a specially designed *off-policy* exploration which turns out to be beneficial for nonstationary low-rank /MDP models rather than the typical *on-policy* exploration taken by previous studies of nonstationary tabular and linear MDP models. (b) PORTAL transfers history data collected under various different transition kernels for benefiting the estimation of the current model. (c) PORTAL updates target

policies with periodic restart. As a comparison, previous work using periodic restart (Fei et al., 2020) chooses the restart period $\tau$ based on a certain smooth visitation assumption. Here, we remove such an assumption and hence our choice of $\tau$ is applicable to more general model classes.

**Step 1. Off-Policy Exploration for Data Collection:** We take *off-policy* exploration, which is beneficial for nonstationary low-rank MDPs than simply using the target policy for *on-policy* exploration taken by the previous studies on nonstationary tabular or linear (mixture) MDPs (Zhong et al., 2021; Fei et al., 2020; Zhou et al., 2020). To further explain, we first note that under tabular or linear (mixture) MDPs studied in the previous work, a bonus term is introduced to the actual reward to serve as a *point-wise* uncertainty level of the estimation error for each state-action pair at any step $h$, so that for any step $h$, $\hat{Q}_h^k$ is a good optimistic estimation for $Q_{h,P^{\star,k},r^k}^\pi$. Hence it suffices to collect samples using the target policy. However, in low-rank MDPs, the bonus term $\hat{b}_h^k$ cannot serve as a point-wise uncertainty measure. For step $h \geq 2$, $\hat{Q}_h^k$ is not a good optimistic estimation for the true value function if the agent only uses target policy to collect data (i.e., for on-policy exploration). Hence, more samples and a novel off-policy exploration are required for a good estimation under low-rank MDPs. Specifically, as line 5 in Algorithm 1, at the beginning of each round $k$, for each step $h \in [H]$, the agent explores the environment by executing the exploration policy $\tilde{\pi}^{k-1}$ to state $\tilde{s}_{h-1}^{k,h}$ and then taking two uniformly chosen actions[1], where $\tilde{\pi}^{k-1}$ is determined in Step 2 of the previous round.

---

**Algorithm 1 PORTAL** (**P**olicy **O**ptimization with **R**epresen**TA**tion **L**earning under nonstationary MDPs)

---

1: **Input:** Rounds $K$, hyper-parameters $\tau, W$, regularizer $\lambda_{k,W}$, coefficient $\tilde{\alpha}_{k,W}$, stepsize $\eta$ and models $\{\Psi, \Phi\}$.

2: **Initialization:** $\pi_0(\cdot|s)$ to be uniform; $\tilde{\mathcal{D}}_h^{(0,0)} = \emptyset$.

3: **for** episode $k = 1, \ldots, K$ **do**

4:   **for** step $h = 1, \ldots, H$ **do**

5:     Roll into $\tilde{s}_{h-1}^{(k,h)}$ using $\tilde{\pi}^{k-1}$, uniformly choose $\tilde{a}_{h-1}^{(k,h)}, \tilde{a}_h^{(k,h)}$, and enter into $\tilde{s}_h^{(k,h)}, \tilde{s}_{h+1}^{(k,h)}$.

6:     Update datasets
$$\tilde{\mathcal{D}}_{h-1}^{(k,h,W)} = \left\{ \tilde{s}_{h-1}^{(i,h)}, \tilde{a}_{h-1}^{(i,h)}, \tilde{s}_h^{(i,h)} \right\}_{i=1 \vee k-W+1}^k,$$
$$\tilde{\mathcal{D}}_h^{(k,h,W)} = \left\{ \tilde{s}_h^{(i,h)}, \tilde{a}_h^{(i,h)}, \tilde{s}_{h+1}^{(i,h)} \right\}_{i=1 \vee k-W+1}^k.$$

7:   **end for**

8:   Receive full information rewards $r^k = \{r_h^k\}_{h \in [H]}$.

9:   Estimate transition kernel and update the exploration policy $\tilde{\pi}^k$ for the next round via:
$$\mathrm{E}^2\mathrm{U}\left(k, \{\tilde{\mathcal{D}}_{h-1}^{(k,h,W)}\}, \{\tilde{\mathcal{D}}_h^{(k,h,W)}\}\right).$$

10:   **for** step $h = 1, \ldots, H$ **do**

11:     Update $\hat{Q}_h^k = Q_{h,\hat{P}^k,r^k}^{\pi^k}$.

12:   **end for**

13:   **if** $k \mod \tau = 1$ **then**

14:     Set $\{\hat{Q}_h^k\}_{h \in [H]}$ as zero functions and $\{\pi_h^k\}_{h \in [H]}$ as uniform distributions on $\mathcal{A}$.

15:   **end if**

16:   **for** step $h = 1, \ldots, H$ **do**

17:     Update the target policy as in Equation (2).

18:   **end for**

19: **end for**

20: **Output:** $\{\pi^k\}_{k=1}^K$.

---

**Step 2. Data-Transfer Model Learning and $\mathrm{E}^2\mathrm{U}$:** In this step, we transfer history data collected under previous different transition kernels for benefiting the estimation of the current model. This is theoretically grounded by our result that the model estimation error can be decomposed into variation

---

[1]The subscript $h-1$ in $\tilde{s}_{h-1}^{k,h}$ indicates that the data is collected at time step $h$ of each trajectory, and the superscript $(k,h)$ indicates in which loop the data is collected (as in line 3 and 4 of Algorithm 1)

budgets plus a diminishing term as the estimation sample size increases, which justifies that the usage of data generated by mismatched distributions within a certain window is beneficial for model learning as long as variation budgets is mild. Then, the estimated model will further facilitate the selection of future exploration policies accurately.

Specifically, the agent selects desirable samples only from the latest $W$ rounds following a *forgetting rule* (Garivier & Moulines, 2011). Since nonstationary low-rank MDPs (compared to tabular and linear MDPs) also have additional variations on representations over time, the choice of $W$ needs to incorporate such additional information. Then the agent passes these selected samples to a subroutine $E^2U$ (see Algorithm 2), in which the agent estimates the transition kernels via the maximum likelihood estimation (MLE). Next, the agent updates the empirical covariance matrix $\hat{U}^{k,W}$ and exploration-driven bonus $\hat{b}^k$ as in lines 4 and 5 in Algorithm 2. We then define a *truncated value function* iteratively using the estimated transition kernel and the exploration-driven reward as follows:

$$
\hat{Q}^\pi_{h,\hat{P}^k,\hat{b}^k}(s_h, a_h) = \min\left\{1, \hat{b}^k_h(s_h, a_h) + \hat{P}^k_h \hat{V}^\pi_{h+1,\hat{P}^k,\hat{b}^k}(s_h, a_h)\right\},
$$
$$
\hat{V}^\pi_{h,\hat{P}^k,\hat{b}^k}(s_h) = \mathbb{E}_\pi\left[\hat{Q}^\pi_{h,\hat{P}^k,\hat{b}^k}(s_h, a_h)\right]. \tag{1}
$$

Although the bonus term $\hat{b}^k_h$ cannot serve as a *point-wise* uncertainty measure, the truncated value function $\hat{V}^\pi_{\hat{P}^k,\hat{b}^k}$ as the cumulative version of $\hat{b}^k_h$ can serve as a *trajectory-wise* uncertainty measure, which can be used to determine future exploration policies. Intuitively, for any policy $\pi$, the model estimation error satisfies $\mathbb{E}_{(s,a)\sim(P^{\star,k},\pi)}[\|\hat{P}^k(\cdot|s,a) - P^{\star,k}(\cdot|s,a)\|_{TV}] \le \hat{V}^\pi_{\hat{P}^k,\hat{b}^k} + \Delta$, where the error term $\Delta$ captures the variation of both transition kernels and representations over time. As a result, by selecting the policy that maximizes $\hat{V}^\pi_{\hat{P}^k,\hat{b}^k}$ as the exploration policy as in line 7, the agent will explore the trajectories whose states and actions have not been estimated sufficiently well so far.

---

**Algorithm 2 $E^2U$** (Model **E**stimation and **E**xploration Policy **U**pdate)

---

1: **Input:** round index $k$, regularizer $\lambda_{k,W}$ and coefficient $\tilde{\alpha}_{k,W}$, datasets $\{\tilde{\mathcal{D}}^{(k,h)}_{h-1}\}, \{\tilde{\mathcal{D}}^{(k,h)}_h\}$ and models $\{\Psi, \Phi\}$.
2: **for** step $h = 1, \dots, H$ **do**
3:   Learn the representation via MLE for step $h$:

$$
\hat{P}^k_h = (\hat{\phi}^k_h, \hat{\mu}^k_h) = \arg\max_{\phi\in\Phi,\mu\in\Psi} \mathbb{E}_{\tilde{\mathcal{D}}^{(k,h)}_h}\left[\log\langle\phi(s_h, a_h), \mu(s_{h+1})\rangle\right].
$$

4:   Compute the empirical covariance matrix as

$$
\hat{U}^{k,W}_h = \sum_{\tilde{\mathcal{D}}^{(k,h+1)}_h} \hat{\phi}^k_h(s_h, a_h)\hat{\phi}^k_h(s_h, a_h)^\top + \lambda_{k,W} I
$$

5:   Define exploration-driven bonus $\hat{b}^k_h(\cdot, \cdot) = \min\{\alpha_{k,W}\|\hat{\phi}^k_h(\cdot, \cdot)\|_{(\hat{U}^{k,W}_h)^{-1}}, 1\}$.
6: **end for**
7: Find exploration policy $\tilde{\pi}^k = \arg\max_\pi \hat{V}^\pi_{\hat{P}^k,\hat{b}^k}$, where $\hat{V}^\pi_{\hat{P}^k,\hat{b}^k}$ is defined as in Equation (1).
8: **Output:** Model $\hat{P}^k$ and exploration policy $\{\tilde{\pi}^k\}$.

---

**Step 3: Target Policy Update with Periodic Restart:** The agent evaluates the target policy by computing the value function under the target policy and the estimated transition kernel. Then, due to the nonstationarity, the target policy is reset every $\tau$ rounds. Compared with the previous work using periodic restart (Fei et al., 2020), whose choice of $\tau$ is based on a certain smooth visitation assumption, we remove such an assumption and hence our choice of $\tau$ is applicable to more general model classes. Finally, the agent uses the estimated value function for target policy update for the next round $k + 1$ via online mirror descend. The update step is inspired by the previous works (Cai et al., 2020; Schulman et al., 2017). Specifically, for any given policy $\pi^0$ and MDP $\mathcal{M}$, define the following function w.r.t. policy $\pi$:

$$
L^{\mathcal{M},\pi_0}(\pi) = V^{\pi^0}_{P,r} + \sum_{h=1}^H \mathbb{E}_{s_h\sim(P,\pi^0)}\left[\left\langle Q^{\pi^0}_{h,P,r}, \pi_h(\cdot|s_h) - \pi^0_h(\cdot|s_h)\right\rangle\right].
$$

$L^{\mathcal{M},\pi_0}(\pi)$ can be regarded as a local linear approximation of $V_{P,r}^\pi$ at "point" $\pi^0$ (Schulman et al., 2017). Consider the following optimization problem:

$$\pi^{k+1} = \arg\max_\pi L^{\mathcal{M}^k,\pi^k}(\pi) - \frac{1}{\eta} \sum_{h \in [H]} \mathbb{E}_{s_h \sim (P^{\star,k},\pi^k)} \left[ D_{KL}(\pi_h(\cdot|s_h) \| \pi_h^k(\cdot|s_h)) \right].$$

This can be regarded as a mirror descent step with KL divergence, where the KL divergence regularizes $\pi$ to be close to $\pi^k$. It further admits a closed-form solution: $\pi_h^{k+1}(\cdot|\cdot) \propto \pi_h^k(\cdot|\cdot) \cdot \exp\{\eta \cdot Q_{h,P^{\star,k},r^k}^{\pi^k}(\cdot,\cdot)\}$. We use the estimated version $\hat{Q}_h^k$ to approximate $Q_{h,P^{\star,k},r^k}^{\pi^k}$ and get

$$\pi_h^{k+1}(\cdot|\cdot) \propto \pi_h^k(\cdot|\cdot) \cdot \exp\{\eta \cdot \hat{Q}_h^k(\cdot,\cdot)\}. \tag{2}$$

### 4.2 Technical Assumptions

Our analysis adopts the following standard assumptions on low-rank MDPs.

**Assumption 4.1.** (Realizability). A learning agent can access to a model class $\{(\Phi, \Psi)\}$ that contains the true model, namely, for any $h \in [H], k \in [K], \phi_h^{\star,k} \in \Phi, \mu_h^{\star,k} \in \Psi$.

While we assume cardinality of the model class to be finite for simplicity, extensions to infinite classes with bounded statistical complexity are not difficult (Sun et al., 2019).

**Assumption 4.2** (Bounded Density). Any model induced by $\Phi$ and $\Psi$ has bounded density, i.e. $\forall P = \langle \phi, \mu \rangle, \phi \in \Phi, \mu \in \Psi$, there exists a constant $B \geq 0$ such that $\max_{(s,a,s') \in \mathcal{S} \times \mathcal{A} \times \mathcal{S}} P(s'|s,a) \leq B$.

**Assumption 4.3** (Reachability). For each round $k$ and step $h$, the true transition kernel $P_h^{\star,k}$ satisfies that for any $(s,a,s') \in \mathcal{S} \times \mathcal{A} \times \mathcal{S}, P_h^{\star,k}(s'|s,a) \geq p_{\min}$.

**Variation Budgets:** We next introduce several measures of nonstationarity of the environment: $\Delta^P = \sum_{k=1}^K \sum_{h=1}^H \max_{(s,a) \in \mathcal{S} \times \mathcal{A}} \|P_h^{\star,k+1}(\cdot|s,a) - P_h^{\star,k}(\cdot|s,a)\|_{TV}, \Delta^{\sqrt{P}} = \sum_{k=1}^K \sum_{h=1}^H \max_{(s,a) \in \mathcal{S} \times \mathcal{A}} \|P_h^{\star,k+1}(\cdot|s,a) - P_h^{\star,k}(\cdot|s,a)\|_{TV}^{1/2}, \Delta^\phi = \sum_{k=1}^K \sum_{h=1}^H \max_{(s,a) \in \mathcal{S} \times \mathcal{A}} \|\phi_h^{\star,k+1}(s,a) - \phi_h^{\star,k}(s,a)\|_2, \Delta^\tau = \sum_{k=1}^K \sum_{h=1}^H \max_{s \in \mathcal{S}} \|\pi_h^{\star,k}(\cdot|s) - \pi_h^{\star,k-1}(\cdot|s)\|_{TV}$. These notions are known as *variation budgets* or *path lengths* in the literature of online convex optimization (Besbes et al., 2015; Hazan, 2016; Hall & Willett, 2015) and nonstationary RL (Fei et al., 2020; Zhong et al., 2021; Zhou et al., 2020). The regret of nonstationary RL naturally depends on these notions that capture the variations of MDP models over time.

### 4.3 Theoretical Guarantee

To present our theoretical result for PORTAL, we first discuss technical challenges in our analysis and the novel tools that we develop. Generally, large nonstationarity of environment can cause significant errors to MLE, empirical covariance and exploration-driven bonus design for low-rank models. Thus, different from static low-rank MDPs (Agarwal et al., 2020b; Uehara et al., 2022), we devise several new techniques in our analysis to capture the errors caused by nonstationarity which we summarize below.

1. Characterizing nonstationary MLE guarantee. We provide a theoretical ground to support our design of leveraging history data collected under various different transition kernels in previous rounds for benefiting the estimation of the current model, which is somewhat surprising. Specifically, we establish an MLE guarantee of the model estimation error, which features a separation of variation budgets from a diminishing term as the estimation sample size $W$ increases. Such a result justifies the usage of data generated by mismatched distributions within a certain window as long as the variation budgets is mild. Such a separation cannot be shown directly. Instead, we bridge the bound of model estimation error and the expected value of the ratio of transition kernels via Hellinger distance, and the latter can be decomposed into the variation budgets and a diminishing term as the estimation sample size increases.

2. Establishing trajectory-wise uncertainty for estimation error $\hat{V}_{\hat{P}^k,\hat{b}^k}^\pi$. To this end, straightforward combination of our nonstationary MLE guarantee with previous techniques on low-rank MDPs would yield a coefficient $\tilde{\alpha}_{k,W}$ that depends on local variation budgets. Instead, we convert the

$\ell_\infty$-norm variation budgets $\Delta^P$ to square-root $\ell_\infty$-norm variation budget $\Delta^{\sqrt{P}}$. In this way, the coefficient no longer depends on the local variation budgets, and the estimation error can be upper bounded by $\hat{V}^\pi_{\hat{P}^k,\hat{b}^k}$ plus an error term only depending on the square-root $\ell_\infty$-norm variation budgets.

3. Error tracking via auxiliary anchor representation. In proving the convergence of average of $\hat{V}^\pi_{\hat{P}^k,\hat{b}^k}$, standard elliptical potential based analysis cannot work, because the representation $\phi^{\star,k}$ in the elliptical potential $\left\|\phi^{\star,k}(s,a)\right\|_{(U^{k,W}_{h,\phi})^{-1}}$ changes across rounds, where $U^{k,W}_{h,\phi}$ is the population version of $\hat{U}^{k,W}_h$. To deal with this challenge, in our analysis, we divide the total $K$ rounds into blocks with equal length of $W$ rounds. Then for each block, we set an *auxiliary anchor representation*. We keep track of the elliptical potential functions using the anchor representation within each block, and control the errors by using anchor representation via variation budgets.

The following theorem characterizes theoretical performance for PORTAL.

**Theorem 4.4.** $\{\mathcal{M}^k\}^K_{k=1}$ *is set of low-rank MDPs with dimension* $d$. *Under Assumptions 4.1 to 4.3, set* $\tilde{\alpha}_{k,W} = \tilde{O}\left(\sqrt{A+d^2}\right)$ *and* $\lambda_{k,W} = \tilde{O}(d)$. *Let* $\{\pi^k\}^K_{k=1}$ *be the output of PORTAL in Algorithm 1. For any* $\delta \in (0,1)$, *with probability at least* $1-\delta$, $\mathrm{Gap}_{\mathrm{Ave}}(K)$ *of PORTAL is at most*

$$\tilde{O}\bigg(\underbrace{\sqrt{\frac{H^4 d^2 A}{W}\left(A+d^2\right)} + \sqrt{\frac{H^3 dA}{K}\left(A+d^2\right)W^2\Delta^\phi} + \sqrt{\frac{H^2 W^3 A}{K^2}}\Delta^{\sqrt{P}}}_{(I)} + \underbrace{\frac{H}{\sqrt{\tau}} + \frac{H\tau}{K}(\Delta^P + \Delta^\pi)}_{(II)}\bigg).$$
(3)

We explain the upper bound in Theorem 4.4 as follows. The basic bound as Equation (3) in Theorem 4.4 contains two parts: the first part $(I)$ captures the estimation error for evaluating the target policy under the true environment via the estimated value function $\hat{Q}^k$ as in line 16 of Algorithm 1. Hence, part $(I)$ decreases with $K$ and increases with the nonstationarity of transition kernels and representation functions. Also, part $(I)$ is greatly affected by the window size $W$, which is determined by the dataset used to estimate the transition kernels. Typically, $W$ is tuned carefully based on the variation of environment. If the environment changes significantly, then the samples far in the past are obsolete and become not very informative for estimating transition kernels. The second part $(II)$ captures the approximation error arising in finding the optimal policy via the policy optimization method as in line 8 of Algorithm 1. Due to the nonstationarity of the environment, the optimal policy keeps changing across rounds, and hence the nonstationarity of optimal policy $\Delta^\pi$ affects the approximation error. Similarly to the window size $W$ in part $(I)$, the policy restart period $\tau$ can also be tuned carefully based on the variation of environment and the optimal policies.

**Corollary 4.5.** *Under the same conditions of Theorem 4.4, if the variation budgets are known, then we can select the hyper-parameters correspondingly to achieve optimality. Specially, if the nonstationarity of the environment is moderate, we have*

$$\mathrm{Gap}_{\mathrm{Ave}}(K) \leq \tilde{O}(H^{\frac{11}{6}} d^{\frac{5}{6}} A^{\frac{1}{2}} \left(A+d^2\right)^{\frac{1}{2}} K^{-\frac{1}{6}}(\Delta^{\sqrt{P}} + \Delta^\phi)^{\frac{1}{6}} + 2HK^{-\frac{1}{3}}(\Delta^P + \Delta^\pi)^{\frac{1}{3}}).$$

*If the environment is near stationary, then the best* $W$ *and* $\tau$ *are* $K$. *The* $\mathrm{Gap}_{\mathrm{Ave}}$ *reduces to* $\tilde{O}(\sqrt{H^4 d^2 A(A+d^2)/K})$, *which matches results of stationary environment (Uehara et al., 2022).*

Detailed discussions and proofs of Theorem 4.4 and Corollary 4.5 are provided in Appendices A and B, respectively.

# 5 Parameter-free Algorithm: Ada-PORTAL

As shown in Theorem 4.4, hyper-parameters $W$ and $\tau$ greatly affect the performance of Algorithm 1. With prior knowledge of variation budgets, the agent is able to optimize the performance as in Corollary 4.5. However, in practice, variation budgets are unknown to the agent. To deal with this issue, in this section, we present a parameter-free algorithm called Ada-PORTAL in Algorithm 3, which is able to tune hyper-parameters without knowing the variation budgets beforehand.

Algorithm 3 is inspired by the BORL method (Cheung et al., 2020). The idea is to use Algorithm 1 as a subroutine and treat the selection of the hyper-parameters such as $W$ and $\tau$ as a bandit problem.

---

**Algorithm 3** Ada-PORTAL (**Ada**ptive **P**olicy **O**ptimization **R**epresen**TA**tion **L**earning)

---

1: **Input:** Confidence level $\delta$, number of episodes $K$, block length $M$, feasible set of window size $\mathcal{J}_W$ and policy restart period $\mathcal{J}_\tau$.

2: **Initialization:** Initialize $\alpha, \beta, \gamma$ and $\{q_{l,1}\}_{l \in [J]}$ as in Equation (4).

3: **for** block $i = 1, \ldots, \lceil K/M \rceil$ **do**

4:      Update the windows size selection distribution $\{u_{(k,l),i}\}_{(k,l) \in \mathcal{J}}$ as in Equation (5).

5:      Sample $(k_i, l_i) \in \mathcal{J}$ from the updated distribution $\{u_{(k,l),i}\}_{(k,l) \in \mathcal{J}}$, then set $W_i = \lfloor M^{k_i/J_M} \rfloor$ and $\tau_i = \lfloor M_\tau^{l_i/J_\tau} \rfloor$.

6:      **for** episode $k = (i-1)M + 1, \ldots, \min\{iM, K\}$ **do**

7:          **Run** PORTAL with $W_i$ and $\tau_i$.

8:      **end for**

9:      Compute the total reward for block $i$ as $R_i(W_i, \tau_i) = \sum_{k=(i-1)M+1}^{\min\{iM,K\}} V_1^k$, where $V_1^k$ is empirical value functions of target policy $\pi^k$, and update the estimated total reward of running different epoch sizes $\{q_{(k,l),i+1}\}_{(k,l) \in \mathcal{J}}$ according to Equation (6).

10: **end for**

11: **Output:** $\{\pi^k\}_{k=1}^K$.

---

Specifically, Ada-PORTAL divides the entire $K$ rounds into $\lceil K/M \rceil$ blocks with equal length of $M$ rounds. Then two sets $\mathcal{J}_W$ and $\mathcal{J}_\tau$ are specified (see later part of this section), from which the window size $W$ and the restart period $\tau$ for each block are drawn.

For each block $i \in \left[ \lceil \frac{K}{M} \rceil \right]$, Ada-PORTAL treats each element of $\mathcal{J}_W \times \mathcal{J}_\tau$ as an arm and take it as a bandit problem to select the best arm for each block. In lines 4 and 5 of Algorithm 3, a master algorithm is run to update parameters and select the desired arm, i.e., the window size $W_i$ and the restart period $\tau_i$. Here we choose EXP3-P (Bubeck & Cesa-Bianchi, 2012) as the master algorithm and discuss the details later. Then Algorithm 1 is called with input $W_i$ and $\tau_i$ as a subroutine for the current block $i$. At the end of each block, the total reward of the current block is computed by summing up all the empirical value functions of the target policy of each episode within the block, which is then used to update the parameters for the next block.

We next set the feasible sets and block length of Algorithm 1. Since optimal $W$ and $\tau$ in PORTAL are chosen differently from previous works (Zhou et al., 2020; Cheung et al., 2019; Touati & Vincent, 2020) on nonstationary MDPs due to the low-rank structure, the feasible set here that covers the optimal choices of $W$ and $\tau$ should also be set differently from those previous works using BORL.

$M_W = d^{\frac{1}{3}} H^{\frac{1}{3}} K^{\frac{1}{3}}$, $M_\tau = K^{\frac{2}{3}}$, $M = d^{\frac{1}{3}} H^{\frac{1}{3}} K^{\frac{2}{3}}$. $J_W = \lfloor \log(M_W) \rfloor$, $\mathcal{J}_W = \{M_W^0, \lfloor M_W^{\frac{1}{J_W}} \rfloor, \ldots, M_W\}$, $J_\tau = \lfloor \log(M_\tau) \rfloor$, $\mathcal{J}_\tau = \{M_\tau^0, \lfloor M_\tau^{\frac{1}{J_\tau}} \rfloor, \ldots, M_\tau\}$, $J = J_W \cdot J_\tau$.

Then the parameters of EXP3-P are intialized as follows:

$$\alpha = 0.95 \sqrt{\frac{\ln J}{J \lceil K/M \rceil}}, \quad \beta = \sqrt{\frac{\ln J}{J \lceil K/M \rceil}}, \gamma = 1.05 \sqrt{\frac{J \ln J}{\lceil K/M \rceil}}, \quad q_{(k,l),1} = 0, (k,l) \in \mathcal{J}, \quad (4)$$

where $\mathcal{J} = \{(k,l) : k \in \{0, 1, \ldots, J_W\}, l \in \{0, 1, \ldots, J_\tau\}\}$. The parameter updating rule is as follows. For any $(k,l) \in \mathcal{J}, i \in \lceil K/M \rceil$,

$$u_{(k,l),i} = (1 - \gamma) \frac{\exp(\alpha q_{(k,l),i})}{\sum_{(k,l) \in \mathcal{J}} \exp(\alpha q_{(k,l),i})} + \frac{\gamma}{J}, \quad (5)$$

where $u_{(k,l),i}$ is a probability over $\mathcal{J}$. From $u_{(k,l),i}$, the agent samples a desired pair $(k_i, l_i)$ for each block $i$, which corresponds to the index of feasible set $\mathcal{J}_W \times \mathcal{J}_\tau$ and is used to set $W_i$ and $\tau_i$.

As a last step, $R_i(W_i, \tau_i)$ is rescaled to update $q_{(k,l),i+1}$.

$$q_{(k,l),i+1} = q_{(k,l),i} + \frac{\beta + 1_{(k,l)=(k_i,l_i)} R_i(W_i, \tau_i)/M}{u_{(k,l),i}}. \quad (6)$$

We next establish a bound on $\text{Gap}_{\text{Ave}}$ for Ada-PORTAL.

**Theorem 5.1.** *Under the same conditions of Theorem 4.4, with probability at least $1 - \delta$, the average dynamic suboptimality gap of Ada-PORTAL in Algorithm 3 is upper-bounded as $\mathrm{Gap}_{\mathrm{Ave}}(K) \leq \tilde{O}(H^{\frac{11}{6}} d^{\frac{5}{6}} A^{\frac{1}{2}} (A + d^2)^{\frac{1}{2}} K^{-\frac{1}{6}} (\Delta^{\sqrt{P}} + \Delta^{\phi} + 1)^{\frac{1}{6}} + 2HK^{-\frac{1}{3}} (\Delta^{P} + \Delta^{\pi} + 1)^{\frac{1}{3}}).$*

**Comparison with Wei & Luo (2021):** It is interesting to compare Ada-PORTAL with an alternative black-box type of approach to see its advantage. A black-box technique called MASTER was proposed in Wei & Luo (2021), which can also work with any base algorithm such as PORTAL to handle unknown variation budgets. Such a combined approach of MASTER+PORTAL turns out to have a worse $\mathrm{Gap}_{\mathrm{Ave}}$ than our Ada-PORTAL in Algorithm 3. To see this, denote $\Delta = \Delta^{\phi} + \Delta^{\sqrt{P}} + \Delta^{\pi}$. The $\mathrm{Gap}_{\mathrm{Ave}}$ of MASTER+PORTAL is $\tilde{O}(K^{-\frac{1}{6}} \Delta^{\frac{1}{3}})$. Then if variation budgets are not too small, i.e. $\Delta \geq \tilde{O}(1)$, this $\mathrm{Gap}_{\mathrm{Ave}}$ is worse than Ada-PORTAL. See detailed discussion in Appendix C.2.

# 6   Conclusion

In the paper, we investigate nonstationary RL under low-rank MDPs. We first propose a notion of average dynamic suboptimality gap $\mathrm{Gap}_{\mathrm{Ave}}$ to evaluate the performance of a series of policies in a nonstationary environment. Then we propose a sample-efficient policy optimization algorithm PORTAL and its parameter-free version Ada-PORTAL. We further provide upper bounds on $\mathrm{Gap}_{\mathrm{Ave}}$ for both algorithms. As future work, it is interesting to investigate the impact of various constraints such as safety requirements in nonstationary RL under function approximations.

# 7   Acknowledgement

The work of Y. Liang was supported in part by the U.S. National Science Foundation under the grants RINGS-2148253, DMS-2134145, and ECCS-2113860. The work of J. Yang was supported by the U.S. National Science Foundation under the grants CNS-1956276 and CNS-2003131.

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

# Supplementary Materials

## A   Proof of Theorem 4.4

We summarize frequently used notations in the following list.

$$
\begin{array}{ll}
\zeta_{k,W} & \frac{2\log(2|\Phi||\Psi|kH/\delta)}{W} \\[4pt]
\lambda_{k,W} & O(d\log(|\Phi|\min\{k,W\}TH/\delta)) \\[4pt]
\alpha_{k,W} & \sqrt{2WA\zeta_{k,W}+\lambda_{k,W}d}=O(\sqrt{4A\log(2|\Phi||\Psi|kH/\delta)+\lambda_{k,W}d}) \\[4pt]
\tilde{\alpha}_{k,W} & 5\sqrt{2WA\zeta_{k,W}+\lambda_{k,W}d} \\[4pt]
\beta_{k,W} & \sqrt{9dA(2WA\zeta_{k,W}+\lambda_{k,W}d)+\lambda_{k,W}d} \\[4pt]
\eta & \sqrt{\frac{L\log A}{K}} \\[4pt]
\Delta_{\mathcal{H},\mathcal{I}}^{P} & \sum_{h\in\mathcal{H}}\sum_{i\in\mathcal{I}}\max_{(s,a)\in\mathcal{S}\times\mathcal{A}}\left\| P_h^{\star,i+1}(\cdot|s,a)-P_h^{\star,i}(\cdot|s,a)\right\|_{TV} \\[4pt]
\Delta_{\mathcal{H},\mathcal{I}}^{\sqrt{P}} & \sum_{h\in\mathcal{H}}\sum_{i\in\mathcal{I}}\max_{(s,a)\in\mathcal{S}\times\mathcal{A}}\sqrt{\left\| P_h^{\star,i+1}(\cdot|s,a)-P_h^{\star,i}(\cdot|s,a)\right\|_{TV}} \\[4pt]
\Delta_{\mathcal{H},\mathcal{I}}^{\phi} & \sum_{h\in\mathcal{H}}\sum_{i\in\mathcal{I}}\max_{(s,a)\in\mathcal{S}\times\mathcal{A}}\left\| \phi_h^{\star,i+1}(s,a)-\phi_h^{\star,i}(s,a)\right\|_{2} \\[4pt]
\Delta_{\mathcal{H},\mathcal{I}}^{r} & \sum_{h\in\mathcal{H}}\sum_{i\in\mathcal{I}}\max_{(s,a)\in\mathcal{S}\times\mathcal{A}}\left\| r_h^{\star,i+1}(s,a)-r_h^{\star,i}(s,a)\right\|_{2} \\[4pt]
\Delta_{\mathcal{H},\mathcal{I}}^{\pi} & \sum_{h\in\mathcal{H}}\sum_{i\in\mathcal{I}}\max_{s\in\mathcal{S}}\left\| \pi_h^{\star,i}(\cdot|s)-\pi_h^{\star,i-1}(\cdot|s)\right\|_{TV} \\[4pt]
f_h^k(s,a) & \|\hat{P}_h^k(\cdot|s,a)-P_h^{\star,k}(\cdot|s,a)\|_{TV} \\[4pt]
U_{h,\phi}^{k,W} & \sum_{i=1\vee k-W}^{k-1}\mathbb{E}_{s_h\sim(P^{\star,i},\tilde{\pi}^i),a_h\sim\mathcal{U}(\mathcal{A})}\left[\phi(s_h,a_h)(\phi(s_h,a_h))^{\top}\right]+\lambda_{k,W}I_d \\[4pt]
\hat{U}_h^{k,W} & \sum_{\tilde{\mathcal{D}}_h^{(k,h+1)}}\hat{\phi}_h(s_h,a_h)\hat{\phi}_h(s_h,a_h)^{\top}+\lambda_{k,W}I_d \\[4pt]
W_{h,\phi}^{k,W} & \sum_{i=1\vee k-W}^{k-1}\mathbb{E}_{(s_h,a_h)\sim(P^{\star,i},\tilde{\pi}^i)}\left[\phi(s_h,a_h)(\phi(s_h,a_h))^{\top}\right]+\lambda_{k,W}I_d \\[4pt]
b_h^k & \min\left\{\alpha_{k,W}\left\|\hat{\phi}_h^k(s_h,a_h)\right\|_{(U_{h,\hat{\phi}^k}^{k,W})^{-1}},1\right\} \\[4pt]
\hat{b}_h^k & \min\left\{\tilde{\alpha}_{k,W}\left\|\hat{\phi}_h^k(s_h,a_h)\right\|_{(\hat{U}_h^{k,W})^{-1}},1\right\}
\end{array}
$$

*Proof Sketch of Theorem 4.4.*  The proof contains the following three main steps.

**Step 1 (Appendix A.1):** We first decompose the average dynamic suboptimal gap into three terms as in Lemma A.1, which can be divided into two parts: one part corresponds to the model estimation error and the other part corresponds to the performance difference between the target policy chosen by the agent and optimal policy. We then bound the two parts separately.

**Step 2 (Appendix A.2):** For the first part corresponding to model estimation error, first by Lemmas A.13 and A.15, we show that the model estimation error can be bounded by the average of the truncated value functions of the bonus terms i.e. $\frac{1}{K}\sum_{k=1}^{K}\hat{V}_{\hat{P}^k,\hat{b}^k}^{\pi}$ plus a term w.r.t. variation budgets. We then upper bound the average of $\hat{V}_{\hat{P}^k,\hat{b}^k}^{\pi}$ as in Lemma A.18. To this end, we divide the total $K$ rounds into blocks with equal length of $W$ and adopt an auxiliary anchor representation for each block to deal with the challenge arising from time-varying representations when using standard elliptical potential based methods.

**Step 3 (Appendix A.3):** For the second part corresponding to the performance difference bound, similarly to dealing with changing representations, since the optimal policy changes over time, we adopt an auxiliary anchor policy and decompose the performance difference bound into two terms as in Equation (10) and bound the two terms separately.  $\square$

We further note that the above analysis techniques can also be applied to RL problems where model mis-specification exists, i.e. $\phi^{\star,k}\notin\Phi,\mu^{\star,k}\notin\Psi$.

**Organization of the Proof for Theorem 4.4.** Our proof of Theorem 4.4 is organized as follows. In Appendix A.1, we provide the decomposition of the average dynamic suboptimality gap $\mathrm{Gap}_{\mathrm{Ave}}$ in Equation (7); in Appendix A.2, we bound the first and third terms of $\mathrm{Gap}_{\mathrm{Ave}}$; in Appendix A.3,

we bound the second term of $\mathrm{Gap_{Ave}}$, and in Appendix A.4 we combine our results to complete the proof of Theorem 4.4. We provide all the supporting lemmas in Appendix A.5.

## A.1 Average Suboptimaility Gap Decomposition

**Lemma A.1.** *We denote* $\pi_h^{\star,k}(\cdot|s) = \arg\max_\pi V_{P^{\star,k},r^k}^\pi$. *Then the average dynamic suboptimality gap has the following decomposition:*

$$
\begin{aligned}
\mathrm{Gap_{Ave}}(K) &= \frac{1}{K}\sum_{k=1}^K V_{P^{\star,k},r^k}^\star - V_{P^{\star,k},r^k}^{\pi^k} \\
&= \frac{1}{K}\sum_{k=1}^K \sum_{h\in[H]} \mathbb{E}_{(s_h,a_h)\sim(P^{\star,k},\pi^{\star,k})}\left[\left\{P_h^{\star,k} - \hat{P}_h^k\right\}V_{h+1,\hat{P}^k,r^k}^{\pi^k}\right] \\
&\quad + \frac{1}{K}\sum_{k=1}^K \sum_{h\in H} \mathbb{E}_{s_h\sim(P^{\star,k},\pi^{\star,k})}\left[\langle Q_{h,\hat{P}^k,r^k}^{\pi^k}(s_h,\cdot), \pi_h^{\star,k}(\cdot|s_h) - \pi_h^k(\cdot|s_h)\rangle\right] \\
&\quad + \frac{1}{K}\sum_{k=1}^K V_{\hat{P}^k,r^k}^{\pi^k} - V_{P^{\star,k},r^k}^{\pi^k} \quad\quad\quad\quad (7)
\end{aligned}
$$

*Proof.* For any function $f : \mathcal{S}\times\mathcal{A}\to\mathbb{R}$ and any $(k,h,s)\in[K]\times[H]\times\mathcal{S}$, define the following operators:

$$
(\mathbb{J}_{k,h}^\star f)(s) = \langle f(s,\cdot), \pi_h^{\star,k}(\cdot|s)\rangle, \quad (\mathbb{J}_{k,h}f)(s) = \langle f(s,\cdot),\pi_h^k(\cdot|s)\rangle.
$$

We next consider the following decomposition:

$$
V_{P^{\star,k},r^k}^\star - V_{P^{\star,k},r^k}^{\pi^k} = \underbrace{V_{P^{\star,k},r^k}^\star - V_{\hat{P}^k,r^k}^{\pi^k}}_{G_1} + V_{\hat{P}^k,r^k}^{\pi^k} - V_{P^{\star,k},r^k}^{\pi^k}, \quad\quad\quad (8)
$$

The term $G_1$ can be bounded as follows:

$$
\begin{aligned}
G_1 &= V_{P^{\star,k},r^k}^\star - V_{\hat{P}^k,r^k}^{\pi^k} \\
&= \left(\mathbb{J}_{k,1}^\star Q_{1,P^{\star,k},r^k}^{\pi^{\star,k}}\right) - \left(\mathbb{J}_{k,1}Q_{1,\hat{P}^k,r^k}^{\pi^k}\right) \\
&= (\mathbb{J}_{k,1}^\star(Q_{1,P^{\star,k},r^k}^{\pi^{\star,k}} - Q_{1,\hat{P}^k,r^k}^{\pi^k})) + ((\mathbb{J}_{k,1}^\star - \mathbb{J}_{k,1})Q_{1,\hat{P}^k,r^k}^{\pi^k}) \\
&= (\mathbb{J}_{k,1}^\star(r_1^k(s,\cdot) + P_1^{\star,k}V_{2,P^{\star,k},r^k}^{\pi^{\star,k}} - (r_1^k(s,\cdot) + \hat{P}_1^k V_{2,\hat{P}^k,r^k}^{\pi^k}))) + ((\mathbb{J}_{k,1}^\star - \mathbb{J}_{k,1})Q_{1,\hat{P}^k,r^k}^{\pi^k}) \\
&= (\mathbb{J}_{k,1}^\star(P_1^{\star,k}V_{2,P^{\star,k},r^k}^{\pi^{\star,k}} - \hat{P}_1^k V_{2,\hat{P}^k,r^k}^{\pi^k})) + ((\mathbb{J}_{k,1}^\star - \mathbb{J}_{k,1})Q_{1,\hat{P}^k,r^k}^{\pi^k}) \\
&= \left(\mathbb{J}_{k,1}^\star\left(P_1^{\star,k}\left\{V_{2,P^{\star,k},r^k}^{\pi^{\star,k}} - V_{2,\hat{P}^k,r^k}^{\pi^k}\right\} + \left\{P_1^{\star,k} - \hat{P}_1^k\right\}V_{2,\hat{P}^k,r^k}^{\pi^k}\right)\right) + ((\mathbb{J}_{k,1}^\star - \mathbb{J}_{k,1})Q_{1,\hat{P}^k,r^k}^{\pi^k}) \\
&= \left(\mathbb{J}_{k,1}^\star\{P_1^{\star,k} - \hat{P}_1^k\}V_{2,\hat{P}^k,r^k}^{\pi^k}\right) + \mathbb{E}_{s_2\sim(P^{\star,k},\pi^{\star,k})}\left[V_{2,P^{\star,k},r^k}^{\pi^{\star,k}}(s_2) - V_{2,\hat{P}^k,r^k}^{\pi^k}(s_2)\right] + ((\mathbb{J}_{k,1}^\star - \mathbb{J}_{k,1})Q_{1,\hat{P}^k,r^k}^{\pi^k}) \\
&= \sum_{h\in[H]}\mathbb{E}_{(s_h,a_h)\sim(P^{\star,k},\pi^{\star,k})}\left[\left\{P_h^{\star,k} - \hat{P}_h^k\right\}V_{h+1,\hat{P}^k,r^k}^{\pi^k}\right] \\
&\quad + \sum_{h\in[H]}\mathbb{E}_{s_h\sim(P^{\star,k},\pi^{\star,k})}\left[\langle Q_{h,\hat{P}^k,r^k}^{\pi^k}(s_h,\cdot), \pi_h^{\star,k}(\cdot|s_h) - \pi_h^k(\cdot|s_h)\rangle\right] \quad\quad (9)
\end{aligned}
$$

Substituting the above result to Equation (8) completes the proof. $\square$

## A.2 First and Third Terms of $\mathrm{Gap_{Ave}}$ in Equation (7): Model Estimation Error Bound

### A.2.1 First Term in Equation (7)

**Lemma A.2.** *With probability at least* $1 - \delta$, *we have*

$$
\frac{1}{K}\sum_{k=1}^K \sum_{h\in[H]}\mathbb{E}_{(s_h,a_h)\sim(P^{\star,k},\pi^{\star,k})}\left[\left\{P_h^{\star,k} - \hat{P}_h^k\right\}V_{h+1,\hat{P},r}^{\pi^k,k}\right]
$$

$$\leq O\left(\frac{H}{K}\left[\sqrt{KdA(A\log(|\Phi||\Psi|KH/\delta)+d^2)}\left[H\sqrt{\frac{Kd}{W}\log(W)}+\sqrt{HW^2\Delta^{\phi}_{[H],[K]}}\right]+\sqrt{W^3AC_B}\Delta^{\sqrt{P}}_{[H],[K]}\right]\right).$$

*Proof.* We proceed the proof by deriving the bound:

$$\frac{1}{K}\sum_{k=1}^{K}\sum_{h\in[H]}\mathbb{E}_{(s_h,a_h)\sim(P^{\star,k},\pi^{\star,k})}\left[\left\{P_h^{\star,k}-\hat{P}_h^k\right\}V_{h+1,\hat{P},r}^{\pi^k,k}\right]$$

$$\leq\frac{1}{K}\sum_{k=1}^{K}\sum_{h\in[H]}\mathbb{E}_{(s_h,a_h)\sim(P^{\star,k},\pi^{\star,k})}\left[f_h^k(s_h,a_h)\right]$$

$$=\frac{1}{K}\sum_{k=1}^{K}\mathbb{E}_{a_1\sim\pi^{\star,k}}\left[f_1^k(s_1,a_1)\right]+\frac{1}{K}\sum_{k=1}^{K}\sum_{h=2}^{H}\mathbb{E}_{(s_h,a_h)\sim(P^{\star,k},\pi^{\star,k})}\left[f_h^k(s_h,a_h)\right]$$

$$=\frac{1}{K}\sum_{k=1}^{K}\mathbb{E}_{a_1\sim\pi^{\star,k}}\left[f_1^k(s_1,a_1)\right]+\frac{1}{K}\sum_{k=1}^{K}\sum_{h=2}^{H}\mathbb{E}_{(s_h,a_h)\sim(\hat{P}^k,\pi^{\star,k})}\left[f_h^k(s_h,a_h)\right]$$

$$+\frac{1}{K}\sum_{k=1}^{K}\sum_{h=2}^{H}\left\{\mathbb{E}_{(s_h,a_h)\sim(P^{\star,k},\pi^{\star,k})}\left[f_h^k(s_h,a_h)\right]-\mathbb{E}_{(s_h,a_h)\sim(\hat{P}^k,\pi^{\star,k})}\left[f_h^k(s_h,a_h)\right]\right\}$$

$$\overset{(i)}{\leq}\frac{2}{K}\sum_{h=2}^{H}\sum_{k=1}^{K}\hat{V}_{\hat{P}^k,\hat{b}^k}^{\pi^{\star,k}}+2\sum_{h=2}^{H}\left[\frac{1}{K}\sum_{k=1}^{K}\sqrt{WA\left(\zeta_{k,W}+\frac{1}{2}C_B\Delta^P_{1,[k-W,k-1]}\right)}\right.$$

$$\left.+\frac{1}{K}\sum_{k=1}^{K}\sum_{h'=2}^{h}\sqrt{\frac{1}{2d}WAC_B\Delta^P_{[h'-1,h'],[k-W,k-1]}}\right]$$

$$\leq\frac{2H}{K}\sum_{k=1}^{K}\hat{V}_{\hat{P}^k,\hat{b}^k}^{\tilde{\pi}^k}+\frac{2H}{K}\left[\sum_{k=1}^{K}\sqrt{WA\left(\zeta_{k,W}+\frac{1}{2}C_B\Delta^P_{1,[k-W,k-1]}\right)}\right.$$

$$\left.+\sum_{k=1}^{K}\sum_{h=2}^{H}\sqrt{\frac{1}{2d}WAC_B\Delta^P_{[h-1,h],[k-W,k-1]}}\right]$$

$$\overset{(ii)}{\leq}\frac{2H}{K}\sum_{k=1}^{K}\hat{V}_{\hat{P}^k,\hat{b}^k}^{\tilde{\pi}^k}+\frac{2H}{K}\sum_{k=1}^{K}\sqrt{WA\zeta_{k,W}}+\frac{2H}{K}\sqrt{W^3AC_B}\Delta^{\sqrt{P}}_{[H],[K]}$$

$$\overset{(iii)}{\leq}O\left(\frac{H}{K}\left[\sqrt{KdA(A\log(|\Phi||\Psi|KH/\delta)+d^2)}\left[H\sqrt{\frac{Kd}{W}\log(W)}+\sqrt{HW^2\Delta^{\phi}_{[H],[K]}}\right]+\sqrt{W^3AC_B}\Delta^{\sqrt{P}}_{[H],[K]}\right]\right),$$

where $(i)$ follows from Lemmas A.13 and A.15, $(ii)$ follows because $\sqrt{a+b}\leq\sqrt{a}+\sqrt{b},\forall a,b\geq 0$ and $\sum_{k=1}^{K}\Delta^{\sqrt{P}}_{\{h\},[k-W,k-1]}\leq W\Delta^{\sqrt{P}}_{\{h\},[K]}$, and $(iii)$ follows from Lemma A.18. □

### A.2.2 Third Term in Equation (7)

**Lemma A.3.** *With probability at least* $1-\delta$*, we have*

$$\frac{1}{K}\sum_{k=1}^{K}\left[V_{\hat{P}^k,r^k}^{\pi^k}-V_{P^{\star,k},r^k}^{\pi^k}\right]$$

$$\leq O\left(\frac{1}{K}\sqrt{KdA(A\log(|\Phi||\Psi|KH/\delta)+d^2)}\left[H\sqrt{\frac{Kd}{W}\log(W)}+\sqrt{HW^2\Delta^{\phi}_{[H],[K]}}\right]+\sqrt{W^3AC_B}\Delta^{\sqrt{P}}_{[H],[K]}\right).$$

*Proof.* We define the model error as $f_h^k(s_h,a_h)=\left\|P_h^{\star}(\cdot|s_h,a_h)-\hat{P}_h(\cdot|s_h,a_h)\right\|_{TV}$. We next derive the following bound:

$$\frac{1}{K}\sum_{k=1}^{K}\left[V_{\hat{P}^k,r^k}^{\pi^k}-V_{P^{\star,k},r^k}^{\pi^k}\right]$$

$$\overset{(i)}{\leq}\frac{1}{K}\sum_{k=1}^{K}\hat{V}_{\hat{P}^k,\hat{b}^k}^{\pi^k}+\frac{1}{K}\sum_{k=1}^{K}\sum_{h=2}^{H}\sqrt{\frac{3}{\lambda_W}WAC_B\Delta^P_{\{h-1,h\},[k-W,k-1]}}$$

$$+ \frac{1}{K} \sum_{k=1}^{K} \sqrt{A \left( \zeta_{k,W} + 2C_B \Delta^P_{1,[k-W,k-1]} \right)}$$

$$\overset{(ii)}{\leq} \frac{1}{K} \sum_{k=1}^{K} \hat{V}^{\tilde{\pi}^k}_{\hat{P}^k, \hat{b}^k} + \frac{1}{K} \sum_{k=1}^{K} \sum_{h=2}^{H} \sqrt{\frac{3}{\lambda_W} W A C_B \Delta^P_{\{h-1,h\},[k-W,k-1]}}$$

$$+ \frac{1}{K} \sum_{k=1}^{K} \sqrt{A \left( \zeta_{k,W} + 2C_B \Delta^P_{1,[k-W,k-1]} \right)}$$

$$\overset{(iii)}{\leq} O \left( \frac{1}{K} \sqrt{K d A (A \log(|\Phi||\Psi|KH/\delta) + d^2)} \left[ H \sqrt{\frac{Kd}{W} \log(W)} + \sqrt{HW^2 \Delta^\phi_{[H],[K]}} \right] \right.$$

$$\left. + \sqrt{W^3 A C_B} \Delta^{\sqrt{P}}_{[H],[K]} \right),$$

where $(i)$ follows from Lemma A.15, $(ii)$ follows from the definition of $\tilde{\pi}^k$, and $(iii)$ follows from Lemma A.18. $\square$

### A.3 Second Term of $\mathrm{Gap}_{\mathrm{Ave}}$ in Equation (7): Performance Difference Bound

The second term in Lemma A.1 $\frac{1}{K} \sum_{k=1}^{K} \sum_{h \in H} \mathbb{E}_{s_h \sim (P^{\star,k}, \pi^{\star,k})}[\langle Q^{\pi^k}_{h,\hat{P}^k,r^k}(s_h,\cdot), \pi^{\star,k}_h(\cdot|s_h) - \pi^k_h(\cdot|s_h)\rangle]$ can be further decomposed as

$$\frac{1}{K} \sum_{k=1}^{K} \sum_{h \in H} \mathbb{E}_{s_h \sim (P^{\star,k}, \pi^{\star,k})} \left[ \langle Q^{\pi^k}_{h,\hat{P}^k,r^k}(s_h,\cdot), \pi^{\star,k}_h(\cdot|s_h) - \pi^k_h(\cdot|s_h)\rangle \right]$$

$$= \frac{1}{K} \sum_{l \in [L]} \sum_{k=(l-1)\tau+1}^{l\tau} \sum_{h \in [H]} \mathbb{E}_{P^{\star,k}, \pi^{\star,k}} \left[ \langle \hat{Q}^k_h(s_h,\cdot), \pi^{\star,k}_h(\cdot|s_h) - \pi^k_h(\cdot|s_h)\rangle \right]$$

$$= \underbrace{\frac{1}{K} \sum_{l \in [L]} \sum_{k=(l-1)\tau+1}^{l\tau} \sum_{h \in [H]} \mathbb{E}_{P^{\star,(l-1)\tau+1}, \pi^{\star,(l-1)\tau+1}} \left[ \langle \hat{Q}^k_h(s_h,\cdot), \pi^{\star,k}_h(\cdot|s_h) - \pi^k_h(\cdot|s_h)\rangle \right]}_{(a)}$$

$$+ \underbrace{\frac{1}{K} \sum_{l \in [L]} \sum_{k=(l-1)\tau+1}^{l\tau} \sum_{h \in [H]} \left( \mathbb{E}_{P^{\star,k}, \pi^{\star,k}} - \mathbb{E}_{P^{\star,(l-1)\tau+1}, \pi^{\star,(l-1)\tau+1}} \right) \left[ \langle \hat{Q}^k_h(s_h,\cdot), \pi^{\star,k}_h(\cdot|s_h) - \pi^k_h(\cdot|s_h)\rangle \right]}_{(b)}.$$

$$(10)$$

#### A.3.1 Bound $(a)$ in Equation (10)

We first present the following lemma of the descent result introduced in Cai et al. (2020).

**Lemma A.4.** *For any distribution $p^\star$ and $p$ supported on $\mathcal{A}$ and state $s \in \mathcal{S}$, and function $Q : \mathcal{S} \times \mathcal{A} \to [0, H]$, it holds for a distribution $p'$ supported on $\mathcal{A}$ with $p'(\cdot) \propto p(\cdot) \cdot \exp^{\eta \cdot Q(s,\cdot)}$ that*

$$\langle Q(s,\cdot), p^\star(\cdot) - p(\cdot)\rangle \leq \frac{1}{2}\eta + \frac{1}{\eta} \left[ D_{KL}(p^\star(\cdot)\|p(\cdot)) - D_{KL}(p^\star(\cdot)\|p'(\cdot)) \right].$$

**Lemma A.5.** *Term $(a)$ in Equation (10) satisfies the following bound:*

$$\sum_{l \in [L]} \sum_{k=(l-1)\tau+1}^{l\tau} \sum_{h \in [H]} \mathbb{E}_{P^{\star,(l-1)\tau+1}, \pi^{\star,(l-1)\tau+1}} \left[ \langle \hat{Q}^k_h(s_h,\cdot), \pi^{\star,k}_h(\cdot|s_h) - \pi^k_h(\cdot|s_h)\rangle \right]$$

$$\leq \frac{1}{2}\eta KH + \frac{1}{\eta} LH \log A + \tau \Delta^\pi_{[H],[K]}.$$

*Proof.* We first decompose $(a)$ into two parts:

$$\sum_{l\in[L]}\sum_{k=(l-1)\tau+1}^{l\tau}\sum_{h\in[H]}\mathbb{E}_{P^\star,(l-1)\tau+1,\pi^\star,(l-1)\tau+1}\left[\langle\hat{Q}_h^k(s_h,\cdot),\pi_h^{\star,k}(\cdot|s_h)-\pi_h^k(\cdot|s_h)\rangle\right]$$

$$=\underbrace{\sum_{l\in[L]}\sum_{k=(l-1)\tau+1}^{l\tau}\sum_{h\in[H]}\mathbb{E}_{P^\star,(l-1)\tau+1,\pi^\star,(l-1)\tau+1}\left[\langle\hat{Q}_h^k(s_h,\cdot),\pi_h^{\star,(l-1)\tau+1}(\cdot|s_h)-\pi_h^k(\cdot|s_h)\rangle\right]}_{(I)}$$

$$+\underbrace{\sum_{l\in[L]}\sum_{k=(l-1)\tau+1}^{l\tau}\sum_{h\in[H]}\mathbb{E}_{P^\star,(l-1)\tau+1,\pi^\star,(l-1)\tau+1}\left[\langle\hat{Q}_h^k(s_h,\cdot),\pi_h^{\star,k}(\cdot|s_h)-\pi_h^{\star,(l-1)\tau+1}(\cdot|s_h)\rangle\right]}_{(II)}$$

**I) Bound the term (I)**
By Lemma A.4, we have

$$(I)\le\frac{1}{2}\eta KH+\sum_{h\in[H]}\frac{1}{\eta}\sum_{l\in[L]}\mathbb{E}_{P^\star,(l-1)\tau+1,\pi^\star,(l-1)\tau+1}$$

$$\times\left[\sum_{k=(l-1)\tau+1}^{l\tau}\left[D_{KL}\left(\pi^{\star,(l-1)\tau+1}(\cdot|s_h)\|\pi^k(\cdot|s_h)\right)-D_{KL}\left(\pi^{\star,(l-1)\tau+1}(\cdot|s_h)\|\pi^{k+1}(\cdot|s_h)\right)\right]\right]$$

$$\le\frac{1}{2}\eta KH+\sum_{h\in[H]}\frac{1}{\eta}\sum_{l\in[L]}\mathbb{E}_{P^\star,(l-1)\tau+1,\pi^\star,(l-1)\tau+1}$$

$$\times\left[D_{KL}\left(\pi^{\star,(l-1)\tau+1}(\cdot|s_h)\|\pi^{(l-1)\tau+1}(\cdot|s_h)\right)-D_{KL}\left(\pi^{\star,(l-1)\tau+1}(\cdot|s_h)\|\pi^{l\tau+1}(\cdot|s_h)\right)\right]$$

$$\le\frac{1}{2}\eta KH+\sum_{h\in[H]}\frac{1}{\eta}\sum_{l\in[L]}\mathbb{E}_{P^\star,(l-1)\tau+1,\pi^\star,(l-1)\tau+1}\left[D_{KL}\left(\pi^{\star,(l-1)\tau+1}(\cdot|s_h)\|\pi^{(l-1)\tau+1}(\cdot|s_h)\right)\right]$$

$$\le\frac{1}{2}\eta KH+\frac{1}{\eta}LH\log A,$$

where the last equation follows because

$$D_{KL}\left(\pi^{\star,(l-1)\tau+1}(\cdot|s_h)\|\pi^{(l-1)\tau+1}(\cdot|s_h)\right)=\sum_{a\in\mathcal{A}}\pi^{\star,(l-1)\tau+1}(a|s_h)\log(A\cdot\pi^{(l-1)\tau+1}(\cdot|s_h))$$

$$=\log A+\sum_a\pi^{\star,(l-1)\tau+1}(a_h|s_h)\cdot\log\pi^{\star,(l-1)\tau+1}(a_h|s_h)$$

$$\le\log A.$$

**II) Bound the term (II)**

$$(II)\le\sum_{l\in[L]}\sum_{k=(l-1)\tau+1}^{l\tau}\sum_{h\in[H]}\mathbb{E}_{P^\star,(l-1)\tau+1,\pi^\star,(l-1)\tau+1}\left[\left\|\pi_h^{\star,k}(\cdot|s_h)-\pi_h^{\star,(l-1)\tau+1}(\cdot|s_h)\right\|_{TV}\right]$$

$$\le\sum_{l\in[L]}\sum_{k=(l-1)\tau+1}^{l\tau}\sum_{h\in[H]}\sum_{t=(l-1)\tau+2}^{k}\mathbb{E}_{P^\star,(l-1)\tau+1,\pi^\star,(l-1)\tau+1}\left[\left\|\pi_h^{\star,t}(\cdot|s_h)-\pi_h^{\star,t-1}(\cdot|s_h)\right\|_{TV}\right]$$

$$\le\sum_{l\in[L]}\sum_{k=(l-1)\tau+1}^{l\tau}\sum_{t=(l-1)\tau+2}^{k}\sum_{h\in[H]}\max_{s\in\mathcal{S}}\left[\left\|\pi_h^{\star,t}(\cdot|s)-\pi_h^{\star,t-1}(\cdot|s)\right\|_{TV}\right]$$

$$\le\sum_{l\in[L]}\sum_{k=(l-1)\tau+1}^{l\tau}\sum_{t=(l-1)\tau+1}^{l\tau}\sum_{h\in[H]}\max_{s\in\mathcal{S}}\left[\left\|\pi_h^{\star,t}(\cdot|s)-\pi_h^{\star,t-1}(\cdot|s)\right\|_{TV}\right]$$

$$\leq \tau \sum_{k\in[K]} \sum_{h\in[H]} \max_{s\in\mathcal{S}} \left[ \left\| \pi_h^{\star,k}(\cdot|s) - \pi_h^{\star,k-1}(\cdot|s) \right\|_{TV} \right]$$

$$\leq \tau \Delta_{[H],[K]}^{\pi}.$$

$\square$

### A.3.2 Bound $(b)$ in Equation (10)

**Lemma A.6.** *The term $(b)$ in Equation* (10) *can be bounded as follows:*

$$\sum_{l\in[L]} \sum_{k=(l-1)\tau+1}^{l\tau} \sum_{h\in[H]} \left( \mathbb{E}_{P^{\star,k},\pi^{\star,k}} - \mathbb{E}_{P^{\star,(l-1)\tau+1},\pi^{\star,(l-1)\tau+1}} \right) \left[ \langle \hat{Q}_h^k(s_h,\cdot), \pi_h^{\star,k}(\cdot|s_h) - \pi_h^k(\cdot|s_h) \rangle \right]$$

$$\leq 2H\tau(\Delta_{[H],[K]}^{P} + \Delta_{[H],[K]}^{\pi}).$$

*Proof.* Denote the indicator function of state $s_h$ as $\mathbb{I}(s_h)$, and then we have

$$\sum_{l\in[L]} \sum_{k=(l-1)\tau+1}^{l\tau} \sum_{h\in[H]} \left( \mathbb{E}_{P^{\star,k},\pi^{\star,k}} - \mathbb{E}_{P^{\star,(l-1)\tau+1},\pi^{\star,(l-1)\tau+1}} \right) \left[ \langle \hat{Q}_h^k(s_h,\cdot), \pi_h^{\star,k}(\cdot|s_h) - \pi_h^k(\cdot|s_h) \rangle \right]$$

$$\leq \sum_{l\in[L]} \sum_{k=(l-1)\tau+1}^{l\tau} \sum_{h\in[H]} \int_{s_h} \left| \mathbb{P}_{P^{\star,k}}^{\pi^{\star,k}}(s_h) - \mathbb{P}_{P^{\star,(l-1)\tau+1}}^{\pi^{\star,(l-1)\tau+1}}(s_h) \right| ds_h$$

$$\leq \sum_{l\in[L]} \sum_{k=(l-1)\tau+1}^{l\tau} \sum_{h\in[H]} \sum_{t=(l-1)\tau+2}^{k} \int_{s_h} \left| \mathbb{P}_{P^{\star,t}}^{\pi^{\star,t}}(s_h) - \mathbb{P}_{P^{\star,t-1}}^{\pi^{\star,t-1}}(s_h) \right| ds_h, \tag{11}$$

where $\mathbb{P}_P^{\pi}(s)$ denotes the visitation probability at state $s$ under model $P$ and policy $\pi$.

Consider $\int_{s_h} \left| \mathbb{P}_{P^{\star,t}}^{\pi^{\star,t}}(s_h) - \mathbb{P}_{P^{\star,t-1}}^{\pi^{\star,t-1}}(s_h) \right| ds_h$ can be further decomposed as

$$\int_{s_h} \left| \mathbb{P}_{P^{\star,t}}^{\pi^{\star,t}}(s_h) - \mathbb{P}_{P^{\star,t-1}}^{\pi^{\star,t-1}}(s_h) \right| ds_h$$

$$\leq \int_{s_h} \left| \mathbb{P}_{P^{\star,t}}^{\pi^{\star,t-1}}(s_h) - \mathbb{P}_{P^{\star,t-1}}^{\pi^{\star,t-1}}(s_h) \right| ds_h + \int_{s_h} \left| \mathbb{P}_{P^{\star,t}}^{\pi^{\star,t}}(s_h) - \mathbb{P}_{P^{\star,t}}^{\pi^{\star,t-1}}(s_h) \right| ds_h.$$

For the first term $\int_{s_h} \left| \mathbb{P}_{P^{\star,t}}^{\pi^{\star,t-1}}(s_h) - \mathbb{P}_{P^{\star,t-1}}^{\pi^{\star,t-1}}(s_h) \right| ds_h$,

$$\int_{s_h} \left| \mathbb{P}_{P^{\star,t}}^{\pi^{\star,t-1}}(s_h) - \mathbb{P}_{P^{\star,t-1}}^{\pi^{\star,t-1}}(s_h) \right| ds_h$$

$$\leq \int_{s_h} \sum_{i=1}^{h} \left| (P_1^{\star,t})^{\pi_1^{\star,t-1}} \cdots (P_i^{\star,t})^{\pi_i^{\star,t-1}} \cdots (P_{h-1}^{\star,t-1})^{\pi_{h-1}^{\star,t-1}}(s_h) \right.$$

$$\left. - (P_1^{\star,t})^{\pi_1^{\star,t-1}} \cdots (P_i^{\star,t-1})^{\pi_i^{\star,t-1}} \cdots (P_{h-1}^{\star,t-1})^{\pi_{h-1}^{\star,t-1}}(s_h) \right| ds_h$$

$$\leq \int_{s_h} \sum_{i=1}^{h} \left| \int_{s_2,\dots,s_{h-1}} \left| (P_i^{\star,t})^{\pi_i^{\star,t-1}}(s_{i+1}|s_i) - (P_i^{\star,t-1})^{\pi_i^{\star,t-1}}(s_{i+1}|s_i) \right| \right.$$

$$\left. \prod_{j=1}^{i-1} (P_j^{\star,t})^{\pi_j^{\star,t-1}}(s_{j+1}|s_j) \prod_{j=i+1}^{h-1} (P_j^{\star,t-1})^{\pi_j^{\star,t-1}}(s_{j+1}|s_j) ds_2 \dots ds_{h-1} \right| ds_h$$

$$\overset{(i)}{\leq} \int_{s_h} \sum_{i=1}^{h} \left| \int_{s_2,\dots,s_i,s_{i+2},\dots s_{h-1}} \int_{s_{i+1}} \left| (P_i^{\star,t})^{\pi_i^{\star,t-1}}(s_{i+1}|s_i) - (P_i^{\star,t-1})^{\pi_i^{\star,t-1}}(s_{i+1}|s_i) \right| \right.$$

$$\max_{s_{i+1}\in\mathcal{S}}\prod_{j=1}^{i-1}(P_j^{\star,t})^{\pi_j^{\star,t-1}}(s_{j+1}|s_j)\prod_{j=i+1}^{h-1}(P_j^{\star,t-1})^{\pi_j^{\star,t-1}}(s_{j+1}|s_j)ds_2\ldots ds_{h-1}\bigg|ds_h$$

$$\overset{(ii)}{\leq}\int_{s_h}\sum_{i=1}^{h}\bigg|\int_{s_2,\ldots,s_{i-1},s_{i+2},\ldots s_{h-1}}\max_{s_i\in\mathcal{S}}\int_{s_{i+1}}\Big|(P_i^{\star,t})^{\pi_i^{\star,t-1}}(s_{i+1}|s_i)-(P_i^{\star,t-1})^{\pi_i^{\star,t-1}}(s_{i+1}|s_i)\Big|$$

$$\int_{s_i}\max_{s_{i+1}\in\mathcal{S}}\prod_{j=1}^{i-1}(P_j^{\star,t})^{\pi_j^{\star,t-1}}(s_{j+1}|s_j)\prod_{j=i+1}^{h-1}(P_j^{\star,t-1})^{\pi_j^{\star,t-1}}(s_{j+1}|s_j)ds_2\ldots ds_{h-1}\bigg|ds_h$$

$$\overset{(iii)}{\leq}\int_{s_h}\sum_{i=1}^{h}\bigg|\max_{(s,a)\in\mathcal{S}\times\mathcal{A}}\Big\|P_i^{\star,t}(\cdot|s,a)-P_i^{\star,t-1}(\cdot|s,a)\Big\|_{TV}\underbrace{\int_{s_1,\ldots,s_i}\prod_{j=1}^{i-1}(P_j^{\star,t})^{\pi_j^{\star,t-1}}(s_{j+1}|s_j)ds_1\ldots ds_i}_{=1}$$

$$\underbrace{\int_{s_{i+2},\ldots,s_{h-1}}\max_{s_{i+1}\in\mathcal{S}}\prod_{j=i+1}^{h-1}(P_j^{\star,t-1})^{\pi_j^{\star,t-1}}(s_{j+1}|s_j)ds_{i+2}\ldots ds_{h-1}}_{\leq 1}\bigg|ds_h$$

$$\leq\sum_{h\in[H]}\max_{(s,a)\in\mathcal{S}\times\mathcal{A}}\Big\|P_i^{\star,t}(\cdot|s,a)-P_i^{\star,t-1}(\cdot|s,a)\Big\|_{TV}, \tag{12}$$

where $(i)$ and $(ii)$ follow from Holder's inequality, and $(iii)$ follows from the definition of total variation distance.

Similarly for the second term and from Lemma A.19

$$\int_{s_h}\Big|\mathbb{P}_{P^{\star,t}}^{\pi^{\star,t}}(s_h)-\mathbb{P}_{P^{\star,t}}^{\pi^{\star,t-1}}(s_h)\Big|ds_h\leq\sum_{h\in[H]}\max_{s\in\mathcal{S}}\Big\|\pi_h^{\star,t}(\cdot|s)-\pi_h^{\star,t-1}(\cdot|s)\Big\|_{TV}. \tag{13}$$

Plug Equation (12) and Equation (13) into Equation (11), we have

$$\sum_{l\in[L]}\sum_{k=(l-1)\tau+1}^{l\tau}\sum_{h\in[H]}\left(\mathbb{E}_{P^{\star,k},\pi^{\star,k}}-\mathbb{E}_{P^{\star,(l-1)\tau+1},\pi^{\star,(l-1)\tau+1}}\right)\left[\langle\hat{Q}_h^k(s_h,\cdot),\pi_h^{\star,k}(\cdot|s_h)-\pi_h^k(\cdot|s_h)\rangle\right]$$

$$\leq 2\sum_{l\in[L]}\sum_{k=(l-1)\tau+1}^{l\tau}\sum_{h\in[H]}\sum_{t=(l-1)\tau+2}^{k}\left(\sum_{i\in[H]}\max_{s\in\mathcal{S}}\Big\|\pi_i^{\star,t}(\cdot|s)-\pi_i^{\star,t-1}(\cdot|s)\Big\|_{TV}\right.$$

$$\left.+\sum_{i\in[H]}\max_{(s,a)\in\mathcal{S}\times\mathcal{A}}\Big\|P_i^{\star,t}(\cdot|s,a)-P_i^{\star,t-1}(\cdot|s,a)\Big\|_{TV}\right)$$

$$\leq 2\sum_{h\in[H]}\left(\sum_{l\in[L]}\sum_{k=(l-1)\tau+1}^{l\tau}\sum_{t=(l-1)\tau+1}^{l\tau}\sum_{i\in[H]}\left(\max_{s\in\mathcal{S}}\Big\|\pi_i^{\star,t}(\cdot|s)-\pi_i^{\star,t-1}(\cdot|s)\Big\|_{TV}\right.\right.$$

$$\left.\left.+\max_{(s,a)\in\mathcal{S}\times\mathcal{A}}\Big\|P_i^{\star,t}(\cdot|s,a)-P_i^{\star,t-1}(\cdot|s,a)\Big\|_{TV}\right)\right)$$

$$=2H\tau(\Delta_{[H],[K]}^P+\Delta_{[H],[K]}^\pi).$$

$\square$

### A.3.3 Combining $(a)$ and $(b)$ Together

**Lemma A.7.** *Let $\eta = \sqrt{\frac{L\log A}{K}}$, and we have*

$$\sum_{l\in[L]}\sum_{k=(l-1)\tau+1}^{l\tau}\sum_{h\in[H]}\mathbb{E}_{\pi^{\star,k}}\left[\langle\hat{Q}_h^{\pi^k,k}(s_h,\cdot),\pi_h^{\star,k}(\cdot|s_h)-\pi_h^k(\cdot|s_h)\rangle\right]$$
$$\leq 2HK\sqrt{\log A/\tau}+3H\tau(\Delta_{[H],[K]}^P+\Delta_{[H],[K]}^\pi).$$

*Proof.* We derive the following bound:

$$\sum_{k=1}^K\sum_{h\in H}\mathbb{E}_{s_h\sim(P^{\star,k},\pi^{\star,k})}\left[\langle Q_{h,\hat{P}^k,r^k}^{\pi^k}(s_h,\cdot),\pi_h^{\star,k}(\cdot|s_h)-\pi_h^k(\cdot|s_h)\rangle\right]$$
$$\overset{(i)}{\leq}\frac{1}{2}\eta KH+\frac{1}{\eta}LH\log A+\tau\Delta_{[H],[K]}^\pi+2H\tau(\Delta_{[H],[K]}^P+\Delta_{[H],[K]}^\pi)$$
$$\overset{(ii)}{\leq}2H\sqrt{KL\log A}+3H\tau(\Delta_{[H],[K]}^P+\Delta_{[H],[K]}^\pi)$$
$$=2HK\sqrt{\log A/\tau}+3H\tau(\Delta_{[H],[K]}^P+\Delta_{[H],[K]}^\pi),$$

where $(i)$ follows from Lemmas A.5 and A.6, and $(ii)$ follows from the choice of $\eta=\sqrt{\frac{L\log A}{K}}$. $\square$

### A.4 Proof Theorem 4.4

*Proof of Theorem 4.4.* Combine Lemmas A.1 to A.3 and A.7, we have

$$\text{Gap}_{\text{Ave}}(K)=\frac{1}{K}\sum_{k=1}^K V_{P^{\star,k},r^k}^\star-V_{P^{\star,k},r^k}^{\pi^k}$$
$$\leq O\Big(\tfrac{H}{K}\Big[\sqrt{KdA(A\log(|\Phi||\Psi|KH/\delta)+d^2)}\Big[H\sqrt{\tfrac{Kd}{W}\log(W)}+\sqrt{HW^2\Delta_{[H],[K]}^\phi}\Big]+\sqrt{W^3AC_B}\Delta_{[H],[K]}^{\sqrt{P}}\Big]\Big)$$
$$+O\left(\frac{2H}{\sqrt{\tau}}+\frac{3H\tau}{K}(\Delta_{[H],[K]}^P+\Delta_{[H],[K]}^\pi)\right)$$
$$=\tilde{O}\underbrace{\left(\sqrt{\frac{H^4d^2A}{W}}\,(A+d^2)+\sqrt{\frac{H^3dA}{K}}\,(A+d^2)\,W^2\Delta_{[H],[K]}^\phi+\sqrt{\frac{H^2W^3A}{K^2}}\Delta_{[H],[K]}^{\sqrt{P}}\right)}_{(I)}$$
$$+\tilde{O}\underbrace{\left(\frac{2H}{\sqrt{\tau}}+\frac{3H\tau}{K}(\Delta_{[H],[K]}^P+\Delta_{[H],[K]}^\pi)\right)}_{(II)}. \tag{14}$$

$\square$

### A.5 Supporting Lemmas

We first provide the following concentration results, which is an extension of Lemma 39 in Zanette et al. (2021).

**Lemma A.8.** *There exists a constant $\lambda_{k,W}=O(d\log(|\Phi|\min\{k,W\}TH/\delta))$, the following inequality holds for any $k\in[K], h\in[H], s_h\in\mathcal{S}, a_h\in\mathcal{A}$ and $\phi\in\Phi$ with probability at least $1-\delta$:*

$$\frac{1}{5}\left\|\phi_h(s,a)\right\|_{(U_{h,\phi}^{k,W})^{-1}}\leq\left\|\phi_h(s,a)\right\|_{(\hat{U}_h^{k,W})^{-1}}\leq 3\left\|\phi_h(s,a)\right\|_{(U_{h,\phi}^{k,W})^{-1}}$$

The following result follows directly from Lemma A.8.

**Corollary A.9.** *The following inequality holds for any $k \in [K], h \in [H], s_h \in \mathcal{S}, a_h \in \mathcal{A}$ with probability at least $1 - \delta$:*

$$\min\left\{ \frac{\tilde{\alpha}_{k,W}}{5} \left\| \hat{\phi}_h^k(s_h, a_h) \right\|_{(U_{h,\hat{\phi}^k}^{k,W})^{-1}}, 1 \right\} \leq \hat{b}_h^k(s_h, a_h) \leq 3\tilde{\alpha}_{k,W} \left\| \hat{\phi}_h^k(s_h, a_h) \right\|_{(U_{h,\hat{\phi}^k}^{k,W})^{-1}},$$

*where $\tilde{\alpha}_{k,W} = 5\sqrt{2WA\zeta_{k,W} + \lambda_{k,W}d}$.*

We next provide the MLE guarantee for nonstationary RL, which shows that under any exploration policy, the estimation error can be bounded with high probability. Differently from Theorem 21 in Agarwal et al. (2020b), we capture the nonstationarity in the analysis.

**Lemma A.10** (Nonstationary MLE Guarantee). *Given $\delta \in (0, 1)$, under Assumptions 4.1 to 4.3, let $C_B = \sqrt{\frac{C_B}{p_{\min}}}$, and consider the transition kernels learned from line 3 in Algorithm 2. We have the following inequality holds for any $n, h \geq 2$ with probability at least $1 - \delta/2$:*

$$\frac{1}{W} \sum_{i=1\vee(k-W)}^{k-1} \mathop{\mathbb{E}}_{\substack{s_{h-1}\sim(P^{\star,i},\pi^i) \\ a_{h-1},a_h\sim\mathcal{U}(\mathcal{A}) \\ s_h\sim P^{\star,i}(\cdot|s_{h-1},a_{h-1})}} \left[ f_h^k(s_h, a_h)^2 \right] \leq \zeta_{k,W} + 2C_B\Delta_{h,[k-W,k-1]}^P, \tag{15}$$

*where, $\zeta_{k,W} := \frac{2\log(2|\Phi||\Psi|kH/\delta)}{W}$. In addition, for $h = 1$,*

$$\mathop{\mathbb{E}}_{a_1\sim\mathcal{U}(\mathcal{A})} \left[ f_1^k(s_1, a_1)^2 \right] \leq \zeta_{k,W} + 2C_B\Delta_{1,[k-W,k-1]}^P.$$

*Proof of Lemma A.10.* For simplification, we denote $x = (s, a) \in \mathcal{X}, \mathcal{X} = \mathcal{S} \times \mathcal{A}, y = p^\star \in \mathcal{Y}, \mathcal{Y} = \mathcal{S}$. The model estimation process in Algorithm 1 can be viewed as a sequential conditional probability estimation setting with an instance space $\mathcal{X}$ and a target space $\mathcal{Y}$, where the conditional density is given by $p^i(y|x) = P^{\star,i}(y|x)$ for any $i$. We are given a dataset $D := \{(x_i, y_i)\}_{i=1\vee(k-W)}^k$, where $x_i \sim \mathcal{D}_i = \mathcal{D}_i(x_{1:i-1}, y_{1:i-1})$ and $y_i \sim p^i(\cdot|x_i)$. Let $D'$ denote a tangent sequence $\{(x_i', y_i')\}_{i=1\vee(k-W)}^k$ where $x_i' \sim \mathcal{D}_i(x_{1:i-1}, y_{1:i-1})$ and $y_i' \sim p^i(\cdot|x_i')$. Further, we consider a function class $\mathcal{F} = \Phi \times \Psi : (\mathcal{X} \times \mathcal{Y}) \to \mathbb{R}$ and assume that the reachability condition $P^{\star,i} \in \mathcal{F}$ holds for any $i$.

We first introduce one useful lemma in Agarwal et al. (2020b) to decouple data.

**Lemma A.11** (Lemma 24 of Agarwal et al. (2020b)). *Let $D$ ba a dataset with at most $W$ samples and $D'$ be the corresponding tangent sequence. Let $L(P, D) = \sum_{i=1\vee(k-W)}^k l(P, (x_i, y_i))$ be any function that decomposes additively across examples where $l$ is any function. Let $\widehat{P}(D)$ be any estimator taking as input random variable $D$ and with range $\mathcal{F}$. Then*

$$\mathbb{E}_D \left[ \exp\left( L(\widehat{P}(D), D) - \log \mathbb{E}_{D'}\left[ \exp(L(\widehat{P}(D), D')) \right] - \log|\mathcal{F}| \right) \right] \leq 1.$$

Suppose $\widehat{f}(D)$ is learned from the following maximum likelihood problem:

$$\widehat{P}(D) := \arg\max_{P\in\mathcal{F}} \sum_{(x_i,y_i)\in D} \log f(x_i, y_i). \tag{16}$$

Combining the Chernoff bound and Lemma A.11, we obtain an exponential tail bound, i.e., with probability at least $1 - \delta$,

$$-\log \mathbb{E}_{D'}\left[ \exp(L(\widehat{P}(D), D')) \right] \leq -L(\widehat{P}(D), D) + \log|\mathcal{F}| + \log(1/\delta). \tag{17}$$

To proceed, we let $L(P, D) = \sum_{i=1\vee(k-W)}^{k-1} -\frac{1}{2}\log(P^{\star,k}(x_i, y_i)/P(x_i, y_i))$ where $D$ is a dataset $\{(x_i, y_i)\}_{i=1\vee(k-W)}^k$(and $D' = \{(x_i', y_i')\}_{i=1\vee(k-W)}^k$ is tangent sequence). Then $L(P^{\star,k}, D) = 0 \leq L(\hat{P}, D)$.

Then, the RHS of Equation (17) can be bounded as

$$\text{RHS of Equation (17)} = \sum_{i=1\vee(k-W)}^{k} \frac{1}{2}\log(P^{\star,k}(x_i,y_i)/\widehat{P}(x_i,y_i)) + \log|\mathcal{F}| + \log(1/\delta)$$

$$\leq \log|\mathcal{F}| + \log(1/\delta) = \log\left(|\Phi||\Psi|/\delta\right), \tag{18}$$

where the inequality follows because $\widehat{P}$ is MLE and from the assumption of reachability, and the last equality follows because $|\mathcal{F}| = |\Phi||\Psi|$.

Consider for any distribution $p$ and $q$ over a domain $\mathcal{Z}$. Then the Hellinger distance $H^2(p\|q) = \int\left(\sqrt{p(z)} - \sqrt{q(z)}\right)^2 dz$ satisfies that

$$H^2(p\|q)$$
$$= \int\left(\sqrt{p(z)} - \sqrt{q(z)}\right)^2 dz = \int p(z) + q(z) - 2\sqrt{p(z)}\sqrt{q(z)}dz$$
$$= 2\left(\int 1 - \sqrt{p(z)}\sqrt{q(z)}dz\right) = 2\left(\int 1 - \sqrt{p(z)}\sqrt{q(z)}dz\right) = 2\mathbb{E}_{z\sim q}\left[1 - \sqrt{p(z)/q(z)}\right]. \tag{19}$$

Next, the LHS of Equation (17) can be bounded as

LHS of Equation (17)

$$\overset{(i)}{=} -\log\mathbb{E}_{D'}\left[\exp\left(\sum_{i=1\vee(k-W)}^{k-1} -\frac{1}{2}\log\left(\frac{P^{\star,k}(x_i',y_i')}{\widehat{P}(x_i',y_i')}\right)\right)\bigg| D\right]$$

$$= \sum_{i=1\vee(k-W)}^{k-1} -\log\mathbb{E}_D\left[\exp\left(-\frac{1}{2}\log\left(\frac{P^{\star,k}(x_i,y_i)}{\widehat{P}(x_i,y_i)}\right)\right)\right]$$

$$= \sum_{i=1\vee(k-W)}^{k-1} -\log\mathbb{E}_D\left[\sqrt{\frac{\widehat{P}(x_i,y_i)}{P^{\star,k}(x_i,y_i)}}\right]$$

$$\overset{(ii)}{\geq} \sum_{i=1\vee(k-W)}^{k-1}\left\{1 - \mathbb{E}_D\left[\sqrt{\frac{\widehat{P}(x_i,y_i)}{P^{\star,k}(x_i,y_i)}}\right]\right\}$$

$$= \sum_{i=1\vee(k-W)}^{k-1}\left\{\mathbb{E}_{x_i\sim D}\left[1 - \mathbb{E}_{y_i\sim P^{\star,i}(\cdot|x_i)}\left[\sqrt{\frac{\widehat{P}(x_i,y_i)}{P^{\star,k}(x_i,y_i)}}\right]\right]\right\}$$

$$= \sum_{i=1\vee(k-W)}^{k-1}\left\{\mathbb{E}_{x_i\sim D}\left[1 - \mathbb{E}_{y_i\sim P^{\star,k}(\cdot|x_i)}\left[\sqrt{\frac{\widehat{P}(x_i,y_i)}{P^{\star,k}(x_i,y_i)}}\right]\right]\right\}$$

$$+ \sum_{i=1\vee(k-W)}^{k-1}\mathbb{E}_{x_i\sim D}\left[\mathbb{E}_{y_i\sim P^{\star,k}(\cdot|x_i)}\left[\sqrt{\frac{\widehat{P}(x_i,y_i)}{P^{\star,k}(x_i,y_i)}}\right] - \mathbb{E}_{y_i\sim P^{\star,i}(\cdot|x_i)}\left[\sqrt{\frac{\widehat{P}(x_i,y_i)}{P^{\star,k}(x_i,y_i)}}\right]\right]$$

$$\overset{(iii)}{\geq} -WC_B\Delta_{h,[k-W,k-1]}^P + \sum_{i=1\vee(k-W)}^{k-1}\mathbb{E}_{x_i'\sim\mathcal{D}_i}\left[H^2\left(P^{\star,k}(\cdot|x_i')\|\hat{P}(\cdot|x_i')\right)\right]$$

$$\overset{(iv)}{\geq} -WC_B\Delta_{h,[k-W,k-1]}^P + \frac{1}{2}\sum_{i=1\vee(k-W)}^{k-1}\mathbb{E}_{x_i\sim\mathcal{D}_i}\left[\left\|P^{\star,k}(x_i,\cdot) - \widehat{P}(x_i,\cdot)\right\|_{TV}^2\right], \tag{20}$$

where $(i)$ follows from the above definition of $L(f,D)$, $(ii)$ follows from the fact that $1 - x \leq -\log(x)$, $(iii)$ follows from the definition of variation budgets and Equation (19), and $(iv)$ follows from the fact that $\|p(\cdot) - q(\cdot)\|_{TV}^2 \leq H^2(p\|q)$ as indicated in Lemma 2.3 in Tsybakov (2009).

Combining Equations (17), (18) and (20), we have

$$\frac{1}{W} \sum_{i=1\vee(k-W)}^{k-1} \mathbb{E}_{\substack{s_{h-1}\sim(P^{\star,i},\pi^i)\\a_{h-1},a_h\sim\mathcal{U}(\mathcal{A})\\s_h\sim P^{\star,i}(\cdot|s_{h-1},a_{h-1})}} \left[f_h^k(s_h,a_h)^2\right] \leq 2C_B\Delta_{h,[k-W,k-1]}^P + \frac{2\log\left(|\Phi||\Psi|/\delta\right)}{W}. \quad (21)$$

We substitute $\delta$ with $\delta/2nH$ to ensure Equation (21) holds for any $h \in [H]$ and $n$ with probability at least $1 - \delta/2$, which finishes the proof. $\qquad\square$

The following lemma extends Lemmas 12 and 13 under infinite discount MDPs under stationary case in Uehara et al. (2022) to episodic MDPs under nonstationary environment, which captures nonstationarity in analysis.

**Lemma A.12** (Nonstationary Step Back). *Let $\mathcal{I} = [1 \vee (k-W), k-1]$ be an index set and $\{P_{h-1}^i\}_{i\in\mathcal{I}} = \{\langle\phi_{h-1}^i, \mu_{h-1}^i\rangle\}$ be a set of generic MDP model, and $\Pi$ be an arbitrary and possibly mixture policy. Define an expected Gram matrix as follows. For any $\phi \in \Phi$,*

$$M_{h-1,\phi} = \lambda_W I + \sum_{i\in\mathcal{I}} \mathbb{E}_{\substack{s_{h-1}\sim(P^{i,\star},\Pi)\\a_{h-1}\sim\Pi}} \left[\phi_{h-1}(s_{h-1},a_{h-1})\left(\phi_{h-1}(s_{h-1},a_{h-1})\right)^\top\right].$$

*Further, let $f_{h-1}^i(s_{h-1}, a_{h-1})$ be the total variation between $P_{h-1}^{i,\star}$ and $P_{h-1}^i$ at time step $h-1$. Suppose $g \in \mathcal{S} \times \mathcal{A} \to \mathbb{R}$ is bounded by $B \in (0, \infty)$, i.e., $\|g\|_\infty \leq B$. Then, $\forall h \geq 2, \forall$ policy $\pi_h$,*

$$\mathbb{E}_{\substack{s_h\sim P_{h-1}^k\\a_h\sim\pi_h}} [g(s_h,a_h)|s_{h-1},a_{h-1}]$$

$$\leq \left\|\phi_{h-1}^k(s_{h-1},a_{h-1})\right\|_{(M_{h-1,\phi^k})^{-1}} \times$$

$$\sqrt{\sum_{i\in\mathcal{I}} A \mathbb{E}_{\substack{s_h\sim(P^{i,\star},\Pi)\\a_h\sim\mathcal{U}}} [g(s_h,a_h)^2] + WB^2\Delta_{h-1,\mathcal{I}}^P + \lambda_{k,W}dB^2 + \sum_{i\in\mathcal{I}} AB^2 \mathbb{E}_{\substack{s_{h-1}\sim(P^{i,\star},\Pi)\\a_{h-1}\sim\Pi}} \left[f_{h-1}^k(s_{h-1},a_{h-1})^2\right]}.$$

*Proof.* We first derive the following bound:

$$\mathbb{E}_{\substack{s_h\sim P_{h-1}^k\\a_h\sim\pi_h}} [g(s_h,a_h)|s_{h-1},a_{h-1}]$$

$$= \int_{s_h}\sum_{a_h} g(s_h,a_h)\pi(a_h|s_h)\langle\phi_{h-1}^k(s_{h-1},a_{h-1}),\mu_{h-1}^k(s_h)\rangle ds_h$$

$$\leq \left\|\phi_{h-1}^k(s_{h-1},a_{h-1})\right\|_{(M_{h-1,\phi^k})^{-1}} \left\|\int\sum_{a_h} g(s_h,a_h)\pi(a_h|s_h)\mu_{h-1}^k(s_h)ds_h\right\|_{M_{h-1,\phi^k}},$$

where the inequality follows from Cauchy-Schwarz inequality. We further expand the second term in the RHS of the above inequality as follows.

$$\left\|\int\sum_{a_h} g(s_h,a_h)\pi(a_h|s_h)\mu_{h-1}^k(s_h)ds_h\right\|_{M_{h-1,\phi^k}}^2$$

$$\overset{(i)}{\leq} \sum_{i\in\mathcal{I}} \mathbb{E}_{\substack{s_{h-1}\sim(P^{i,\star},\Pi)\\a_{h-1}\sim\Pi}} \left[\left(\int_{s_h}\sum_{a_h} g(s_h,a_h)\pi_h(a_h|s_h)\mu^k(s_h)^\top\phi^k(s_{h-1},a_{h-1})ds_h\right)^2\right] + \lambda_{k,W}dB^2$$

$$= \sum_{i\in\mathcal{I}} \mathbb{E}_{\substack{s_{h-1}\sim(P^{i,\star},\Pi)\\a_{h-1}\sim\Pi}} \left[\left(\mathbb{E}_{\substack{s_h\sim P_{h-1}^k\\a_h\sim\pi_h}} \left[g(s_h,a_h)\Big|s_{h-1},a_{h-1}\right]\right)^2\right] + \lambda_{k,W}dB^2$$

$$\overset{(ii)}{\leq} \sum_{i\in\mathcal{I}} \mathbb{E}_{\substack{s_{h-1}\sim(P^{i,\star},\Pi)\\a_{h-1}\sim\Pi}} \left[\mathbb{E}_{\substack{s_h\sim P_{h-1}^k\\a_h\sim\pi_h}} \left[g(s_h,a_h)^2\Big|s_{h-1},a_{h-1}\right]\right] + \lambda_{k,W}dB^2$$

$$\overset{(iii)}{\le} \sum_{i\in\mathcal{I}} \mathop{\mathbb{E}}_{\substack{s_{h-1}\sim(P^{i,\star},\Pi)\\a_{h-1}\sim\Pi}} \left[ \mathop{\mathbb{E}}_{\substack{s_h\sim P^{i,\star}_{h-1}\\a_h\sim\pi_h}} \left[ g(s_h,a_h)^2 \Big| s_{h-1},a_{h-1} \right] \right] + \lambda_{k,W} dB^2$$

$$+ \sum_{i\in\mathcal{I}} B^2 \mathop{\mathbb{E}}_{\substack{s_{h-1}\sim(P^{i,\star},\Pi)\\a_{h-1}\sim\Pi}} \left[ \left\| P^{i,\star}_{h-1}(s_{h-1},a_{h-1}) - P^{k,\star}_{h-1}(s_{h-1},a_{h-1}) \right\|_{TV} \right]$$

$$+ \sum_{i\in\mathcal{I}} B^2 \mathop{\mathbb{E}}_{\substack{s_{h-1}\sim(P^{i,\star},\Pi)\\a_{h-1}\sim\Pi}} \left[ f^k_{h-1}(s_{h-1},a_{h-1})^2 \right]$$

$$\overset{(iv)}{\le} \sum_{i\in\mathcal{I}} A \mathop{\mathbb{E}}_{\substack{s_h\sim(P^{i,\star},\Pi)\\a_h\sim\mathcal{U}}} \left[ g(s_h,a_h)^2 \right] + \lambda_{k,W} dB^2 + WB^2 \Delta^P_{h-1,\mathcal{I}} + \sum_{i\in\mathcal{I}} B^2 \mathop{\mathbb{E}}_{\substack{s_{h-1}\sim(P^{i,\star},\Pi)\\a_{h-1}\sim\Pi}} \left[ f^k_{h-1}(s_{h-1},a_{h-1})^2 \right],$$

where $(i)$ follows from the assumption that $\|g\|_\infty \le B$, $(ii)$ follows from Jensen's inequality, $(iii)$ follows because $f^k_{h-1}(s_{h-1},a_{h-1})$ is the total variation between $P^{k,\star}_{h-1}$ and $P^k_{h-1}$ at time step $h-1$, and $(iv)$ follows from importance sampling and the definition of $\Delta^P_{h-1,\mathcal{I}}$. This finishes the proof. $\square$

Recall that $f^k_h(s,a) = \|\hat{P}^k_h(\cdot|s,a) - P^{\star,k}_h(\cdot|s,a)\|_{TV}$. Using Lemma A.12, we have the following lemma to bound the expectation of $f^k_h(s,a)$ under estimated transition kernels.

**Lemma A.13.** *Denote $\alpha_{k,W} = \sqrt{2WA\zeta_{k,W} + \lambda_{k,W}d}$. For any $k \in [K]$, policy $\pi$ and reward $r$, for all $h \ge 2$, we have*

$$\mathbb{E}_{(s_h,a_h)\sim(\hat{P}^k,\pi)} \left[ f^k_h(s_h,a_h) \Big| s_{h-1},a_{h-1} \right]$$

$$\le \min\left\{ \alpha_{k,W} \left\| \hat{\phi}^k_{h-1}(s_{h-1},a_{h-1}) \right\|_{(U^k_{h-1,\hat{\phi}^k})^{-1}}, 1 \right\} + \sqrt{\tfrac{1}{2d}WAC_B\Delta^P_{[h-1,h],[k-W,k-1]}}, \tag{22}$$

*and for $h = 1$, we have*

$$\mathop{\mathbb{E}}_{a_1\sim\pi} \left[ f^k_1(s_1,a_1) \right] \le \sqrt{A\left(\zeta_{k,W} + \tfrac{1}{2}C_B\Delta^P_{1,[k-W,k-1]}\right)}. \tag{23}$$

*Proof.* For $h = 1$, we have

$$\mathop{\mathbb{E}}_{a_1\sim\pi} \left[ f^k_1(s_1,a_1) \right] \overset{(i)}{\le} \sqrt{\mathop{\mathbb{E}}_{a_1\sim\pi}\left[ f^k_1(s_1,a_1)^2 \right]} \overset{(ii)}{\le} \sqrt{A\left(\zeta_{k,W} + 2C_B\Delta^P_{1,[k-W,k-1]}\right)},$$

where $(i)$ follows from Jensen's inequality, and $(ii)$ follows from the importance sampling.

Then for $h \ge 2$, we derive the following bound:

$$\mathop{\mathbb{E}}_{(s_h,a_h)\sim(\hat{P}^k,\pi)} \left[ f^k_h(s_h,a_h) \Big| s_{h-1},a_{h-1} \right]$$

$$\overset{(i)}{\le} \mathop{\mathbb{E}}_{a_{h-1}\sim\pi} \left[ \left\| \hat{\phi}^k_{h-1}(s_{h-1},a_{h-1}) \right\|_{(U^k_{h-1,\hat{\phi}^k})^{-1}} \times \left( A \sum_{i=1\vee(k-W)}^{k-1} \mathop{\mathbb{E}}_{\substack{s_{h-1}\sim(P^{\star,i},\pi^i)\\a_{h-1},a_h\sim\mathcal{U}(\mathcal{A})\\s_h\sim P^{\star,i}(\cdot|s_{h-1},a_{h-1})}} \left[ f^k_h(s_h,a_h)^2 \right] \right.\right.$$

$$\left.\left. + W\Delta^P_{h-1,[k-W,k-1]} + \lambda_{k,W}d + A \sum_{i=1\vee(k-W)}^{k-1} \mathop{\mathbb{E}}_{\substack{s_{h-2}\sim(P^{\star,i},\pi^i)\\a_{h-2},a_{h-1}\sim\mathcal{U}(\mathcal{A})\\s_{h-1}\sim P^{\star,i}(\cdot|s_{h-2},a_{h-2})}} \left[ f^k_{h-1}(s_{h-1},a_{h-1})^2 \right] \right)^{-\frac{1}{2}} \right]$$

$$\overset{(ii)}{\le} \mathbb{E}_{a_{h-1}\sim\pi} \left[ \sqrt{WA(\zeta_{k,W} + 2C_B\Delta^P_{h,[k-W,k-1]}) + WA(\zeta_{k,W} + 2C_B\Delta^P_{h-1,[k-W,k-1]}) + W\Delta^P_{h-1,[k-W,k-1]} + \lambda_{k,W}d} \right.$$

$$\times \left\| \hat{\phi}_{h-1}^k(s_{h-1}, a_{h-1}) \right\|_{(U_{h-1,\hat{\phi}^k}^k)^{-1}} \Bigg]$$

$$\stackrel{(iii)}{=} \mathbb{E}_{a_{h-1} \sim \pi} \left[ \alpha_{k,W} \left\| \hat{\phi}_{h-1}^k(s_{h-1}, a_{h-1}) \right\|_{(U_{h-1,\hat{\phi}^k}^k)^{-1}} \right] + \sqrt{\frac{3}{\lambda_W} W A C_B \Delta_{\{h-1,h\},[k-W,k-1]}^P},$$

where $(i)$ follows from Lemma A.12 and because $|f_h^k(s_h, a_h)| \leq 1$; specially the first term inside the square root follows from the definition of $U_{h-1,\widehat{\phi}^k}^k$, the third term inside the square root follows from the importance sampling; $(ii)$ follows from Lemma A.10, and $(iii)$ follows because $\sqrt{a+b} \leq \sqrt{a} + \sqrt{b}, \forall a, b \geq 0$ and $\left\| \hat{\phi}_{h-1}^k(s_{h-1}, a_{h-1}) \right\|_{(U_{h-1,\hat{\phi}^k}^k)^{-1}} \leq \sqrt{\frac{1}{\lambda_W}}$. $\qquad \square$

The next lemma follows a similar argument to that of Lemma A.13 with the only difference being the expectation over which $f_h^k(s_h, a_h)$ takes.

**Lemma A.14.** *Denote* $\alpha_{k,W} = \sqrt{2WA\zeta_{k,W} + \lambda_{k,W} d}$. *For any* $k \in [K]$, *policy* $\pi$ *and reward* $r$, *for all* $h \geq 2$, *we have*

$$\mathbb{E}_{(s_h,a_h) \sim (P^{\star,k}, \pi)} \left[ f_h^k(s_h, a_h) \Big| s_{h-1}, a_{h-1} \right]$$

$$\leq \min \left\{ \alpha_{k,W} \left\| \phi_{h-1}^{\star,k}(s_{h-1}, a_{h-1}) \right\|_{(U_{h-1,\phi^{\star,k}}^k)^{-1}}, 1 \right\} + \sqrt{\frac{1}{2d} W A C_B \Delta_{\{h-1,h\},[k-W,k-1]}^P},$$

(24)

*and for* $h = 1$, *we have*

$$\mathbb{E}_{a_1 \sim \pi} \left[ f_1^k(s_1, a_1) \right] \leq \sqrt{A \left( \zeta_{k,W} + \frac{1}{2} C_B \Delta_{1,[k-W,k-1]}^P \right)}.$$

(25)

*Proof.* For $h = 1$,

$$\mathbb{E}_{a_1 \sim \pi} \left[ f_1^k(s_1, a_1) \right] \stackrel{(i)}{\leq} \sqrt{\mathbb{E}_{a_1 \sim \pi} \left[ f_1^k(s_1, a_1)^2 \right]} \stackrel{(ii)}{\leq} \sqrt{A \left( \zeta_{k,W} + \frac{1}{2} C_B \Delta_{1,[k-W,k-1]}^P \right)},$$

where $(i)$ follows from Jensen's inequality, and $(ii)$ follows from the importance sampling.

Then for $h \geq 2$, we derive the following bound:

$$\mathbb{E}_{(s_h,a_h) \sim (P^{\star,k}, \pi)} \left[ f_h^k(s_h, a_h) \Big| s_{h-1}, a_{h-1} \right]$$

$$\stackrel{(i)}{\leq} \mathbb{E}_{a_{h-1} \sim \pi} \left[ \left\| \phi_{h-1}^{\star,k}(s_{h-1}, a_{h-1}) \right\|_{(U_{h-1,\phi^{\star,k}}^k)^{-1}} \times \sqrt{A \sum_{i=1 \vee (k-W)}^{k-1} \mathbb{E}_{\substack{s_{h-1} \sim (P^{\star,i}, \pi^i) \\ a_{h-1}, a_h \sim \mathcal{U}(A) \\ s_h \sim P^{\star,k}(\cdot | s_{h-1}, a_{h-1})}} \left[ f_h^k(s_h, a_h)^2 \right] + \lambda_{k,W} d} \right]$$

$$\stackrel{(ii)}{\leq} \mathbb{E}_{a_{h-1} \sim \pi} \left[ \sqrt{wA(\zeta_{k,W} + \frac{1}{2} C_B \Delta_{h,[k-W,k-1]}^P) + wA(\zeta_{k,W} + \frac{1}{2} C_B \Delta_{h-1,[k-W,k-1]}^P) + A C_B \Delta_{h,[k-W,k-1]}^P + \lambda_{k,W} d} \right.$$

$$\left. \times \left\| \phi_{h-1}^{\star,k}(s_{h-1}, a_{h-1}) \right\|_{(U_{h-1,\phi^{\star,k}}^k)^{-1}} \right]$$

$$\stackrel{(iii)}{=} \mathbb{E}_{a_{h-1} \sim \pi} \left[ \alpha_{k,W} \left\| \phi_{h-1}^{\star,k}(s_{h-1}, a_{h-1}) \right\|_{(U_{h-1,\phi^{\star,k}}^k)^{-1}} \right] + \sqrt{\frac{1}{2d} W A C_B \Delta_{[h-1,h],[k-W,k-1]}^P},$$

where $(i)$ follows from Lemma A.12 and because $|f_h^k(s_h, a_h)| \leq 1$, the first term inside the square root follows from the definition of $U_{h-1,\widehat{\phi}^k}^k$, the third term inside the square root follows from the importance sampling, and $(ii)$ follows from Lemma A.10.

The proof is completed by noting that $|f_h^k(s_h, a_h)| \leq 1$. $\qquad \square$

The following lemma is a direct application of Lemma A.13. By this lemma, we can show that the difference of value functions can be bounded by truncated value function plus a variation term.

**Lemma A.15** (Bounded difference of value functions). *For $k \in [K]$, $\delta \geq 0$ any policy $\pi$ and reward $r$, with probability at least $1 - \delta$, we have*

$$
\left| V^{\pi}_{P^{\star,k},r} - V^{\pi}_{\hat{P}^k,r} \right|
$$
$$
\leq \hat{V}^{\pi}_{\hat{P}^k,\hat{b}^k} + \sum_{h=2}^{H} \sqrt{\frac{3}{\lambda_W} W A C_B \Delta^P_{\{h-1,h\},[k-W,k-1]}} + \sqrt{A \left( \zeta_{k,W} + 2 C_B \Delta^P_{1,[k-W,k-1]} \right)}.
$$

$$(26)$$

*Proof.* **(1):** We first show that $\left| V^{\pi}_{P^{\star,k},r} - V^{\pi}_{\hat{P}^k,r} \right| \leq \hat{V}^{\pi}_{\hat{P}^k,f^k}$.

Recall the definition of the estimated value functions $\hat{V}^{\pi}_{h,\hat{P}^k,r}(s_h)$ and $\hat{Q}^{\pi}_{h,\hat{P}^k,r}(s_h,a_h)$ for policy $\pi$:

$$
\hat{Q}^{\pi}_{h,\hat{P}^k,r}(s_h,a_h) = \min \left\{ 1, r_h(s_h,a_h) + \hat{P}^k_h \hat{V}^{\pi}_{h+1,\hat{P}^k,r}(s_h,a_h) \right\},
$$
$$
\hat{V}^{\pi}_{h,\hat{P}^k,r}(s_h) = \mathbb{E}_{\pi} \left[ \hat{Q}^{\pi}_{h,\hat{P}^k,r}(s_h,a_h) \right].
$$

We develop the proof by backward induction.

When $h = H + 1$, we have $\left| V^{\pi}_{H+1,P^{\star,k},r}(s_{H+1}) - V^{\pi}_{H+1,\hat{P}^k,r}(s_{H+1}) \right| = 0 = \hat{V}^{\pi}_{H+1,\hat{P}^k,f^k}(s_{H+1})$.

Suppose that for $h + 1$, $\left| V^{\pi}_{h+1,P^{\star,k},r}(s_{h+1}) - V^{\pi}_{h+1,\hat{P}^k,r}(s_{h+1}) \right| \leq \hat{V}^{\pi}_{h+1,\hat{P}^k,f^k}(s_{h+1})$ holds for any $s_{h+1}$.

Then, for $h$, by Bellman equation, we have,

$$
\left| Q^{\pi}_{P^{\star,k},r}(s_h,a_h) - Q^{\pi}_{h,\hat{P}^k,r}(s_h,a_h) \right|
$$
$$
= \left| P^{\star,k}_h V^{\pi}_{h+1,P^{\star,k},r}(s_h,a_h) - \hat{P}^k_h V^{\pi}_{h,\hat{P}^k,r}(s_h,a_h) \right|
$$
$$
= \left| \hat{P}^k_h \left( V^{\pi}_{h+1,P^{\star,k},r} - V^{\pi}_{h+1,\hat{P}^k,r} \right)(s_h,a_h) + \left( P^{\star,k}_h - \hat{P}^k_h \right) V^{\pi}_{h,P^{\star,k},r}(s_h,a_h) \right|
$$
$$
\overset{(i)}{\leq} \min \left\{ 1, f^k_h(s_h,a_h) + \hat{P}^k_h \left| V^{\pi}_{h+1,P^{\star,k},r} - V^{\pi}_{h+1,\hat{P}^k,r} \right| (s_h,a_h) \right\}
$$
$$
\overset{(ii)}{\leq} \min \left\{ 1, f^k_h(s_h,a_h) + \hat{P}^k_h \hat{V}^{\pi}_{h+1,\hat{P}^k,f^k}(s_h,a_h) \right\}
$$
$$
= \hat{Q}^{\pi}_{h,\hat{P}^k,f^k}(s_h,a_h), \tag{27}
$$

where $(i)$ follows because $\left\| \hat{P}^k_h(\cdot|s_h,a_h) - P^{\star,k}_h(\cdot|s_h,a_h) \right\|_{TV} = f^k_h(s_h,a_h)$ and the value function is at most 1, and $(ii)$ follows from the induction hypothesis.

Then, by the definition of $\hat{V}^{\pi}_{h,\hat{P}^k,r}(s_h)$, we have

$$
\left| V^{\pi}_{h,\hat{P}^k,r}(s_h) - V^{\pi}_{h,P^{\star,k},r}(s_h) \right|
$$
$$
= \left| \mathbb{E}_{\pi} \left[ Q^{\pi}_{h,\hat{P}^k,r}(s_h,a_h) \right] - \mathbb{E}_{\pi} \left[ Q^{\pi}_{h,P^{\star,k},r}(s_h,a_h) \right] \right|
$$
$$
\leq \mathbb{E}_{\pi} \left[ \left| Q^{\pi}_{h,\hat{P}^k,r}(s_h,a_h) - Q^{\pi}_{h,P^{\star,k},r}(s_h,a_h) \right| \right]
$$
$$
\overset{(i)}{\leq} \mathbb{E}_{\pi} \left[ \hat{Q}^{\pi}_{h,\hat{P}^k,f^k}(s_h,a_h) \right]
$$

$$= \hat{V}^{\pi}_{H,\hat{P}^k,f^k}(s_h),$$

where $(i)$ follows from Equation (27).

Therefore, by induction, we have

$$\left| V^{\pi}_{P^{\star,k},r} - V^{\pi}_{\hat{P}^k,r} \right| \leq \hat{V}^{\pi}_{\hat{P}^k,f^k}.$$

**(2):** Then we prove that

$$\hat{V}^{\pi}_{\hat{P}^k,f^k} \leq \hat{V}^{\pi}_{\hat{P}^k,\hat{b}^k} + \sum_{h=2}^{H} \sqrt{\frac{3}{\lambda_W} WAC_B \Delta^P_{\{h-1,h\},[k-W,k-1]}} + \sqrt{A\left(\zeta_{k,W} + 2C_B \Delta^P_{1,[k-W,k-1]}\right)}.$$

By Lemma A.13 and the fact that the total variation distance is upper bounded by 1, $\forall h \geq 2$, with probability at least $1 - \delta/2$, we have

$$\mathop{\mathbb{E}}_{\hat{P}^k,\pi}\left[ f^k_h(s_h,a_h) \Big| s_{h-1} \right] \leq \mathop{\mathbb{E}}_{\pi}\left[ \min\left( \alpha_{k,W} \left\| \hat{\phi}^k_{h-1}(s_{h-1},a_{h-1}) \right\|_{(U^k_{h-1,\hat{\phi}^k})^{-1}}, 1 \right) \right]$$
$$+ \sqrt{\frac{3}{\lambda_W} WAC_B \Delta^P_{\{h-1,h\},[k-W,k-1]}}. \tag{28}$$

Similarly, when $h = 1$,

$$\mathop{\mathbb{E}}_{a_1 \sim \pi}\left[ f^k_1(s_1,a_1) \right] \leq \sqrt{A\left(\zeta_{k,W} + 2C_B \Delta^P_{1,[k-W,k-1]}\right)}. \tag{29}$$

Based on Corollary A.9, Equation (28) and $\tilde{\alpha}_{k,W} = 5\alpha_{k,W}$, we have

$$\mathop{\mathbb{E}}_{\pi}\left[ \hat{b}^k_h(s_h,a_h) \Big| s_h \right] + \sqrt{\frac{3}{\lambda_W} WAC_B \Delta^P_{\{h,h+1\},[k-W,k-1]}}$$
$$\geq \mathop{\mathbb{E}}_{\pi}\left[ \min\left( \alpha_{k,W} \left\| \hat{\phi}^k_h(s_h,a_h) \right\|_{(U^k_{h,\hat{\phi}^k})^{-1}}, 1 \right) \right] + \sqrt{\frac{3}{\lambda_W} WAC_B \Delta^P_{\{h-1,h\},[k-W,k-1]}}$$
$$\geq \mathop{\mathbb{E}}_{\hat{P}^k,\pi}\left[ f^k_{h+1}(s_{h+1},a_{h+1}) \Big| s_h \right]. \tag{30}$$

For the base case $h = H$, we have

$$\mathop{\mathbb{E}}_{\hat{P}^k,\pi}\left[ \hat{V}^{\pi}_{H,\hat{P}^k,f^k}(s_H) \Big| s_{H-1},a_{H-1} \right]$$
$$= \mathop{\mathbb{E}}_{\hat{P}^k,\pi}\left[ f^k_H(s_H,a_H) \Big| s_{H-1} \right]$$
$$\leq \mathop{\mathbb{E}}_{\pi}\left[ b^k_{H-1}(s_{H-1},a_{H-1}) | s_{H-1} \right] + \sqrt{\frac{3}{\lambda_W} WAC_B \Delta^P_{\{H-1,H\},[k-W,k-1]}}$$
$$\leq \min\left\{ 1, \mathop{\mathbb{E}}_{\pi}\left[ \hat{Q}^{\pi}_{H-1,\hat{P}^k,\hat{b}^k}(s_{H-1},a_{H-1}) \Big| s_{H-1},a_{H-1} \right] \right\} + \sqrt{\frac{3}{\lambda_W} WAC_B \Delta^P_{\{H-1,H\},[k-W,k-1]}}$$
$$= \hat{V}^{\pi}_{H-1,\hat{P}^k,\hat{b}^k}(s_{H-1}) + \sqrt{\frac{3}{\lambda_W} WAC_B \Delta^P_{\{H-1,H\},[k-W,k-1]}}.$$

For any step $h + 1, h \geq 2$, assume that $\mathop{\mathbb{E}}_{\hat{P}^k,\pi}\left[ \hat{V}^{\pi}_{h+1,\hat{P}^k,f^k}(s_{h+1}) \Big| s_h \right] \leq \hat{V}^{\pi}_{h,\hat{P}^k,\hat{b}^k}(s_h) + \sum_{h'=h+1}^{H} \sqrt{\frac{3}{\lambda_W} WAC_B \Delta^P_{\{h'-1,h'\},[k-W,k-1]}}$ holds . Then, by Jensen's inequality, we obtain

$$\mathop{\mathbb{E}}_{\hat{P}^k,\pi}\left[ \hat{V}^{\pi}_{h,\hat{P}^k,f^k}(s_h) \Big| s_{h-1},a_{h-1} \right]$$

$$\leq \min\left\{1, \mathbb{E}_{\hat{P}^k,\pi}\left[f_h^k(s_h,a_h) + \hat{P}_h^k\hat{V}_{h+1,\hat{P}^k,f^k}^{\pi}(s_h,a_h)\Big|s_{h-1},a_{h-1}\right]\right\}$$

$$\overset{(i)}{\leq} \min\left\{1, \mathbb{E}_{\pi}\left[\hat{b}_{h-1}^k(s_{h-1},a_{h-1})\right] + \sqrt{\frac{3}{\lambda_W}WAC_B\Delta_{\{h-1,h\},[k-W,k-1]}^P}\right.$$

$$\left. + \mathbb{E}_{\hat{P}^k,\pi}\left[\mathbb{E}_{\hat{P}^k,\pi}\left[\hat{V}_{h+1,\hat{P}^k,f^k}^{\pi}(s_{h+1})\Big|s_h\right]\Big|s_{h-1},a_{h-1}\right]\right\}$$

$$\overset{(ii)}{\leq} \min\left\{1, \mathbb{E}_{\pi}\left[b_{h-1}^k(s_{h-1},a_{h-1})\right] + \sqrt{\frac{3}{\lambda_W}WAC_B\Delta_{\{h-1,h\},[k-W,k-1]}^P}\right.$$

$$\left. + \mathbb{E}_{\hat{P}^k,\pi}\left[\hat{V}_{h,\hat{P}^k,\hat{b}^k}^{\pi}(s_h)\Big|s_{h-1},a_{h-1}\right] + \sum_{h'=h+1}^{H}\sqrt{\frac{3}{\lambda_W}WAC_B\Delta_{\{h'-1,h'\},[k-W,k-1]}^P}\right\}$$

$$= \min\left\{1, \mathbb{E}_{\pi}\left[\hat{Q}_{h-1,\hat{P}^k,\hat{b}^k}^{\pi}(s_{h-1},a_{h-1})\right]\right\} + \sum_{h'=h}^{H}\sqrt{\frac{3}{\lambda_W}WAC_B\Delta_{\{h'-1,h'\},[k-W,k-1]}^P}$$

$$= \hat{V}_{h-1,\hat{P}^k,\hat{b}^k}^{\pi}(s_{h-1}) + \sum_{h'=h}^{H}\sqrt{\frac{3}{\lambda_W}WAC_B\Delta_{\{h'-1,h'\},[k-W,k-1]}^P},$$

where $(i)$ follows from Equation (30), and $(ii)$ is due to the induction hypothesis.

By induction, we conclude that

$$\hat{V}_{\hat{P}^k,f^k}^{\pi} = \mathbb{E}_{\pi}\left[f_1^{(s)}(s_1,a_1)\right] + \mathbb{E}_{\hat{P}^k,\pi}\left[\hat{V}_{2,\hat{P}^k,f^k}^{\pi}(s_2)\Big|s_1\right]$$

$$\leq \sqrt{A\left(\zeta_{k,W} + 2C_B\Delta_{1,[k-W,k-1]}^P\right)} + \hat{V}_{\hat{P}^k,\hat{b}^k}^{\pi} + \sum_{h'=2}^{H}\sqrt{\frac{3}{\lambda_W}WAC_B\Delta_{\{h'-1,h'\},[k-W,k-1]}^P}.$$

Combining Step 1 and Step 2, we conclude that

$$\left|V_{P^*,r}^{\pi} - V_{\hat{P}^k,r}^{\pi}\right|$$

$$\leq \hat{V}_{\hat{P}^k,\hat{b}^k}^{\pi} + \sqrt{A\left(\zeta_{k,W} + 2C_B\Delta_{1,[k-W,k-1]}^P\right)} + \sum_{h'=2}^{H}\sqrt{\frac{3}{\lambda_W}WAC_B\Delta_{\{h'-1,h'\},[k-W,k-1]}^P}.$$

$\square$

Similarly to Lemma A.15, we can prove that the total variance distance is bounded by $\hat{V}_{\hat{P}^k,\hat{b}^k}^{\tilde{\pi}_k}$ plus a variation budget term as follows. Lemmas A.15 and A.16 together justify the choice of exploration policy for the off-policy exploration.

**Lemma A.16.** *Fix* $\delta \in (0,1)$, *for any* $h \in [H], k \in [K]$, *any policy* $\pi$, *with probability at least* $1 - \delta/2$,

$$\mathbb{E}_{\substack{s_h\sim(P^{\star,k},\pi)\\s_h\sim\pi}}\left[f_h^k(s_h,a_h)\right]$$

$$\leq 2\left(\hat{V}_{\hat{P}^k,\hat{b}^k}^{\pi} + \sum_{h=2}^{H}\sqrt{\frac{3}{\lambda_W}WAC_B\Delta_{\{h-1,h\},[k-W,k-1]}^P} + \sqrt{A\left(\zeta_{k,W} + 2C_B\Delta_{1,[k-W,k-1]}^P\right)}\right).$$

*Proof.* Fix any policy $\pi$, for any $h \geq 2$, we have

$$\mathbb{E}_{\substack{s_h\sim(\hat{P}^k,\pi)\\a_h\sim\pi}}\left[\hat{Q}_{h,\hat{P}^k,\hat{b}^k}^{\pi}(s_h,a_h)\right]$$

$$= \mathbb{E}_{\substack{s_{h-1}\sim(\hat{P}^k,\pi)\\a_{h-1}\sim\pi}}\left[\hat{P}_h^k\hat{V}_{h,\hat{P}^k,\hat{b}^k}^{\pi}(s_{h-1},a_{h-1})\right]$$

$$\leq \underset{\substack{s_{h-1}\sim(\hat{P}^k,\pi)\\a_{h-1}\sim\pi}}{\mathbb{E}} \left[\min\left\{1, \hat{b}_{h-1}^k(s_{h-1},a_{h-1}) + \hat{P}_{h-1}^k \hat{V}_{h,\hat{P}^k,\hat{b}^k}^\pi(s_{h-1},a_{h-1})\right\}\right]$$

$$= \underset{\substack{s_{h-1}\sim(\hat{P}^k,\pi)\\a_{h-1}\sim\pi}}{\mathbb{E}} \left[\hat{Q}_{h-1,\hat{P}^k,\hat{b}^k}^\pi(s_{h-1},a_{h-1})\right]$$

$$\leq \dots$$

$$\leq \underset{a_1\sim\pi}{\mathbb{E}} \left[\hat{Q}_{1,\hat{P}^k,\hat{b}^k}^\pi(s_1,a_1)\right]$$

$$= \hat{V}_{\hat{P}^k,\hat{b}^k}^\pi. \tag{31}$$

Hence, for $h \geq 2$, we have

$$\underset{\substack{s_h\sim(\hat{P}^k,\pi)\\a_h\sim\pi}}{\mathbb{E}} \left[f_h^k(s_h,a_h)\right] \overset{(i)}{\leq} \underset{\substack{s_{h-1}\sim(\hat{P}^k,\pi)\\a_{h-1}\sim\pi}}{\mathbb{E}} \left[\hat{b}_{h-1}^k(s_{h-1},a_{h-1})\right] + \sqrt{\frac{3}{\lambda_W}WAC_B\Delta_{\{h-1,h\},[k-W,k-1]}^P}$$

$$\overset{(ii)}{\leq} \underset{\substack{s_{h-1}\sim(\hat{P}^k,\pi)\\a_{h-1}\sim\pi}}{\mathbb{E}} \left[\hat{Q}_{h-1,\hat{P}^k,\hat{b}^k}^\pi(s_{h-1},a_{h-1})\right] + \sqrt{\frac{3}{\lambda_W}WAC_B\Delta_{\{h-1,h\},[k-W,k-1]}^P}$$

$$\overset{(iii)}{\leq} \hat{V}_{\hat{P}^k,\hat{b}^k}^\pi + \sqrt{\frac{3}{\lambda_W}WAC_B\Delta_{\{h-1,h\},[k-W,k-1]}^P}, \tag{32}$$

where $(i)$ follows from Equation (30), $(ii)$ follows from the definition of $\hat{Q}_{h-1,\hat{P}^k,\hat{b}^k}^\pi(s_{h-1},a_{h-1})$, and $(iii)$ follows from Equation (31).

$$\underset{\substack{s_h\sim(P^{\star,k},\pi)\\s_h\sim\pi}}{\mathbb{E}} \left[f_h^k(s_h,a_h)\right]$$

$$\leq \underset{\substack{s_h\sim(\hat{P}^k,\pi)\\a_h\sim\pi}}{\mathbb{E}} \left[f_h^k(s_h,a_h)\right] + \left|\underset{\substack{s_h\sim(P^{\star,k},\pi)\\a_h\sim\pi}}{\mathbb{E}} \left[f_h^k(s_h,a_h)\right] - \underset{\substack{s_h\sim(\hat{P}^k,\pi)\\a_h\sim\pi}}{\mathbb{E}} \left[f_h^k(s_h,a_h)\right]\right|$$

$$\leq 2\left(\hat{V}_{\hat{P}^k,\hat{b}^k}^\pi + \sum_{h=2}^{H}\sqrt{\frac{3}{\lambda_W}WAC_B\Delta_{\{h-1,h\},[k-W,k-1]}^P} + \sqrt{A\left(\zeta_{k,W} + 2C_B\Delta_{1,[k-W,k-1]}^P\right)}\right),$$

where the last equation follows from Equation (32) and Lemma A.15. $\qquad\square$

**Lemma A.17.** *Denote* $\tilde{\alpha}_{k,W} = 5\alpha_{k,W}$, $\alpha_{k,W} = \sqrt{2WA\zeta_{k,W} + \lambda_{k,W}d}$, *and* $\beta_{k,W} = \sqrt{9dA\alpha_{k,W}^2 + \lambda_{k,W}d}$. *For any* $k \in [K]$, *policy* $\pi$ *and reward* $r$, *for all* $h \geq 2$, *we have*

$$\underset{(s_h,a_h)\sim(P^{\star,k},\pi)}{\mathbb{E}} \left[\hat{b}_h^k(s_h,a_h)\Big|s_{h-1},a_{h-1}\right]$$

$$\leq \beta_{k,W}\left\|\phi_{h-1}^{\star,k}(s_{h-1},a_{h-1})\right\|_{(W_{h-1,\phi^\star,k}^k)^{-1}} + \sqrt{\frac{A}{d}}\Delta_{h-1,[k-W,k-1]}^{\sqrt{P}}, \tag{33}$$

*and for* $h = 1$, *we have*

$$\underset{a_1\sim\pi}{\mathbb{E}} \left[\hat{b}_1^k(s_1,a_1)\right] \leq \sqrt{\frac{9Ad\alpha_{k,W}^2}{W}}. \tag{34}$$

*Proof.* For $h = 1$,

$$\underset{a_1\sim\pi}{\mathbb{E}} \left[\hat{b}_1^k(s_1,a_1)\right] \overset{(i)}{\leq} \sqrt{\underset{a_1\sim\pi}{\mathbb{E}} \left[\hat{b}_1^k(s_1,a_1)^2\right]} \overset{(ii)}{\leq} \sqrt{\frac{9Ad\alpha_{k,W}^2}{W}},$$

where $(i)$ follows from Jensen's inequality, and $(ii)$ follows from the importance sampling.

Then for $h \geq 2$, we first notice that

$$\sum_{i=1\vee(k-W)}^{k-1} \mathbb{E}_{\substack{s_h\sim(P^{\star,i},\pi^i)\\a_h\sim\mathcal{U}(\mathcal{A})}} \left[\hat{b}_h^k(s_h,a_h)^2\right]$$

$$= \sum_{i=1\vee(k-W)}^{k-1} \mathbb{E}_{\substack{s_h\sim(P^{\star,i},\pi^i)\\a_h\sim\mathcal{U}(\mathcal{A})}} \left[\alpha_{k,W}^2 \left\|\hat{\phi}_h^k(s_h,a_h)\right\|_{(\hat{U}_h^k)^{-1}}^2\right]$$

$$\leq \sum_{i=1\vee(k-W)}^{k-1} \mathbb{E}_{\substack{s_h\sim(P^{\star,i},\pi^i)\\a_h\sim\mathcal{U}(\mathcal{A})}} \left[9\alpha_{k,W}^2 \left\|\hat{\phi}_h^k(s_h,a_h)\right\|_{(U_{h,\hat{\phi}^k}^k)^{-1}}^2\right]$$

$$\leq 9\alpha_{k,W}^2 \mathrm{tr}\left(\sum_{i=1\vee(k-W)}^{k-1} \mathbb{E}_{\substack{s_h\sim(P^{\star,i},\pi^i)\\a_h\sim\mathcal{U}(\mathcal{A})}} \left[\hat{\phi}_h^k(s_h,a_h)\hat{\phi}_h^k(s_h,a_h)^\top\right](U_{h,\hat{\phi}^k}^k)^{-1}\right)$$

$$\leq 9\alpha_{k,W}^2 \mathrm{tr}(I_d) = 9d\alpha_{k,W}^2, \tag{35}$$

Because $\sqrt{a+b} \leq \sqrt{a} + \sqrt{b}$ and for any $k \in [K]$, $h \in [H]$, $\sqrt{\max_{(s,a)\in\mathcal{S}\times\mathcal{A}}\left\|P_h^{\star,k+1}(\cdot|s,a) - P_h^{\star,k}(\cdot|s,a)\right\|_{TV}} = \max_{(s,a)\in\mathcal{S}\times\mathcal{A}}\sqrt{\left\|P_h^{\star,k+1}(\cdot|s,a) - P_h^{\star,k}(\cdot|s,a)\right\|_{TV}}$, then for any $\mathcal{H},\mathcal{I}$, we can convert $\ell_\infty$ variation budgets to square-root $\ell_\infty$ variation budgets.

$$\sqrt{\Delta_{\mathcal{H},\mathcal{I}}^P} \leq \Delta_{\mathcal{H},\mathcal{I}}^{\sqrt{P}}. \tag{36}$$

Recall that $W_{h,\phi}^k = \sum_{i=1\vee(k-W)}^{k-1} \mathbb{E}_{(s_h,a_h)\sim(P^{\star,i},\pi^i)}\left[\phi_h(s_h,a_h)\phi_h(s_h,a_h)^\top\right] + \lambda_{k,W}I_d$. We derive the following bound:

$$\mathbb{E}_{(s_h,a_h)\sim(P^{\star,k},\pi)}\left[\hat{b}_h^k(s_h,a_h)\Big|s_{h-1},a_{h-1}\right]$$

$$\overset{(i)}{\leq} \mathbb{E}_{a_{h-1}\sim\pi}\Bigg[\left\|\phi_{h-1}^{\star,k}(s_{h-1},a_{h-1})\right\|_{(W_{h-1,\phi^{\star,k}}^k)^{-1}}$$

$$\times \sqrt{A\sum_{i=1\vee(k-W)}^{k-1} \mathbb{E}_{\substack{s_{h-1},a_{h-1}\sim(P^{\star,i},\pi^i)\\s_h\sim P_{h-1}^{(\star,i)}(\cdot|s_{h-1},a_{h-1})\\a_h\sim\mathcal{U}(\mathcal{A})}}\left[\hat{b}_h^k(s_h,a_h)^2\right] + A\Delta_{h-1,[k-W,k-1]}^P + \lambda_{k,W}d}\Bigg]$$

$$\overset{(ii)}{\leq} \mathbb{E}_{a_{h-1}\sim\pi}\left[\left\|\phi_{h-1}^{\star,k}(s_{h-1},a_{h-1})\right\|_{(W_{h-1,\phi^{\star,k}}^k)^{-1}} \times \sqrt{9dA\alpha_{k,W}^2 + \lambda_{k,W}d}\right] + \sqrt{\frac{A}{d}\Delta_{h-1,[k-W,k-1]}^P}$$

$$\overset{(iii)}{\leq} \mathbb{E}_{a_{h-1}\sim\pi}\left[\left\|\phi_{h-1}^{\star,k}(s_{h-1},a_{h-1})\right\|_{(W_{h-1,\phi^{\star,k}}^k)^{-1}} \times \sqrt{9dA\alpha_{k,W}^2 + \lambda_{k,W}d}\right] + \sqrt{\frac{A}{d}}\Delta_{h-1,[k-W,k-1]}^{\sqrt{P}}$$

where $(i)$ follows from Lemma A.12, $(ii)$ follows from Equation (35), and $(iii)$ follows Equation (36). $\square$

Before next lemma, we first introduce a notion related to matrix norm. For any matrix $A$, $\|A\|_2$ denotes the matrix norms induced by vector $\ell_2$-norm. Note that $\|A\|_2$ is also known as the spectral norm of matrix $A$ and is equal to the largest singular value of matrix $A$.

**Lemma A.18.** *With probability at least $1-\delta$, the summation of the truncated value functions $\hat{V}_{\hat{P}^k,\hat{b}^k}^{\pi_k}$ under exploration policies $\{\tilde{\pi}_k\}_{k\in[K]}$ is bounded by:*

$$\sum_{k=1}^K \hat{V}_{\hat{P}^k,\hat{b}^k}^{\tilde{\pi}^k} \leq O\left(\sqrt{KdA(A\log(|\Phi||\Psi|KH/\delta) + d^2)}\left[H\sqrt{\frac{Kd}{W}\log(W)} + \sqrt{HW^2\Delta_{[H],[K]}^\phi}\right]\right)$$

$$+\sqrt{W^3 A C_B}\Delta_{[H],[K]}^{\sqrt{P}}\Big).$$

*Proof.* For any $k \in [K]$, any policy $\pi$, we have

$$\hat{V}_{\hat{P}^k,\hat{b}^k}^{\pi} - V_{P^{\star,k},\hat{b}^k}^{\pi} \le \mathbb{E}_{\pi}\left[\hat{P}_1^k \hat{V}_{2,\hat{P}^k,\hat{b}^k}^{\pi}(s_1,a_1) - P_1^{\star,k} V_{2,P^{\star,k},\hat{b}^k}^{\pi}(s_1,a_1)\right]$$

$$= \mathbb{E}_{\pi}\left[\left(\hat{P}_1^k - P_1^{\star,k}\right)\hat{V}_{2,\hat{P}^k,\hat{b}^k}^{\pi}(s_1,a_1) + P_1^{\star,k}\left(\hat{V}_{2,\hat{P}^k,\hat{b}^k}^{\pi} - V_{2,P^{\star,k},\hat{b}^k}^{\pi}\right)(s_1,a_1)\right]$$

$$\le \mathbb{E}_{\pi}\left[f_1^k(s_1,a_1) + P_1^{\star,k}\left(\hat{V}_{2,\hat{P}^k,\hat{b}^k}^{\pi} - V_{2,P^{\star,k},\hat{b}^k}^{\pi}\right)\right]$$

$$\le \mathbb{E}_{\pi}\left[f_1^k(s_1,a_1)\right] + \mathbb{E}_{P^{\star,k},\pi}\left[\hat{V}_{2,\hat{P}^k,\hat{b}^k}^{\pi} - V_{2,P^{\star,k},\hat{b}^k}^{\pi}\right]$$

$$\le \mathbb{E}_{P^{\star,k},\pi}\left[\sum_{h=1}^H f^k(s_h,a_h)\right] = V_{P^{\star,k},f^k}^{\pi}, \tag{37}$$

As a result, we have

$$\sum_{k=1}^K \hat{V}_{\hat{P}^k,\hat{b}^k}^{\pi} \le \sum_{k=1}^K V_{P^{\star,k},f^k}^{\pi} + \sum_{k=1}^K V_{P^{\star,k},\hat{b}^k}^{\pi}.$$

**Step 1:** We first bound $\sum_{k=1}^K V_{P^{\star,k},f^k}^{\tilde{\pi}^k}$ via an **auxiliary anchor representation**.

Recall that $U_{h,\phi^{\star,k}}^{k,W} = \sum_{i=1\vee(k-W)}^{k-1}\mathbb{E}_{\substack{s_h\sim(P^{\star,i},\tilde{\pi}^i)\\a_h\sim\mathcal{U}(\mathcal{A})}}\left[\phi_h^{\star,k}(s_h,a_h)\phi_h^{\star,k}(s_h,a_h)^\top\right] + \lambda_{k,W}I_d$ and we

define $\tilde{U}_{h,\phi^{\star,k}}^{k,W,t} = \sum_{i=tW+1}^{k-1}\mathbb{E}_{\substack{s_h\sim(P^{\star,i},\tilde{\pi}^i)\\a_h\sim\mathcal{U}(\mathcal{A})}}\left[\phi_h^{\star,k}(s_h,a_h)\phi_h^{\star,k}(s_h,a_h)^\top\right] + \lambda_{k,W}I_d$. We first note that for any $h$, the following equation holds.

$$\sum_{k\in[K]}\mathbb{E}_{(s_h,a_h)\sim(P^{\star,k},\tilde{\pi}^k)}\left[\alpha_{k,W}\left\|\phi_h^{\star,k}(s_h,a_h)\right\|_{(U_{h,\phi^{\star,k}}^k)^{-1}}\right]$$

$$= \sqrt{\sum_{k\in[K]}\alpha_{k,W}^2\sum_{k\in[K]}\mathbb{E}_{(s_h,a_h)\sim(P^{\star,k},\tilde{\pi}^k)}\left[\left\|\phi_h^{\star,k}(s_h,a_h)\right\|_{(U_{h,\phi^{\star,k}}^{k,W})^{-1}}\right]^2}$$

$$= \sqrt{\sum_{k\in[K]}\alpha_{k,W}^2\sum_{t=0}^{\lfloor K/W\rfloor}\sum_{k=tW+1}^{(t+1)W}\mathbb{E}_{(s_h,a_h)\sim(P^{\star,k},\tilde{\pi}^k)}\left[\left\|\phi_h^{\star,k}(s_h,a_h)\right\|_{(U_{h,\phi^{\star,k}}^{k,W})^{-1}}\right]^2} \tag{38}$$

The $\phi^{\star,k}$ and $U$ in Equation (38) both change with the round index $k$. To deal with such an issue, we divide the entire round into $\lfloor\frac{K}{W}\rfloor + 1$ blocks with an equal length of $W$. For each block $t \in \{0,\ldots,\lfloor\frac{K}{W}\rfloor\}$, we select an **auxiliary anchor representation** $\phi^{\star,tW+1}$ and decompose Equation (38) as follows. We first derive the following equation:

$$\sum_{k=tW+1}^{(t+1)W}\mathbb{E}_{(s_h,a_h)\sim(P^{\star,k},\tilde{\pi}^k)}\left[\left\|\phi_h^{\star,k}(s_h,a_h)\right\|_{(U_{h,\phi^{\star,k}}^{k,W})^{-1}}^2\right]$$

$$- \sum_{k=tW+1}^{(t+1)W}\mathbb{E}_{(s_h,a_h)\sim(P^{\star,k},\tilde{\pi}^k)}\left[\left\|\phi_h^{\star,tW+1}(s_h,a_h)\right\|_{(U_{h,\phi^{\star,tW+1}}^{k,W})^{-1}}^2\right]$$

$$= \sum_{k=tW+1}^{(t+1)W}\mathbb{E}_{(s_h,a_h)\sim(P^{\star,k},\tilde{\pi}^k)}\left[\left\|\phi_h^{\star,k}(s_h,a_h)\right\|_{(U_{h,\phi^{\star,k}}^{k,W})^{-1}}^2 - \left\|\phi_h^{\star}(s_h,a_h)\right\|_{(U_{h,\phi^{\star,tW+1}}^{k,W})^{-1}}^2\right]$$

$$= \sum_{k=tW+1}^{(t+1)W}\mathbb{E}_{(s_h,a_h)\sim(P^{\star,k},\tilde{\pi}^k)}\left[\left\|\phi_h^{\star,k}(s_h,a_h)\right\|_{(U_{h,\phi^{\star,k}}^{k,W})^{-1}}^2 - \left\|\phi_h^{\star,k}(s_h,a_h)\right\|_{(U_{h,\phi^{\star,tW+1}}^{k,W})^{-1}}^2\right]$$

$$+ \left\| \phi_h^{\star,k}(s_h, a_h) \right\|^2_{(U_{h,\phi^\star,tW+1}^{k,W})^{-1}} - \left\| \phi_h^{\star,tW+1}(s_h, a_h) \right\|^2_{(U_{h,\phi^\star,tW+1}^{k,W})^{-1}} \Bigg].$$

$$= \underbrace{\sum_{k=tW+1}^{(t+1)W} \mathbb{E}_{(s_h,a_h)\sim(P^{\star,k},\tilde{\pi}^k)} \left[ \left\| \phi_h^{\star,k}(s_h, a_h) \right\|^2_{(U_{h,\phi^\star,k}^{k,W})^{-1}} - \left\| \phi_h^{\star,k}(s_h, a_h) \right\|^2_{(U_{h,\phi^\star,tW+1}^{k,W})^{-1}} \right]}_{(I)}$$

$$+ \underbrace{\sum_{k=tW+1}^{(t+1)W} \mathbb{E}_{(s_h,a_h)\sim(P^{\star,k},\tilde{\pi}^k)} \left[ \left\| \phi_h^{\star,k}(s_h, a_h) \right\|^2_{(U_{h,\phi^\star,tW+1}^{k,W})^{-1}} - \left\| \phi_h^{\star,tW+1}(s_h, a_h) \right\|^2_{(U_{h,\phi^\star,tW+1}^{k,W})^{-1}} \right]}_{(II)}.$$

$$(39)$$

**For term** $(II)$, we have

$$\sum_{k=tW+1}^{(t+1)W} \mathbb{E}_{(s_h,a_h)\sim(P^{\star,k},\tilde{\pi}^k)} \left[ \left\| \phi_h^{\star,k}(s_h, a_h) \right\|^2_{(U_{h,\phi^\star,tW+1}^{k,W})^{-1}} - \left\| \phi_h^{\star,tW+1}(s_h, a_h) \right\|^2_{(U_{h,\phi^\star,tW+1}^{k,W})^{-1}} \right]$$

$$\leq \sum_{k=tW+1}^{(t+1)W} \mathbb{E}_{(s_h,a_h)\sim(P^{\star,k},\tilde{\pi}^k)} \left[ \phi_h^{\star,k}(s_h,a_h)^\top (U_{h,\phi^\star,tW+1}^{k,W})^{-1} \phi_h^{\star,k}(s_h,a_h) - \phi_h^{\star,k}(s_h,a_h)^\top (U_{h,\phi^\star,tW+1}^{k,W})^{-1} \phi_h^{\star,tW+1}(s_h,a_h) \right.$$

$$\left. + \phi_h^{\star,k}(s_h,a_h)^\top (U_{h,\phi^\star,tW+1}^{k,W})^{-1} \phi_h^{\star,tW+1}(s_h,a_h) - \phi_h^{\star,tW+1}(s_h,a_h)^\top (U_{h,\phi^\star,tW+1}^{k,W})^{-1} \phi_h^{\star,tW+1}(s_h,a_h) \right]$$

$$\overset{(i)}{\leq} \sum_{k=tW+1}^{(t+1)W} \mathbb{E}_{(s_h,a_h)\sim(P^{\star,k},\tilde{\pi}^k)} \left[ \left\| \phi_h^{\star,k}(s_h, a_h) \right\|_2 \left\| (U_{h,\phi^\star,tW+1}^{k,W})^{-1} \right\|_2 \left\| \phi_h^{\star,k}(s_h, a_h) - \phi_h^{\star,tW+1}(s_h, a_h) \right\|_2 \right.$$

$$\left. + \left\| \phi_h^{\star,k}(s_h, a_h) - \phi_h^{\star,tW+1}(s_h, a_h) \right\|_2 \left\| (U_{h,\phi^\star,tW+1}^{k,W})^{-1} \right\|_2 \left\| \phi_h^{\star,tW+1}(s_h, a_h) \right\|_2 \right]$$

$$\leq \sum_{k=tW+1}^{(t+1)W} \mathbb{E}_{(s_h,a_h)\sim(P^{\star,k},\tilde{\pi}^k)} \left[ \frac{2}{\lambda_W} \left\| \phi_h^{\star,k}(s_h, a_h) - \phi_h^{\star,tW+1}(s_h, a_h) \right\|_2 \right]$$

$$\leq \frac{2W}{\lambda_W} \Delta_{\{h\},[tW+1,t(W+1)-1]}^\phi, \tag{40}$$

where $(i)$ follows from the property of the matrix norms induced by vector $\ell_2$-norm.

**For term** $(I)$, we have

$$\sum_{k=tW+1}^{(t+1)W} \mathbb{E}_{(s_h,a_h)\sim(P^{\star,k},\tilde{\pi}^k)} \left[ \left\| \phi_h^{\star,k}(s_h, a_h) \right\|^2_{(U_{h,\phi^\star,k}^{k,W})^{-1}} - \left\| \phi_h^{\star,k}(s_h, a_h) \right\|^2_{(U_{h,\phi^\star,tW+1}^{k,W})^{-1}} \right]$$

$$= \sum_{k=tW+1}^{(t+1)W} \mathbb{E}_{(s_h,a_h)\sim(P^{\star,k},\tilde{\pi}^k)} \left[ \phi_h^{\star,k}(s_h, a_h)^\top \left( (U_{h,\phi^\star,k}^{k,W})^{-1} - (U_{h,\phi^\star,tW+1}^{k,W})^{-1} \right) \phi_h^{\star,k}(s_h, a_h) \right]$$

$$= \sum_{k=tW+1}^{(t+1)W} \mathbb{E}_{(s_h,a_h)\sim(P^{\star,k},\tilde{\pi}^k)} \left[ \phi_h^{\star,k}(s_h, a_h)^\top (U_{h,\phi^\star,k}^{k,W})^{-1} \left( U_{h,\phi^\star,tW+1}^{k,W} - U_{h,\phi^\star,k}^{k,W} \right) (U_{h,\phi^\star,tW+1}^{k,W})^{-1} \phi_h^{\star,k}(s_h, a_h) \right]$$

$$\overset{(i)}{\leq} \sum_{k=tW+1}^{(t+1)W} \mathbb{E}_{(s_h,a_h)\sim(P^{\star,k},\tilde{\pi}^k)} \left[ \left\| \phi_h^{\star,k}(s_h, a_h) \right\|_2 \left\| (U_{h,\phi^\star,k}^{k,W})^{-1} \right\|_2 \left\| \left( U_{h,\phi^\star,tW+1}^{k,W} - U_{h,\phi^\star,k}^{k,W} \right) \right\|_2 \left\| (U_{h,\phi^\star,tW+1}^{k,W})^{-1} \right\|_2 \left\| \phi_h^{\star,k}(s_h, a_h) \right\|_2 \right]$$

$$\overset{(ii)}{\leq} \frac{1}{\lambda_W^2} \sum_{k=tW+1}^{(t+1)W} \mathbb{E}_{(s_h,a_h)\sim(P^{\star,k},\tilde{\pi}^k)} \left[ \left\| \sum_{i=1\vee k-W}^{k-1} \mathbb{E}_{(s_h,a_h)\sim(P^{\star,i},\tilde{\pi}^i)} \left[ \phi_h^{\star,tW+1}(s_h,a_h)\phi_h^{\star,tW+1}(s_h,a_h)^\top - \phi_h^{\star,k}(s_h,a_h)\phi_h^{\star,k}(s_h,a_h)^\top \right] \right\|_2 \right]$$

$$\leq \frac{1}{\lambda_W^2} \sum_{k=tW+1}^{(t+1)W} \sum_{i=1\vee k-W}^{k-1} \mathbb{E}_{(s_h,a_h)\sim(P^{\star,i},\tilde{\pi}^i)} \left[ \left\| \phi_h^{\star,tW+1}(s_h,a_h)\phi_h^{\star,tW+1}(s_h,a_h)^\top - \phi_h^{\star,k}(s_h,a_h)\phi_h^{\star,k}(s_h,a_h)^\top \right\|_2 \right]$$

$$\leq \frac{1}{\lambda_W^2} \sum_{k=tW+1}^{(t+1)W} \sum_{i=1\vee k-W}^{k-1} \mathbb{E}_{(s_h,a_h)\sim(P^{\star,i},\tilde{\pi}^i)} \left[ \left\| \phi_h^{\star,tW+1}(s_h,a_h)\phi_h^{\star,tW+1}(s_h,a_h)^\top - \phi_h^{\star,tW+1}(s_h,a_h)\phi_h^{\star,k}(s_h,a_h)^\top \right\|_2 \right.$$

$$\left. + \left\| \phi_h^{\star,tW+1}(s_h,a_h)\phi_h^{\star,k}(s_h,a_h)^\top - \phi_h^{\star,k}(s_h,a_h)\phi_h^{\star,k}(s_h,a_h)^\top \right\|_2 \right]$$

$$\leq \frac{2}{\lambda_W^2} \sum_{k=tW+1}^{(t+1)W} \sum_{i=1\vee k-W}^{k-1} \mathbb{E}_{(s_h,a_h)\sim(P^{\star,i},\tilde{\pi}^i)} \left[ \left\| \phi_h^{\star,k}(s_h,a_h) - \phi_h^{\star,tW+1}(s_h,a_h) \right\|_2 \right]$$

$$\leq \sum_{k=tW+1}^{(t+1)W} \sum_{i=1\vee k-W}^{k-1} \frac{2}{\lambda_W^2} \Delta_{h,[tW+1,k-1]}^\phi, \tag{41}$$

where $(i)$ follows from the property of the matrix norms induced by vector $\ell_2$-norm and $(ii)$ follows from that $\left\| \phi_h^{\star,k}(s_h,a_h) \right\|_2 \leq 1$.

Furthermore,

$$\sum_{k=tW+1}^{(t+1)W} \mathbb{E}_{(s_h,a_h)\sim(P^{\star,k},\tilde{\pi}^k)} \left[ \left\| \phi_h^{\star,tW+1}(s_h,a_h) \right\|_{(U_{h,\phi^\star,tW+1}^{k,W})^{-1}}^2 \right]$$

$$= \sum_{k=tW+1}^{(t+1)W} \mathrm{tr} \left( \mathbb{E}_{(s_h,a_h)\sim(P^{\star,k},\tilde{\pi}^k)} \left[ \phi_h^{\star,tW+1}(s_h,a_h) \phi_h^{\star,tW+1}(s_h,a_h)^\top \right] (U_{h,\phi^\star,tW+1}^{k,W})^{-1} \right)$$

$$\leq \sum_{k=tW+1}^{(t+1)W} \mathrm{tr} \left( \mathbb{E}_{(s_h,a_h)\sim(P^{\star,k},\tilde{\pi}^k)} \left[ \phi_h^{\star,tW+1}(s_h,a_h) \phi_h^{\star,tW+1}(s_h,a_h)^\top \right] (\tilde{U}_{h,\phi^\star,tW+1}^{k,W,t})^{-1} \right)$$

$$\leq \sum_{k=tW+1}^{(t+1)W} A \mathop{\mathbb{E}}_{\substack{s_h\sim(P^{\star,k},\tilde{\pi}^k) \\ a_h\sim\mathcal{U}(\mathcal{A})}} \mathrm{tr} \left[ \phi_h^{\star,tW+1}(s_h,a_h) \phi_h^{\star,tW+1}(s_h,a_h)^\top \right] (\tilde{U}_{h,\phi^\star,tW+1}^{k,W,t})^{-1}$$

$$\leq 2Ad \log(1 + \frac{W}{d\lambda_0}), \tag{42}$$

where the last equation follows from Lemma D.2.
Then combining Equations (39) to (42), we have

$$\sum_{k=tW+1}^{(t+1)W} \mathbb{E}_{(s_h,a_h)\sim(P^{\star,k},\tilde{\pi}^k)} \left[ \left\| \phi_h^{\star,k}(s_h,a_h) \right\|_{(U_{h,\phi^\star,k}^{k,W})^{-1}}^2 \right]$$

$$\leq 2Ad \log(1 + \frac{W}{d\lambda_0}) + \frac{2W}{\lambda_W} \Delta_{\{h\},[tW+1,t(W+1)-1]}^\phi + \sum_{k=tW+1}^{(t+1)W} \sum_{i=1\vee k-W}^{k-1} \frac{2}{\lambda_W^2} \Delta_{h,[tW+1,k-1]}^\phi. \tag{43}$$

Substituting Equation (43) into Equation (38), we have

$$\sum_{k\in[K]} \mathbb{E}_{(s_h,a_h)\sim(P^{\star,k},\tilde{\pi}^k)} \left[ \alpha_{k,W} \left\| \phi_h^{\star,k}(s_h,a_h) \right\|_{(U_{h,\phi^\star,k}^k)^{-1}} \right]$$

$$\leq \sqrt{ \sum_{k\in[K]} \alpha_{k,W}^2 \sum_{t=0}^{\lfloor K/W \rfloor} \sum_{k=tW+1}^{(t+1)W} \mathbb{E}_{(s_h,a_h)\sim(P^{\star,k},\tilde{\pi}^k)} \left[ \left\| \phi_h^{\star,k}(s_h,a_h) \right\|_{(U_{h,\phi^\star,k}^{k,W})^{-1}} \right]^2 }$$

$$\leq \sqrt{ \sum_{k\in[K]} \alpha_{k,W}^2 \sum_{t=0}^{\lfloor K/W \rfloor} \left[ 2Ad\log(1+\frac{W}{d\lambda_0}) + \frac{2W}{\lambda_W}\Delta_{\{h\},[tW+1,t(W+1)-1]}^\phi + \sum_{k=tW+1}^{(t+1)W}\sum_{i=1\vee k-W}^{k-1}\frac{2}{\lambda_W^2}\Delta_{h,[tW+1,k-1]}^\phi \right] }$$

$$\leq \sqrt{ K\left(2WA\zeta_{k,W} + \lambda_{k,W}d\right) \left[ \frac{2KAd}{W}\log(1+\frac{W}{d\lambda_0}) + \frac{2W}{\lambda_W}\Delta_{\{h\},[K]}^\phi + \frac{2W^2}{\lambda_W^2}\Delta_{\{h\},[K]}^\phi \right] } \tag{44}$$

where the second equation follows from Equation (43).
Then we derive the following bound:

$$\sum_{k=1}^K V_{P^{\star,k},f^k}^{\tilde{\pi}^k}$$

$$= \sum_{k\in[K]}\sum_{h\in[H]}\mathbb{E}_{(s_h,a_h)\sim(P^{\star,k},\tilde{\pi}^k)}\left[f_h^k(s_h,a_h)\right]$$

$$\overset{(i)}{\leq}\sum_{k\in[K]}\left\{\sum_{h=2}^{H}\left[\mathbb{E}_{(s_{h-1},a_{h-1})\sim(P^{\star,k},\tilde{\pi}^k)}\left[\alpha_{k,W}\left\|\phi_{h-1}^{\star,k}(s_{h-1},a_{h-1})\right\|_{(U_{h-1,\phi^{\star},k}^{k})^{-1}}\right]+\sqrt{\tfrac{1}{2d}WAC_B\Delta_{[h-1,h],[k-W,k-1]}^{P}}\right]\right.$$

$$\left.+\sqrt{A\left(\zeta_{k,W}+\frac{1}{2}C_B\Delta_{1,[k-W,k-1]}^{P}\right)}\right\}$$

$$\leq\sum_{h=1}^{H-1}\sum_{k\in[K]}\mathbb{E}_{(s_h,a_h)\sim(P^{\star,k},\tilde{\pi}^k)}\left[\alpha_{k,W}\left\|\phi_h^{\star,k}(s_h,a_h)\right\|_{(U_{h,\phi^{\star},k}^{k})^{-1}}\right]$$

$$+\sum_{k\in[K]}\sum_{h=1}^{H}\sqrt{WAC_B\Delta_{[h-1,h],[k-W,k-1]}^{P}}+\sum_{k\in[K]}\sqrt{A\zeta_{k,W}}$$

$$\overset{(ii)}{\leq}\sum_{h=1}^{H-1}\sqrt{K\left(2WA\zeta_{k,W}+\lambda_{k,W}d\right)\left[\frac{2AKd}{W}\log(1+\frac{W}{d\lambda_0})+\frac{2W}{\lambda_W}\Delta_{\{h\},[K]}^{\phi}+\frac{2W^2}{\lambda_W^2}\Delta_{\{h\},[K]}^{\phi}\right]}$$

$$+\sum_{k\in[K]}\sum_{h=1}^{H}\sqrt{WAC_B\Delta_{[h-1,h],[k-W,k-1]}^{P}}+\sum_{k\in[K]}\sqrt{A\zeta_{k,W}}$$

$$\leq\sqrt{K\left(2WA\zeta_{k,W}+\lambda_{k,W}d\right)}\left[H\sqrt{\frac{2AKd}{W}\log(1+\frac{W}{d\lambda_0})}+\sqrt{\frac{2HW}{\lambda_W}\Delta_{[H],[K]}^{\phi}}+\sqrt{\frac{2HW^2}{\lambda_W^2}\Delta_{[H],[K]}^{\phi}}\right]$$

$$+\sqrt{W^3AC_B}\Delta_{[H],[K]}^{\sqrt{P}}+\sum_{k\in[K]}\sqrt{A\zeta_{k,W}}$$

$$\leq O\left(\sqrt{K(A\log(|\Phi||\Psi|KH/\delta)+d^2)}\left[H\sqrt{\frac{2AKd}{W}\log(W)}+\sqrt{HW^2\Delta_{[H],[K]}^{\phi}}\right]+\sqrt{W^3AC_B}\Delta_{[H],[K]}^{\sqrt{P}}\right),$$
(45)

where $(i)$ follows from Lemma A.14, and $(ii)$ follows from Equation (44).

**Step 2:** We next bound $\sum_{k=1}^{K}V_{P^{\star,k},\hat{b}^k}^{\tilde{\pi}^k}$ via an **auxiliary anchor representation**.

Similarly to the proof **Step 1**, we further bound $\sum_{k\in[K]}\mathbb{E}_{(s_h,a_h)\sim(P^{\star,k},\tilde{\pi}^k)}\left[\beta_{k,W}\left\|\phi_h^{\star,k}(s_h,a_h)\right\|_{(W_{h,\phi^{\star},k}^{k})^{-1}}\right]$.

We define $W_{h,\phi^{\star},k}^{k,W}=\sum_{i=1\vee(k-W)}^{k-1}\mathbb{E}_{(s_h,a_h)\sim(P^{\star,i},\tilde{\pi}^i)}\left[\phi_h^{\star,k}(s_h,a_h)\phi_h^{\star,k}(s_h,a_h)^{\top}\right]+\lambda_{k,W}I_d$ and

$\tilde{W}_{h,\phi^{\star},k}^{k,W,t}=\sum_{i=tW+1}^{k-1}\mathbb{E}_{(s_h,a_h)\sim(P^{\star,i},\tilde{\pi}^i)}\left[\phi_h^{\star,k}(s_h,a_h)\phi_h^{\star,k}(s_h,a_h)^{\top}\right]+\lambda_{k,W}I_d$. We first note that for any $h$, we have

$$\sum_{k\in[K]}\mathbb{E}_{(s_h,a_h)\sim(P^{\star,k},\tilde{\pi}^k)}\left[\beta_{k,W}\left\|\phi_h^{\star,k}(s_h,a_h)\right\|_{(W_{h,\phi^{\star},k}^{k})^{-1}}\right]$$

$$=\sqrt{\sum_{k\in[K]}\beta_{k,W}^2\sum_{k\in[K]}\mathbb{E}_{(s_h,a_h)\sim(P^{\star,k},\tilde{\pi}^k)}\left[\left\|\phi_h^{\star,k}(s_h,a_h)\right\|_{(W_{h,\phi^{\star},k}^{k,W})^{-1}}\right]^2}$$

$$=\sqrt{\sum_{k\in[K]}\beta_{k,W}^2\sum_{t=0}^{\lfloor K/W\rfloor}\sum_{k=tW+1}^{(t+1)W}\mathbb{E}_{(s_h,a_h)\sim(P^{\star,k},\tilde{\pi}^k)}\left[\left\|\phi_h^{\star,k}(s_h,a_h)\right\|_{(W_{h,\phi^{\star},k}^{k,W})^{-1}}\right]^2}$$
(46)

The $\phi^{\star,k}$ and $W$ in Equation (46) both change with the round index $k$. To deal with this issue, we decompose it as follows. We first derive the following equation:

$$\sum_{k=tW+1}^{(t+1)W}\mathbb{E}_{(s_h,a_h)\sim(P^{\star,k},\tilde{\pi}^k)}\left[\left\|\phi_h^{\star,k}(s_h,a_h)\right\|_{(W_{h,\phi^{\star},k}^{k,W})^{-1}}^2\right]$$

$$- \sum_{k=tW+1}^{(t+1)W} \mathbb{E}_{(s_h,a_h)\sim(P^{\star,k},\tilde{\pi}^k)} \left[ \left\| \phi_h^{\star,tW+1}(s_h,a_h) \right\|_{(W_{h,\phi^\star,tW+1}^{k,W})^{-1}}^2 \right]$$

$$= \sum_{k=tW+1}^{(t+1)W} \mathbb{E}_{(s_h,a_h)\sim(P^{\star,k},\tilde{\pi}^k)} \left[ \left\| \phi_h^{\star,k}(s_h,a_h) \right\|_{(W_{h,\phi^\star,k}^{k,W})^{-1}}^2 - \left\| \phi_h^\star(s_h,a_h) \right\|_{(W_{h,\phi^\star,tW+1}^{k,W})^{-1}}^2 \right]$$

$$= \sum_{k=tW+1}^{(t+1)W} \mathbb{E}_{(s_h,a_h)\sim(P^{\star,k},\tilde{\pi}^k)} \left[ \left\| \phi_h^{\star,k}(s_h,a_h) \right\|_{(W_{h,\phi^\star,k}^{k,W})^{-1}}^2 - \left\| \phi_h^{\star,k}(s_h,a_h) \right\|_{(W_{h,\phi^\star,tW+1}^{k,W})^{-1}}^2 \right.$$
$$\left. + \left\| \phi_h^{\star,k}(s_h,a_h) \right\|_{(W_{h,\phi^\star,tW+1}^{k,W})^{-1}}^2 - \left\| \phi_h^{\star,tW+1}(s_h,a_h) \right\|_{(W_{h,\phi^\star,tW+1}^{k,W})^{-1}}^2 \right].$$

$$= \underbrace{\sum_{k=tW+1}^{(t+1)W} \mathbb{E}_{(s_h,a_h)\sim(P^{\star,k},\tilde{\pi}^k)} \left[ \left\| \phi_h^{\star,k}(s_h,a_h) \right\|_{(W_{h,\phi^\star,k}^{k,W})^{-1}}^2 - \left\| \phi_h^{\star,k}(s_h,a_h) \right\|_{(W_{h,\phi^\star,tW+1}^{k,W})^{-1}}^2 \right]}_{(III)}$$

$$+ \underbrace{\sum_{k=tW+1}^{(t+1)W} \mathbb{E}_{(s_h,a_h)\sim(P^{\star,k},\tilde{\pi}^k)} \left[ \left\| \phi_h^{\star,k}(s_h,a_h) \right\|_{(W_{h,\phi^\star,tW+1}^{k,W})^{-1}}^2 - \left\| \phi_h^{\star,tW+1}(s_h,a_h) \right\|_{(W_{h,\phi^\star,tW+1}^{k,W})^{-1}}^2 \right]}_{(IV)}.$$

$$(47)$$

**For term** $(IV)$, we have

$$\sum_{k=tW+1}^{(t+1)W} \mathbb{E}_{(s_h,a_h)\sim(P^{\star,k},\tilde{\pi}^k)} \left[ \left\| \phi_h^{\star,k}(s_h,a_h) \right\|_{(W_{h,\phi^\star,tW+1}^{k,W})^{-1}}^2 - \left\| \phi_h^{\star,tW+1}(s_h,a_h) \right\|_{(W_{h,\phi^\star,tW+1}^{k,W})^{-1}}^2 \right]$$

$$\leq \sum_{k=tW+1}^{(t+1)W} \mathbb{E}_{(s_h,a_h)\sim(P^{\star,k},\tilde{\pi}^k)} \left[ \phi_h^{\star,k}(s_h,a_h)^\top (W_{h,\phi^\star,tW+1}^{k,W})^{-1} \phi_h^{\star,k}(s_h,a_h) - \phi_h^{\star,k}(s_h,a_h)^\top (W_{h,\phi^\star,tW+1}^{k,W})^{-1} \phi_h^{\star,tW+1}(s_h,a_h) \right.$$
$$\left. + \phi_h^{\star,k}(s_h,a_h)^\top (W_{h,\phi^\star,tW+1}^{k,W})^{-1} \phi_h^{\star,tW+1}(s_h,a_h) - \phi_h^{\star,tW+1}(s_h,a_h)^\top (W_{h,\phi^\star,tW+1}^{k,W})^{-1} \phi_h^{\star,tW+1}(s_h,a_h) \right]$$

$$\overset{(i)}{\leq} \sum_{k=tW+1}^{(t+1)W} \mathbb{E}_{(s_h,a_h)\sim(P^{\star,k},\tilde{\pi}^k)} \left[ \left\| \phi_h^{\star,k}(s_h,a_h) \right\|_2 \left\| (W_{h,\phi^\star,tW+1}^{k,W})^{-1} \right\|_2 \left\| \phi_h^{\star,k}(s_h,a_h) - \phi_h^{\star,tW+1}(s_h,a_h) \right\|_2 \right.$$
$$\left. + \left\| \phi_h^{\star,k}(s_h,a_h) - \phi_h^{\star,tW+1}(s_h,a_h) \right\|_2 \left\| (W_{h,\phi^\star,tW+1}^{k,W})^{-1} \right\|_2 \left\| \phi_h^{\star,tW+1}(s_h,a_h) \right\|_2 \right]$$

$$\leq \sum_{k=tW+1}^{(t+1)W} \mathbb{E}_{(s_h,a_h)\sim(P^{\star,k},\tilde{\pi}^k)} \left[ \frac{2}{\lambda_W} \left\| \phi_h^{\star,k}(s_h,a_h) - \phi_h^{\star,tW+1}(s_h,a_h) \right\|_2 \right]$$

$$\leq \frac{2W}{\lambda_W} \Delta_{\{h\},[tW+1,t(W+1)-1]}^\phi,$$

$$(48)$$

where $(i)$ follows from the property of the matrix norms induced by vector $\ell_2$-norm.

**For term** $(III)$, we derive the following bound:

$$\sum_{k=tW+1}^{(t+1)W} \mathbb{E}_{(s_h,a_h)\sim(P^{\star,k},\tilde{\pi}^k)} \left[ \left\| \phi_h^{\star,k}(s_h,a_h) \right\|_{(W_{h,\phi^\star,k}^{k,W})^{-1}}^2 - \left\| \phi_h^{\star,k}(s_h,a_h) \right\|_{(W_{h,\phi^\star,tW+1}^{k,W})^{-1}}^2 \right]$$

$$= \sum_{k=tW+1}^{(t+1)W} \mathbb{E}_{(s_h,a_h)\sim(P^{\star,k},\tilde{\pi}^k)} \left[ \phi_h^{\star,k}(s_h,a_h)^\top \left( (W_{h,\phi^\star,k}^{k,W})^{-1} - (W_{h,\phi^\star,tW+1}^{k,W})^{-1} \right) \phi_h^{\star,k}(s_h,a_h) \right]$$

$$= \sum_{k=tW+1}^{(t+1)W} \mathbb{E}_{(s_h,a_h)\sim(P^{\star,k},\tilde{\pi}^k)} \left[ \phi_h^{\star,k}(s_h,a_h)^\top (W_{h,\phi^\star,k}^{k,W})^{-1} \right.$$
$$\left. \times \left( W_{h,\phi^\star,tW+1}^{k,W} - W_{h,\phi^\star,k}^{k,W} \right) (W_{h,\phi^\star,tW+1}^{k,W})^{-1} \phi_h^{\star,k}(s_h,a_h) \right]$$

$$\overset{(i)}{\leq} \sum_{k=tW+1}^{(t+1)W} \mathbb{E}_{(s_h,a_h)\sim(P^{\star,k},\tilde{\pi}^k)} \left[ \left\| \phi_h^{\star,k}(s_h,a_h) \right\|_2 \left\| (W_{h,\phi^{\star,k}}^{k,W})^{-1} \right\|_2 \left\| \left( W_{h,\phi^{\star},tW+1}^{k,W} - W_{h,\phi^{\star,k}}^{k,W} \right) \right\|_2 \right.$$

$$\left. \times \left\| (W_{h,\phi^{\star},tW+1}^{k,W})^{-1} \right\|_2 \left\| \phi_h^{\star,k}(s_h,a_h) \right\|_2 \right]$$

$$\overset{(ii)}{\leq} \frac{1}{\lambda_W^2} \sum_{k=tW+1}^{(t+1)W} \mathbb{E}_{(s_h,a_h)\sim(P^{\star,k},\tilde{\pi}^k)} \left[ \left\| \sum_{i=1\vee k-W}^{k-1} \mathbb{E}_{(s_h,a_h)\sim(P^{\star,i},\tilde{\pi}^i)} \left[ \phi_h^{\star,tW+1}(s_h,a_h)\phi_h^{\star,tW+1}(s_h,a_h)^\top \right. \right. \right.$$

$$\left. \left. \left. - \phi_h^{\star,k}(s_h,a_h)\phi_h^{\star,k}(s_h,a_h)^\top \right] \right\|_2 \right]$$

$$\leq \frac{1}{\lambda_W^2} \sum_{k=tW+1}^{(t+1)W} \sum_{i=1\vee k-W}^{k-1} \mathbb{E}_{(s_h,a_h)\sim(P^{\star,i},\tilde{\pi}^i)} \left[ \left\| \phi_h^{\star,tW+1}(s_h,a_h)\phi_h^{\star,tW+1}(s_h,a_h)^\top - \phi_h^{\star,k}(s_h,a_h)\phi_h^{\star,k}(s_h,a_h)^\top \right\|_2 \right]$$

$$\leq \frac{1}{\lambda_W^2} \sum_{k=tW+1}^{(t+1)W} \sum_{i=1\vee k-W}^{k-1} \mathbb{E}_{(s_h,a_h)\sim(P^{\star,i},\tilde{\pi}^i)} \left[ \left\| \phi_h^{\star,tW+1}(s_h,a_h)\phi_h^{\star,tW+1}(s_h,a_h)^\top - \phi_h^{\star,tW+1}(s_h,a_h)\phi_h^{\star,k}(s_h,a_h)^\top \right\|_2 \right.$$

$$\left. + \left\| \phi_h^{\star,tW+1}(s_h,a_h)\phi_h^{\star,k}(s_h,a_h)^\top - \phi_h^{\star,k}(s_h,a_h)\phi_h^{\star,k}(s_h,a_h)^\top \right\|_2 \right]$$

$$\leq \frac{2}{\lambda_W^2} \sum_{k=tW+1}^{(t+1)W} \sum_{i=1\vee k-W}^{k-1} \mathbb{E}_{(s_h,a_h)\sim(P^{\star,i},\tilde{\pi}^i)} \left[ \left\| \phi_h^{\star,k}(s_h,a_h) - \phi_h^{\star,tW+1}(s_h,a_h) \right\|_2 \right]$$

$$\leq \sum_{k=tW+1}^{(t+1)W} \sum_{i=1\vee k-W}^{k-1} \frac{2}{\lambda_W^2} \Delta_{h,[tW+1,k-1]}^\phi. \tag{49}$$

where $(i)$ follows from the property of the matrix norms induced by vector $\ell_2$-norm and $(ii)$ follows from that $\left\| \phi_h^{\star,k}(s_h,a_h) \right\|_2 \leq 1$.

Furthermore, we derive the following bound:

$$\sum_{k=tW+1}^{(t+1)W} \mathbb{E}_{(s_h,a_h)\sim(P^{\star,k},\tilde{\pi}^k)} \left[ \left\| \phi_h^{\star,tW+1}(s_h,a_h) \right\|_{(W_{h,\phi^{\star},tW+1}^{k,W})^{-1}}^2 \right]$$

$$= \sum_{k=tW+1}^{(t+1)W} \text{tr} \left( \mathbb{E}_{(s_h,a_h)\sim(P^{\star,k},\tilde{\pi}^k)} \left[ \phi_h^{\star,tW+1}(s_h,a_h)\phi_h^{\star,tW+1}(s_h,a_h)^\top \right] (W_{h,\phi^{\star},tW+1}^{k,W})^{-1} \right)$$

$$\leq \sum_{k=tW+1}^{(t+1)W} \text{tr} \left( \mathbb{E}_{(s_h,a_h)\sim(P^{\star,k},\tilde{\pi}^k)} \left[ \phi_h^{\star,tW+1}(s_h,a_h)\phi_h^{\star,tW+1}(s_h,a_h)^\top \right] (\tilde{W}_{h,\phi^{\star},tW+1}^{k,W,t})^{-1} \right)$$

$$\leq \sum_{k=tW+1}^{(t+1)W} \mathbb{E}_{(s_h,a_h)\sim(P^{\star,k},\tilde{\pi}^k)} \text{tr} \left[ \phi_h^{\star,tW+1}(s_h,a_h)\phi_h^{\star,tW+1}(s_h,a_h)^\top \right] (\tilde{W}_{h,\phi^{\star},tW+1}^{k,W,t})^{-1}$$

$$\leq 2d\log(1 + \frac{W}{d\lambda_0}), \tag{50}$$

where the last equation follows from Lemma D.2.
Then combining Equations (39) to (41) and (50), we have

$$\sum_{k=tW+1}^{(t+1)W} \mathbb{E}_{(s_h,a_h)\sim(P^{\star,k},\tilde{\pi}^k)} \left[ \left\| \phi_h^{\star,k}(s_h,a_h) \right\|_{(W_{h,\phi^{\star},k}^{k,W})^{-1}}^2 \right]$$

$$\leq 2d\log(1 + \frac{W}{d\lambda_0}) + \frac{2W}{\lambda_W} \Delta_{\{h\},[tW+1,t(W+1)-1]}^\phi + \sum_{k=tW+1}^{(t+1)W} \sum_{i=1\vee k-W}^{k-1} \frac{2}{\lambda_W^2} \Delta_{h,[tW+1,k-1]}^\phi. \tag{51}$$

Substituting Equation (51) into Equation (46), we have

$$\sum_{k\in[K]} \mathbb{E}_{(s_h,a_h)\sim(P^{\star,k},\tilde{\pi}^k)} \left[ \beta_{k,W} \left\| \phi_h^{\star,k}(s_h,a_h) \right\|_{(U_{h,\phi^{\star},k}^k)^{-1}} \right]$$

$$\leq \sqrt{\sum_{k\in[K]} \beta_{k,W}^2 \sum_{t=0}^{\lfloor K/W \rfloor} \sum_{k=tW+1}^{(t+1)W} \mathbb{E}_{(s_h,a_h)\sim(P^{\star,k},\tilde{\pi}^k)} \left[ \left\| \phi_h^{\star,k}(s_h,a_h) \right\|_{(U_{h,\phi^{\star},k}^{k,W})^{-1}} \right]^2 }$$

$$\leq \sqrt{\sum_{k\in[K]} \beta_{k,W}^2 \sum_{t=0}^{\lfloor K/W \rfloor} \left[ 2d\log(1+\tfrac{W}{d\lambda_0}) + \tfrac{2W}{\lambda_W}\Delta_{\{h\},[tW+1,t(W+1)-1]}^\phi + \sum_{k=tW+1}^{(t+1)W}\sum_{i=1\vee k-W}^{k-1} \tfrac{2}{\lambda_W^2}\Delta_{h,[tW+1,k-1]}^\phi \right]}$$

$$\leq \sqrt{K(9dA(2WA\zeta_{k,W}+\lambda_{k,W}d)+\lambda_{k,W}d)\left[ \tfrac{2Kd}{W}\log(1+\tfrac{W}{d\lambda_0}) + \tfrac{2W}{\lambda_W}\Delta_{\{h\},[K]}^\phi + \tfrac{2W^2}{\lambda_W^2}\Delta_{\{h\},[K]}^\phi \right]}. \tag{52}$$

where the second equation follows from Equation (51).
Then, we derive the following bound:

$$\sum_{k\in[K]} V_{P^{\star,k},\hat{b}^k}^{\tilde{\pi}^k} = \sum_{k\in[K]}\sum_{h\in[H]} \mathbb{E}_{(s_h,a_h)\sim(P^{\star,k},\tilde{\pi}^k)} \left[ \hat{b}_h^k(s_h,a_h) \right]$$

$$\overset{(i)}{\leq} \sum_{k\in[K]} \left\{ \sum_{h=2}^{H} \left\{ \mathbb{E}_{(s_{h-1},a_{h-1})\sim(P^{\star,k},\tilde{\pi}^k)} \left[ \beta_{k,W} \left\| \phi_{h-1}^{\star,k}(s_{h-1},a_{h-1}) \right\|_{(W_{h-1,\phi^{\star},k}^k)^{-1}} \right] \right.\right.$$

$$\left.\left. + \sqrt{\tfrac{A}{d}\Delta_{h-1,[k-W,k-1]}^P} \right\} + \sqrt{\tfrac{9Ad\alpha_{k,W}^2}{w}} \right\}$$

$$\leq \sum_{h=1}^{H-1}\sum_{k\in[K]} \mathbb{E}_{(s_h,a_h)\sim(P^{\star,k},\tilde{\pi}^k)} \left[ \beta_{k,W} \left\| \phi_h^{\star,k}(s_h,a_h) \right\|_{(W_{h,\phi^{\star},k}^k)^{-1}} \right]$$

$$+ W\sqrt{\tfrac{A}{d}}\Delta_{[H],[K]}^{\sqrt{P}} + \sum_{k\in[K]} \sqrt{\tfrac{9Ad\alpha_{k,W}^2}{W}}$$

$$\overset{(ii)}{\leq} \sqrt{K(9dA(2WA\zeta_{k,W}+\lambda_{k,W}d)+\lambda_{k,W}d)} \left[ H\sqrt{\tfrac{2Kd}{W}\log(1+\tfrac{W}{d\lambda_0})} + \sqrt{\tfrac{2HW}{\lambda_W}\Delta_{[H],[K]}^\phi} \right.$$

$$\left. + \sqrt{\tfrac{2HW^2}{\lambda_W^2}\Delta_{[H],[K]}^\phi} \right] + W\sqrt{\tfrac{A}{d}}\Delta_{[H],[K]}^{\sqrt{P}}$$

$$\leq O\left( \sqrt{KdA(A\log(|\Phi||\Psi|KH/\delta)+d^2)} \left[ H\sqrt{\tfrac{Kd}{W}\log(W)} + \sqrt{HW^2\Delta_{[H],[K]}^\phi} \right] \right.$$

$$\left. + W\sqrt{A}\Delta_{[H],[K]}^{\sqrt{P}} \right), \tag{53}$$

where $(i)$ follows from Lemma A.17, and $(ii)$ follows from Equation (52).
Finally, combining Equations (37), (45) and (53), we have

$$\sum_{k=1}^{K} \hat{V}_{\hat{P}^k,\hat{b}^k}^{\tilde{\pi}^k} \leq O\left( \sqrt{K(A\log(|\Phi||\Psi|KH/\delta)+d^2)}\left[ H\sqrt{\tfrac{2AKd}{W}\log(W)} + \sqrt{HW^2\Delta_{[H],[K]}^\phi} \right] + \sqrt{W^3AC_B}\Delta_{[H],[K]}^{\sqrt{P}} \right)$$

$$+ O\left( \sqrt{KdA(A\log(|\Phi||\Psi|KH/\delta)+d^2)}\left[ H\sqrt{\tfrac{Kd}{W}\log(W)} + \sqrt{HW^2\Delta_{[H],[K]}^\phi} \right] + W\sqrt{A}\Delta_{[H],[K]}^{\sqrt{P}} \right)$$

$$\leq O\left( \sqrt{KdA(A\log(|\Phi||\Psi|KH/\delta)+d^2)}\left[ H\sqrt{\tfrac{Kd}{W}\log(W)} + \sqrt{HW^2\Delta_{[H],[K]}^\phi} \right] + \sqrt{W^3A}\Delta_{[H],[K]}^{\sqrt{P}} \right).$$

$\square$

The following visitation probability difference lemma is similar to lemma 5 in Fei et al. (2020), but we remove their Assumption 1.

**Lemma A.19.** *For any transition kernels $\{P_h\}_{h=1}^H, h\in[H], j\in[h-1], s_h\in\mathcal{S}$ and policies $\{\pi_i\}_{i=1}^H$ and $\pi_j'$, we have*

$$\left| P_1^{\pi_1}\dots P_j^{\pi_j}\dots P_{h-1}^{\pi_{h-1}}(s_h) - P_1^{\pi_1}\dots P_j^{\pi_j'}\dots P_{h-1}^{\pi_{h-1}}(s_h) \right| \leq \max_{s\in\mathcal{S}} \left\| \pi_j(\cdot|s) - \pi_j'(\cdot|s) \right\|_{TV}$$

*Proof.* To prove this lemma, the only difference from lemma is that we need to show $\max_{s_j} \sum_{s_{j+1}} |P_j^{\pi_j}(s_{j+1}|s_j) - P_j^{\pi_j'}(s_{j+1}|s_j)| \le 2\max_{s\in\mathcal{S}} \left\|\pi_j(\cdot|s) - \pi_j'(\cdot|s)\right\|_{TV}$ holds without assumption. We show this as follows:

$$\max_{s_j} \sum_{s_{j+1}} |P^{\pi_j}(s_{j+1}|s_j) - P^{\pi_j'}(s_{j+1}|s_j)|$$

$$= \max_{s_j} \sum_{s_{j+1}} |\sum_a P(s_{j+1}|s_j,a)\pi_j(a|s_j) - \sum_a P(s_{j+1}|s_j,a)\pi_j'(a|s_j)|$$

$$\le \max_{s_j} \sum_{s_{j+1}} \sum_a P(s_{j+1}|s_j,a)|\pi_j(a|s_j) - \pi_j'(a|s_j)|$$

$$= \max_{s_j} \sum_{s_{j+1}} \sum_a P(s_{j+1}|s_j,a)|\pi_j(a|s_j) - \pi_j'(a|s_j)|$$

$$= \max_{s_j} \sum_a |\pi_j(a|s_j) - \pi_j'(a|s_j)| = 2\max_{s\in\mathcal{S}} \|\pi_j(\cdot|s) - \pi_j'(\cdot|s)\|_{TV}.$$

$\square$

# B  Further Discussion and Proof of Corollary 4.5

In this section, we first provide a detailed version and further discussion of Corollary 4.5 in Appendix B.1, then present the proof in Appendix B.2, and finally present an interesting special case in Appendix B.3.

## B.1  Further Discussion of Corollary 4.5

We present a detailed version of Corollary 4.5 as follows. Let $\Pi_{[1,K]}(N) = \min\{K, \max\{1, N\}\}$ for any $K, N \in \mathbb{N}$.

**Corollary B.1** (Detailed version of Corollary 4.5). *Under the same conditions of Theorem 4.4, if the variation budgets are known, then for different variation budget regimes, we can select the hyper-parameters correspondingly to attain the optimality for both $(I)$ w.r.t. $W$ and $(II)$ w.r.t. $\tau$ in Equation* (3). *For $(I)$, with $W = \Pi_{[1,K]}(\lfloor H^{\frac{1}{3}} d^{\frac{1}{3}} K^{\frac{1}{3}} (\Delta^{\sqrt{P}} + \Delta^\phi)^{-\frac{1}{3}} \rfloor)$, part $(I)$ is upper-bounded by*

$$\begin{cases} \sqrt{\frac{H^4 d^2 A}{K}(A+d^2)}, & \left(\Delta^{\sqrt{P}}+\Delta^\phi\right) \le \frac{Hd}{K^2}, \\ H^2 d^{\frac{5}{6}} A^{\frac{1}{2}} \left(A+d^2\right)^{\frac{1}{2}} (HK)^{-\frac{1}{6}} \left(\Delta^{\sqrt{P}}+\Delta^\phi\right)^{\frac{1}{6}}, & \left(\Delta^{\sqrt{P}}+\Delta^\phi\right) > \frac{Hd}{K^2}, \end{cases} \tag{54}$$

*For $(II)$ in Equation* (3)*, with $\tau = \Pi_{[1,K]}(\lfloor K^{\frac{2}{3}}(\Delta^P + \Delta^\pi)^{-\frac{2}{3}} \rfloor)$, part $(II)$ is upper bounded by*

$$\begin{cases} \frac{2H}{\sqrt{K}}, & (\Delta^P+\Delta^\pi) \le \frac{1}{\sqrt{K}}, \\ 2H^{\frac{4}{3}}(HK)^{-\frac{1}{3}}(\Delta^P+\Delta^\pi)^{\frac{1}{3}}, & \frac{1}{\sqrt{K}} < (\Delta^P+\Delta^\pi) \le K, \\ H + \frac{H(\Delta^P+\Delta^\pi)}{K}, & K < (\Delta^P+\Delta^\pi) \end{cases} \tag{55}$$

*For any $\epsilon \ge 0$, if nonstationarity is not significantly large, i.e., there exists a constant $\gamma < 1$ such that $(\Delta^P + \Delta^\pi) \le (2HK)^\gamma$ and $(\Delta^{\sqrt{P}} + \Delta^\phi) \le (2HK)^\gamma$, then PORTAL can achieve $\epsilon$-average suboptimal with polynomial trajectories.*

As a direct consequence of Theorem 4.4, Corollary 4.5 indicates that if variation budgets are known, then the agent can choose the best hyper-parameters directly based on the variation budgets. The $\mathrm{Gap}_{\mathrm{Ave}}$ can be different depending on which regime the variation budgets fall into, as can be seen in Equations (54) and (55).

At the high level, we further explain how the window size $W$ depends on the variations of environment as follows. If the nonstationarity is moderate and not significantly large, Corollary 4.5 indicates that for any $\epsilon$, Algorithm 1 achieves $\epsilon$-average suboptimal with polynomial trajectories (see the specific form in Equation (59) in Appendix B.2). If the environment is near stationary and the variation is relatively small, i.e., $(\Delta^{\sqrt{P}} + \Delta^\phi) \le Hd/K^2, (\Delta^P + \Delta^\pi) \le 1/\sqrt{K}$, then the best window size $W$

and the policy restart period $\tau$ are both $K$. This indicates that the agent does not need to take any forgetting rules to handle the variation. Then the $\mathrm{Gap}_{\mathrm{Ave}}$ reduces to $\tilde{O}\left(\sqrt{H^4 d^2 A(A + d^2)/K}\right)$, which matches the result under a stationary environment.[2]

Furthermore, it is interesting to consider a special mildly changing environment, in which the representation $\phi^\star$ stays identical and only the state-embedding function $\mu^{\star,k}$ changes over time. The average dynamic suboptimality gap in Equation (3) reduces to

$$\tilde{O}\bigg(\underbrace{\sqrt{\frac{H^4 d^2 A(A+d^2)}{W}} + \sqrt{\frac{H^2 W^3 A}{K^2}}\Delta^{\sqrt{P}}}_{(I)} + \underbrace{\frac{H}{\sqrt{\tau}} + \frac{H\tau(\Delta^P + \Delta^\pi)}{K}}_{(II)}\bigg).$$

The part $(II)$ is the same as $(II)$ in Equation (3) and by choosing the best window size of $\overline{W} = H^{\frac{1}{2}} d^{\frac{1}{2}}(A + d^2)^{\frac{1}{4}} K^{\frac{1}{2}}(\Delta^{\sqrt{P}})^{-\frac{1}{2}}$, part $(I)$ becomes

$$\tilde{O}\left(H^2 d^{\frac{3}{4}} A^{\frac{1}{2}}\left(A + d^2\right)^{\frac{3}{8}}(HK)^{-\frac{1}{4}}\left(\Delta^{\sqrt{P}}\right)^{\frac{1}{4}}\right). \tag{56}$$

Compared with the second regime in Equation (54), Equation (56) is much smaller, benefited from identical representation function $\phi^\star$. In this way, samples in previous rounds can help to estimate the representation space so that $\overline{W}$ can be larger than $W$ in terms of the order of $K$, which yields efficiency gain compared with changing $\phi^\star$.

On the other hand, if the nonstationarity is significantly large, for example, scales linearly with $K$, then for each round, the previous samples cannot help to estimate current best policy. Thus, the best $W$ and $\tau$ are both 1, and the average dynamic suboptimality gap reduces to $\tilde{O}\left(\sqrt{H^4 d^2 A\left(A + d^2\right)}\right)$. This indicates that for a fixed small accuracy $\epsilon \geq 0$, no matter how large the round $K$ is, Algorithm 1 can never achieve $\epsilon$-average suboptimality.

### B.2 Proof of Corollary B.1 (i.e., Detailed Version of Corollary 4.5)

If variation budgets are known, for different variation budgets regimes, we can tune the hyper-parameters correspondingly to reach the optimality for both the term $(I)$ that contains $W$ and the term $(II)$ that contains $\tau$.
For the first term $(I)$ in Equation (14), there are two regimes:

- Small variation: $\left(\Delta^{\sqrt{P}} + \Delta^\phi\right) \leq \frac{Hd}{K^2}$,
  - The best window size $W$ is $K$, which means that the variation is pretty mild and the environment is near stationary. In this case, by choosing window size $W = K$, the agent takes no forgetting rules to handle the variation. Then the first term $(I)$ reduces to $\sqrt{\frac{H^4 d^2 A}{K}\left(A + d^2\right)}$, which matches the result under a stationary environment.[3]
  - Then for any $\epsilon \geq 0$, with $HK$ no more than $\tilde{O}\left(\frac{H^5 d^2 A(A+d^2)}{\epsilon^2}\right)$, term $(I) \leq \epsilon$.

- Large variation: $\left(\Delta^{\sqrt{P}} + \Delta^\phi\right) > \frac{Hd}{K^2}$.
  - By choosing the window size $W = H^{\frac{1}{3}} d^{\frac{1}{3}} K^{\frac{1}{3}}\left(\Delta^{\sqrt{P}} + \Delta^\phi\right)^{-\frac{1}{3}}$, the term $(I)$ reduces to $H^2 d^{\frac{5}{6}} A^{\frac{1}{2}}\left(A + d^2\right)^{\frac{1}{2}}(HK)^{-\frac{1}{6}}\left(\Delta^{\sqrt{P}} + \Delta^\phi\right)^{\frac{1}{6}}$.
  - Since $\left(\Delta^{\sqrt{P}} + \Delta^\phi\right) \leq 2HK$, there exists $\gamma \leq 1$ s.t. $\left(\Delta^{\sqrt{P}} + \Delta^\phi\right) \leq (2HK)^\gamma$. Then $(I) \leq 2H^2 d^{\frac{5}{6}} A^{\frac{1}{2}}\left(A + d^2\right)^{\frac{1}{2}}(HK)^{-\frac{1-\gamma}{6}}$. Then for any $\epsilon \geq 0$, if $\gamma \neq 1$, with $HK$ no more than $\tilde{O}\left(\frac{d^{\frac{5}{1-\gamma}} H^{\frac{12}{1-\gamma}} A^{\frac{3}{1-\gamma}}(A+d^2)^{\frac{3}{1-\gamma}}}{\epsilon^{\frac{6}{1-\gamma}}}\right)$, term $(I) \leq \epsilon$.

---

[2]We convert the sample complexity bound under infinite horizon MDPs in Uehara et al. (2022) to the average dynamic suboptimality gap under episodic MDPs.

[3]We convert the regret bound under infinite horizon MDPs in Uehara et al. (2022) to the average dynamic suboptimality gap under episodic MDP.

For the second term $(II)$ in Equation (14), there are three regimes elaborated as follows:

- Small variation: $(\Delta^P + \Delta^\pi) \le \frac{1}{\sqrt{K}}$,

  - The best policy restart period $\tau$ is $K$, which means that the variation is pretty mild and the agent does not need to handle the variation. Then term $(II) \le \frac{2H}{\sqrt{K}}$.

  - Then for any $\epsilon \ge 0$, with $HK$ no more than $\tilde{O}\left(\frac{H^2}{\epsilon^2}\right)$, term $(II) \le \epsilon$.

- Moderate variation: $\frac{1}{\sqrt{K}} < (\Delta^P + \Delta^\pi) \le K$,

  - The best policy restart period $\tau = K^{\frac{2}{3}}(\Delta^P + \Delta^\pi)^{-\frac{2}{3}}$, and the term $(II)$ reduces to $2HK^{-\frac{1}{3}}(\Delta^P + \Delta^\pi)^{\frac{1}{3}}$.

  - Since $\left(\Delta^P + \Delta^\pi\right) \le 2HK$, there exists $\gamma \le 1$ s.t. $\left(\Delta^P + \Delta^\pi\right) \le (2HK)^\gamma$. Then the term $(II) \le 4H^{\frac{4}{3}}(HK)^{-\frac{1-\gamma}{3}}$. Then for any $\epsilon \ge 0$, if $\gamma \ne 1$, with $HK$ no more than $\tilde{O}\left(\frac{H^{\frac{4}{1-\gamma}}}{\epsilon^{\frac{3}{1-\gamma}}}\right)$, term $(II) \le \epsilon$.

- Large variation: $K < (\Delta^P + \Delta^\pi)$,

  - The variation budgets scale linearly with $K$, which indicates that the nonstationarity of the environment is significantly large and lasts for the entire rounds. Hence in each round, the previous sample can not help to estimate the current best policy. So the best policy restart period $\tau = 1$, and the second term $(II)$ reduces to $H + \frac{H(\Delta^P + \Delta^\pi)}{K} = O(H)$, which implies that Algorithm 1 can never achieve small average dynamic suboptimality gap for any large $K$.

In conclusion, the first term is upper bounded by

$$
(I) \le \begin{cases} \sqrt{\frac{H^4 d^2 A}{K}(A + d^2)}, & \left(\Delta^{\sqrt{P}} + \Delta^\phi\right) \le \frac{Hd}{K^2}, \\ H^2 d^{\frac{5}{6}} A^{\frac{1}{2}}\left(A + d^2\right)^{\frac{1}{2}}(HK)^{-\frac{1}{6}}\left(\Delta^{\sqrt{P}} + \Delta^\phi\right)^{\frac{1}{6}}, & \left(\Delta^{\sqrt{P}} + \Delta^\phi\right) > \frac{Hd}{K^2}, \end{cases} \tag{57}
$$

and the second term is upper bounded by

$$
(II) \le \begin{cases} \dfrac{2H}{\sqrt{K}}, & (\Delta^P + \Delta^\pi) \le \dfrac{1}{\sqrt{K}}, \\ 2H^{\frac{4}{3}}(HK)^{-\frac{1}{3}}(\Delta^P + \Delta^\pi)^{\frac{1}{3}}, & \dfrac{1}{\sqrt{K}} < (\Delta^P + \Delta^\pi) \le K, \\ H + \dfrac{H(\Delta^P + \Delta^\pi)}{K}, & K < (\Delta^P + \Delta^\pi) \end{cases} \tag{58}
$$

In addition, if the variation budgets are not significantly large, i.e. scale linearly with $K$, for any $\epsilon \ge 0$, Algorithm 1 can achieve $\epsilon$-average dynamic suboptimality gap with at most polynomial samples. Specifically, if there exists a constant $\gamma < 1$ such that the variation budgets satisfying $\left(\Delta^P + \Delta^\pi\right) \le (2HK)^\gamma$ and $\left(\Delta^{\sqrt{P}} + \Delta^\phi\right) \le (2HK)^\gamma$, then to achieve $\epsilon$-average dynamic suboptimality gap, i.e.,

$\text{Gap}_{\text{Ave}}(K) \le \epsilon$, Algorithm 1 only needs to collect trajectories no more than

$$
\begin{cases}
\tilde{O}\left(\dfrac{H^5 d^2 A (A+d^2)}{\epsilon^2}\right), \\
\qquad\qquad\qquad\qquad \text{if} \quad \left(\Delta^{\sqrt{P}} + \Delta^\phi\right) \le \dfrac{Hd}{K^2}, (\Delta^P + \Delta^\pi) \le \dfrac{1}{\sqrt{K}}; \\[4mm]
\tilde{O}\left(\dfrac{d^{\frac{5}{1-\gamma}} H^{\frac{12}{1-\gamma}} A^{\frac{3}{1-\gamma}} (A+d^2)^{\frac{3}{1-\gamma}}}{\epsilon^{\frac{6}{1-\gamma}}} + \dfrac{H^2}{\epsilon^2}\right), \\
\qquad\qquad\qquad\qquad \text{if} \quad \dfrac{Hd}{K^2} < \left(\Delta^{\sqrt{P}} + \Delta^\phi\right) \le (HK)^\gamma, (\Delta^P + \Delta^\pi) \le \dfrac{1}{\sqrt{K}}; \\[4mm]
\tilde{O}\left(\dfrac{H^5 d^2 A (A+d^2)}{\epsilon^2} + \dfrac{H^{\frac{4}{1-\gamma}}}{\epsilon^{\frac{3}{1-\gamma}}}\right), \\
\qquad\qquad\qquad\qquad \text{if} \quad \left(\Delta^{\sqrt{P}} + \Delta^\phi\right) \le \dfrac{Hd}{K^2}, \dfrac{1}{\sqrt{K}} < (\Delta^P + \Delta^\pi) \le (HK)^\gamma; \\[4mm]
\tilde{O}\left(\dfrac{d^{\frac{5}{1-\gamma}} H^{\frac{12}{1-\gamma}} A^{\frac{3}{1-\gamma}} (A+d^2)^{\frac{3}{1-\gamma}}}{\epsilon^{\frac{6}{1-\gamma}}}\right), \\
\qquad\qquad\qquad\qquad \text{if} \quad \dfrac{Hd}{K^2} < \left(\Delta^{\sqrt{P}} + \Delta^\phi\right) \le (HK)^\gamma, \dfrac{1}{\sqrt{K}} < (\Delta^P + \Delta^\pi) \le (HK)^\gamma.
\end{cases}
\tag{59}
$$

### B.3 A Special Case

In this subsection, we provide a characterization of a special case, where the representation $\phi^\star$ stays identical and only the state-embedding function $\mu^{\star,k}$ changes over time. In such a scenario, the variation budget $\Delta^\phi_{[H],[K]} = 0$ and the average dynamic suboptimality gap bound in Equation (14) reduces to

$$
\text{Gap}_{\text{Ave}}(K)
$$
$$
\le \tilde{O}\left(\sqrt{\dfrac{H^4 d^2 A}{W}(A+d^2)} + \sqrt{\dfrac{H^2 W^3 A}{K^2}} \Delta^{\sqrt{P}}_{[H],[K]} + \dfrac{2H}{\sqrt{\tau}} + \dfrac{3H\tau}{K}(\Delta^P_{[H],[K]} + \Delta^\pi_{[H],[K]})\right)
$$
$$
\le \tilde{O}\left(H^{\frac{7}{4}} d^{\frac{3}{4}} A^{\frac{1}{2}} (A+d^2)^{\frac{3}{8}} K^{-\frac{1}{4}} \left(\Delta^{\sqrt{P}}_{[H],[K]}\right)^{\frac{1}{4}} + HK^{-\frac{1}{3}} (\Delta^P_{[H],[K]} + \Delta^\pi_{[H],[K]})^{\frac{1}{3}}\right),
$$

where the last equation follows from the choice of the window side $W = \tilde{O}\left(H^{\frac{1}{2}} d^{\frac{1}{2}} (A+d^2)^{\frac{1}{4}} K^{\frac{1}{2}} \left(\Delta^{\sqrt{P}}_{[H],[K]}\right)^{-\frac{1}{2}}\right)$ and the policy restart period $\tau = \tilde{O}\left(K^{\frac{2}{3}} (\Delta^P_{[H],[K]} + \Delta^\pi_{[H],[K]})^{-\frac{2}{3}}\right)$ with known variation budgets.

## C Proof of Theorem 5.1 and Detailed Comparison with Wei & Luo (2021)

### C.1 Proof of Theorem 5.1

*Proof of Theorem 5.1.* Before our formal proof, we first explain several notations on different choices of $W$ and $\tau$ here.

- $(W^\star, \tau^\star)$: We denote $W^\star = d^{\frac{1}{3}} H^{\frac{1}{3}} K^{\frac{1}{3}} (\Delta^\phi + \Delta^{\sqrt{P}} + 1)^{-\frac{1}{3}}$, and $\tau^\star = \tilde{O}\left(K^{\frac{2}{3}} (\Delta^P + \Delta^\pi + 1)^{-\frac{2}{3}}\right)$.

- $(\overline{W}, \overline{\tau})$: Because $W^\star = d^{\frac{1}{3}} H^{\frac{1}{3}} K^{\frac{1}{3}} (\Delta^\phi + \Delta^{\sqrt{P}} + 1)^{-\frac{1}{3}} \le d^{\frac{1}{3}} H^{\frac{1}{3}} K^{\frac{1}{3}} \le J_W$ and $\tau^\star = \tilde{O}\left(K^{\frac{2}{3}} (\Delta^P + \Delta^\pi + 1)^{-\frac{2}{3}}\right) \le K^{\frac{2}{3}} \le J_\tau$. As a result, there exists a $\overline{W} \in \mathcal{J}_W$ such that $\overline{W} \le W^\star \le 2\overline{W}$ and a $\overline{\tau} \in \mathcal{J}_\tau$ such that $\overline{\tau} \le \tau^\star \le 2\overline{\tau}$.

- $(W^\dagger, \tau^\dagger)$: $(W^\dagger, \tau^\dagger)$ denotes the set of best choices of the window size $W$ and the policy restart period $\tau$ in feasible set that maximize $\sum_{i=1}^{\lceil K/M \rceil} R_i(W, \tau)$.

Then we can decompose the average dynamic suboptimality gap as

$$
\mathrm{Gap}_{\mathrm{Ave}}(K) = \frac{1}{K} \sum_{k \in [K]} \left[ V_{P^\star, k}^{\pi^\star} - V_{P^\star, k}^{\pi^k} \right]
$$

$$
= \frac{1}{K} \underbrace{\sum_{k=1}^{K} V_{P^\star, k}^{\pi^\star} - \sum_{i=1}^{\lceil K/M \rceil} \mathbb{E}\left[ R_i(\overline{W}, \overline{\tau}) \right]}_{(I)} + \frac{1}{K} \underbrace{\sum_{i=1}^{\lceil K/M \rceil} \mathbb{E}[R_i(\overline{W}, \overline{\tau})] - \sum_{i=1}^{\lceil K/M \rceil} \mathbb{E}[R_i(W_i, \tau_i)]}_{(II)},
$$

where the last inequality follows because if $\{\pi^k\}_{k=1}^{K}$ is the output of Algorithm 3 with the chosen window size $\{W_i\}_{i=1}^{\lceil T/M \rceil}$, $\mathbb{E}[R_i(W_i, \tau_i)] = \mathbb{E}\left[ \sum_{k=(i-1)M+1}^{\min\{iM,K\}} V_1^k \right] = \sum_{k=(i-1)M+1}^{\min\{iM,K\}} V_{P^\star, k}^{\pi^k}$ holds.

We next bound Terms $(I)$ and $(II)$ separately.

**Term (I):** We derive the following bound:

$$
\frac{1}{K} \left\{ \sum_{k=1}^{K} V_1^{\pi^{\star,k}, k} - \sum_{i=1}^{\lceil K/M \rceil} R_i(\overline{W}, \overline{\tau}) \right\}
$$

$$
\overset{(i)}{\leq} \tilde{O}\left( \sqrt{\frac{H^4 d^2 A}{\overline{W}} (A + d^2)} + \sqrt{\frac{H^3 dA}{K} (A + d^2)} \, \overline{W}^2 \Delta_{[H],[K]}^\phi + \sqrt{\frac{H^2 \overline{W}^3 A}{K^2}} \Delta_{[H],[K]}^{\sqrt{P}} \right)
$$

$$
+ \tilde{O}\left( \frac{2H}{\sqrt{\overline{\tau}}} + \frac{3H\overline{\tau}}{K} (\Delta_{[H],[K]}^P + \Delta_{[H],[K]}^\pi) \right)
$$

$$
\overset{(ii)}{\leq} \tilde{O}\left( \sqrt{\frac{H^4 d^2 A}{W^\star} (A + d^2)} + \sqrt{\frac{H^3 dA}{K} (A + d^2)} \, W^{\star 2} \Delta_{[H],[K]}^\phi + \sqrt{\frac{H^2 W^{\star 3} A}{K^2}} \Delta_{[H],[K]}^{\sqrt{P}} \right)
$$

$$
+ \tilde{O}\left( \frac{2H}{\sqrt{\tau^\star}} + \frac{3H\tau^\star}{K} (\Delta_{[H],[K]}^P + \Delta_{[H],[K]}^\pi) \right)
$$

$$
\overset{(iii)}{\leq} \tilde{O}\left( H^{\frac{11}{6}} d^{\frac{5}{6}} A^{\frac{1}{2}} (A + d^2)^{\frac{1}{2}} K^{-\frac{1}{6}} \left( \Delta^{\sqrt{P}} + \Delta^\phi + 1 \right)^{\frac{1}{6}} \right) + \tilde{O}\left( 2HK^{-\frac{1}{3}} (\Delta^P + \Delta^\pi + 1)^{\frac{1}{3}} \right),
$$

where $(i)$ follows from Equation (14), $(ii)$ follows from the definition of $\overline{W}$ and $(iii)$ follows from the definition of $W^\star$ at the beginning of the proof.

**Term (II):** We derive the following bound:

$$
\frac{1}{K} \sum_{i=1}^{\lceil K/M \rceil} \mathbb{E}\left[ R_i(\overline{W}, \overline{\tau}) \right] - \sum_{i=1}^{\lceil K/M \rceil} \mathbb{E}\left[ R_i(W_i, \tau_i) \right]
$$

$$
\overset{(i)}{\leq} \frac{1}{K} \sum_{i=1}^{\lceil K/M \rceil} \mathbb{E}\left[ R_i(W^\dagger, \tau^\dagger) \right] - \sum_{i=1}^{\lceil K/M \rceil} \mathbb{E}\left[ R_i(W_i, \tau_i) \right]
$$

$$
\overset{(ii)}{\leq} \tilde{O}(M\sqrt{J\lceil K/M \rceil}/K)
$$

$$
= \tilde{O}(\sqrt{JKM}) = \tilde{O}(H^{\frac{1}{6}} d^{\frac{1}{6}} K^{-\frac{1}{6}}),
$$

where $(i)$ follows from the definition of $W^\dagger$ and $(ii)$ follows from Theorem 3.3 in Bubeck & Cesa-Bianchi (2012) with the adaptation that reward $R_i \leq M$ and the number of iteration is $\lceil K/M \rceil$. Then combining the bounds on terms $(I)$ and $(II)$, we have

$$
\mathrm{Gap}_{\mathrm{Ave}}(K) \leq \tilde{O}\left( H^{\frac{11}{6}} d^{\frac{5}{6}} A^{\frac{1}{2}} (A + d^2)^{\frac{1}{2}} K^{-\frac{1}{6}} \left( \Delta^{\sqrt{P}} + \Delta^\phi + 1 \right)^{\frac{1}{6}} \right) + \tilde{O}\left( 2HK^{-\frac{1}{3}} (\Delta^P + \Delta^\pi + 1)^{\frac{1}{3}} \right).
$$

$\square$

## C.2 Detailed Comparison with Wei & Luo (2021)

Based on the theoretical results Theorem 4.4 and taking Algorithm 1 PORTAL as a base algorithm, we can also use the black-box techniques called MASTER in Wei & Luo (2021) to handle the unknown variation budgets. But such an approach of MASTER+PORTAL turns out to have a larger average dynamic suboptimality gap than Algorithm 3. Denote $\Delta = \Delta^\phi + \Delta^{\sqrt{P}} + \Delta^\pi$.

To explain, first choose $\tau = k$ and $W = K$ in Algorithm 1. Then Algorithm 1 reduces to a base algorithm with the suboptimality gap of $\tilde{O}(\sqrt{H^4 d^2 A(A+d^2)/K} + \sqrt{H^3 dAK(A+d^2)}\Delta)$, which satisfies Assumption 1 in Wei & Luo (2021) with $\rho(t) = \sqrt{H^4 d^2 A(A+d^2)/t}$ and $\Delta_{[1,t]} = \sqrt{H^3 dAt(A+d^2)}\Delta$. Then by Theorem 2 in Wei & Luo (2021), the dynamic regret using MASTER+PORTAL can be upper-bounded by $\tilde{O}(H^{\frac{11}{6}} d^{\frac{5}{6}} A^{\frac{1}{2}} (A+d^2)^{\frac{1}{2}} K^{\frac{2}{3}} \Delta^{\frac{1}{3}}) = \tilde{O}(K^{\frac{5}{6}} \Delta^{\frac{1}{3}})$. Here, we find the order dependency on $d, H, A$ is the same as Theorem 5.1 and hence mainly focus on the order dependency on $K$ and $\Delta$ in the following comparison. Such a bound on dynamic regret can be converted to the average dynamic suboptimality gap as $\tilde{O}(K^{-\frac{1}{6}} \Delta^{\frac{1}{3}})$. Then it can obersed that for not too small variation budgets, i.e., $\Delta \geq \tilde{O}(1)$, the order dependency on $\Delta$ is higher than that of Algorithm 3.

Specifically, if we consider the special case when the representation stays identical and denote $\tilde{\Delta} = \Delta^{\sqrt{P}} + \Delta^\pi$, then the average dynamic suboptimality gap of MASTER+PORTAL is still $\tilde{O}(K^{-\frac{1}{6}} \tilde{\Delta}^{\frac{1}{3}})$. With small modifications on the parameters, by following the analysis similar to that of Theorem 5.1 and Appendix B.3, we can show that the average dynamic suboptimality gap of Algorithm 3 satisfies $\tilde{O}(K^{-\frac{1}{4}} \tilde{\Delta}^{\frac{1}{4}})$, which is smaller than MASTER+PORTAL.

# D   Auxiliary Lemmas

In this section, we provide two lemmas that are commonly used for the analysis of MDP problems.

The following lemma (Dann et al., 2017) is useful to measure the difference between two value functions under two MDPs and reward functions.

**Lemma D.1.** (Simulation Lemma). *Suppose $P_1$ and $P_2$ are two MDPs and $r_1$, $r_2$ are the corresponding reward functions. Given a policy $\pi$, we have,*

$$
V_{h,P_1,r_1}^\pi(s_h) - V_{h,P_2,r_2}^\pi(s_h)
$$
$$
= \sum_{h'=h}^H \mathop{\mathbb{E}}_{\substack{s_{h'} \sim (P_2,\pi) \\ a_{h'} \sim \pi}} \left[ r_1(s_{h'}, a_{h'}) - r_2(s_{h'}, a_{h'}) + (P_{1,h'} - P_{2,h'}) V_{h'+1,P_1,r}^\pi(s_{h'}, a_{h'}) | s_h \right]
$$
$$
= \sum_{h'=h}^H \mathop{\mathbb{E}}_{\substack{s_{h'} \sim (P_1,\pi) \\ a_{h'} \sim \pi}} \left[ r_1(s_{h'}, a_{h'}) - r_2(s_{h'}, a_{h'}) + (P_{1,h'} - P_{2,h'}) V_{h'+1,P_2,r}^\pi(s_{h'}, a_{h'}) | s_h \right].
$$

The following lemma is a standard inequality in the regret analysis for linear MDPs in reinforcement learning (see Lemma G.2 in Agarwal et al. (2020b) and Lemma 10 in Uehara et al. (2022)).

**Lemma D.2** (Elliptical Potential Lemma). *Consider a sequence of $d \times d$ positive semidefinite matrices $X_1, \ldots, X_N$ with $\mathrm{tr}(X_n) \leq 1$ for all $n \in [N]$. Define $M_0 = \lambda_0 I$ and $M_n = M_{n-1} + X_n$. Then the following bound holds:*

$$
\sum_{n=1}^N \mathrm{tr}\left(X_n M_{n-1}^{-1}\right) \leq 2\log \det(M_N) - 2\log \det(M_0) \leq 2d \log\left(1 + \frac{N}{d\lambda_0}\right).
$$

