# OpenReview forum: "Provably Efficient Algorithm for Nonstationary Low-Rank MDPs"
_NeurIPS.cc/2023/Conference — NeurIPS 2023 poster_

### Official Review · Reviewer_5jBS · 2023-07-06

**Soundness:** 3 good
**Presentation:** 3 good
**Contribution:** 3 good
**Rating:** 7
**Confidence:** 3

**Summary:**

The paper presents a new way to perform off-policy RL coupled with representation learning over non-stationary low-rank MDPs, both in a parameter-based and a parameter-free fashion. The paper proposes some interesting new algorithmic techniques to tackle this (off-policy
exploration, data-transfer model learning, and target policy update with periodic restart).

**Strengths:**

# originality
The work seems quite original in both the analysis and the contribution.


# quality

The quality of this work is satisfactory and the analysis is thoughtful.

# clarity

The exposition is clear and neat, I would suggest presenting the algorithms differently (See weaknesses)


# significance

The work seems to provide a significant contribution to the field, due to the several techniques it introduces.

**Weaknesses:**

My only comment is about the exposition of the algorithm. I would suggest not citing equations but reporting them, and providing the output of the subroutines.


**Questions:**

No questions

**Limitations:**

I didn't find any section where the limitations of the proposed method were addressed

---

> ### Author Rebuttal · Authors · 2023-08-09
>
> We thank the reviewer for providing the helpful review! Below we address the reviewer’s comments.
>
> Q: My only comment is about the exposition of the algorithm. I would suggest not citing equations but reporting them, and providing the output of the subroutines.
>
> A: Thanks for the suggestions. We will revise the presentation of the algorithm as the reviewer suggested.
>
> Finally, we thank the reviewer again for the helpful comments and suggestions for our work. We are more than happy to answer any further questions that you may have during the discussion period.

---

### Official Review · Reviewer_ga5z · 2023-07-07

**Soundness:** 1 poor
**Presentation:** 3 good
**Contribution:** 2 fair
**Rating:** 3
**Confidence:** 3

**Summary:**

This paper present an algorithm for solving non-stationary low-rank MDP. It also provide a theoretical analysis of their algorithm,

**Strengths:**

This is an interesting question. The presentation is clear.

**Weaknesses:**

I sketch through the results and the proof. I doubt that the proof might be wrong.  In page 14, the inequality (i) require the author to provide an upper bound of E_{P^*} f_h^k - E_{\hat P}f_h^k. The paper says that inequality (i) holds since Lemma A.13 and Lemma A.15. However, none of the lemma is related with the term I mentioned above. Therefore, I am worried that the proof might be wrong.

More over, the writing of the proof can be improve, and the clearity of the paper have room for improvement. For example, it would be helpful to provide the definition of $\lambda$ in the main part. It would also be helpful to mention that $W\leq K$ in the main part.

**Questions:**

See the 'weakness' section.

**Limitations:**

See the 'weakness' section.

---

> ### Author Rebuttal · Authors · 2023-08-09
>
> We thank the reviewer for providing the helpful review! Below we address the reviewer’s comments.
>
> Q1: I sketch through the results and the proof. I doubt that the proof might be wrong. In page 14, the inequality (i) require the author to provide an upper bound of $E_{P^*,{\pi^{\star,k}}} f_h^k - E_{\hat P,{\pi^{\star,k}}}f_h^k$.  The paper says that inequality (i) holds since Lemma A.13 and Lemma A.15. However, none of the lemma is related with the term I mentioned above. Therefore, I am worried that the proof might be wrong.
>
>
>
> A1: **We confirm that the proof is correct!** The term $E_{P^*,{\pi^{\star,k}}} f_h^k - E_{\hat P,{\pi^{\star,k}}}f_h^k$ can indeed be upper bounded by using Lemma A.15. To see this, Lemma A.15 provides upper bounds on the value function difference between estimated models and true models for any generic reward. Now consider a specific reward function $r^k$, which takes the value $f_h^k$ at step $h$ and takes the value 0 at all other steps. For such a reward, we have $ V_{P^{\star,k},r^k}^{\pi^{\star,k}}-V_{\hat{P}^k,r^k}^{\pi^{\star,k}}=E_{P^*,{\pi^{\star,k}}} f_h^k - E_{\hat P,{\pi^{\star,k}}}f_h^k$.
> In this way, the bound in Lemma A.15 on
> $ V_{P^{\star,k},r^k}^{\pi^{\star,k}}-V_{\hat{P}^k,r^k}^{\pi^{\star,k}} $
> becomes a bound on
> $E_{P^*,{\pi^{\star,k}}} f_h^k - E_{\hat P,{\pi^{\star,k}}}f_h^k$.
>
> Q2: More over, the writing of the proof can be improve, and the clearity of the paper have room for improvement. For example, it would be helpful to provide the definition of $\lambda$ in the main part. It would also be helpful to mention that $W \leq K$ in the main part.
>
> A2: Thanks for the suggestions. We will make our best efforts to improve both the clarity and readability of our paper in the revision. Specifically, regarding the regularizer $\lambda$,
> we have defined it in line 1 of Algorithm 1 and specified its value in line 285 in Theorem 4.4. We will further emphasize these in the revision. We will mention $W \leq K$ as the reviewer suggested in the main part.
>
> Finally, if our response resolves your concerns to a satisfactory level, we kindly ask the reviewer to consider raising the score of your evaluation. Certainly, we are more than happy to answer any further questions that you may have during the discussion period. We thank the reviewer again for the helpful comments and suggestions for our work.

---

> ### Author Response · Authors · 2023-08-17
> **Dear Reviewer ga5z, your feedback is important to us**
>
> Dear Reviewer ga5z,
>
> As the author-reviewer discussion period has started for one week, and will end very soon, we would greatly appreciate any feedback you could provide. Could you please check our response at your earliest convenience? This way, if you have further questions, we will still have time to respond before this discussion period ends. We thank the reviewer very much in advance for your time and efforts.
>
> Best regards,
>
> Authors.

---

> ### Author Response · Authors · 2023-08-20
> **Your feedback is important for us**
>
> Dear Reviewer ga5z,
>
> Since the author-reviewer discussion period will end in about one and half days, could you please check our response at your earliest convenience? This way, if you have further questions, we will still have time to respond. We thank the reviewer very much in advance for your time and efforts.
>
> Best Regards,
> Authors

---

### Official Review · Reviewer_FzWn · 2023-07-10

**Soundness:** 3 good
**Presentation:** 2 fair
**Contribution:** 3 good
**Rating:** 6
**Confidence:** 2

**Summary:**

This paper investigates nonstationary RL in the context of episodic low-rank MDPs, where both transition kerns and rewards may change from round to round, and where the low-rank model contains unknown representations. The goal of this work is then to develop methods with low average dynamic suboptimality gap, which is a measure of how well the agent performs across all rounds. The authors start by proposing PORTAL, a base algorithm with three main ideas: (a) off-policy exploration for data collection which is particularly useful for non-stationarity; (b) the transfer of recent history data collected under previous different transition kernels for benefiting the estimation of the current model; and (c) updating target policies with periodic restart. PORTAL requires few hyperparameters, which greatly affect performance but may not be easy to set. for this reason, the authors also propose Ada-PORTAL, a second method that uses PORTAL as a subroutine and additionally treats the selection of the hyperparameters as a bandit problem. Under standard assumptions for low-rank MDPs and assuming that non-stationarity is not significant, the authors show that both methods achieve arbitrarily small average dynamic suboptimality gap with polynomial sample complexity.

**Strengths:**

1. The paper extends the literature in low-rank MDPs by incorporating non-stationarity. In particular, the work advances the recent research on nonstationary linear (mixture) MDPs. Overall, the contribution is interesting, given the attention that both low-rank MDPs as well as non-stationarity have recently attracted.
2. The paper makes assumptions that are consistent with both the low-rank MDP literature as well as the non-stationary RL literature.
3. I was only able to check part of the derivations, which looked correct. I liked that the authors clarified how their work is different from recent works on non-stationarity or low-rank MDPs, not just in terms of the problem statement but also in terms of the new technical challenges.

**Weaknesses:**

1. I appreciate the fact that the authors provided insights and clarifications in several parts of the paper. But the paper is still hard to follow. I think what generally helps with such works is to provide an overview of the proof strategy, explain which exactly elements from prior works are used, and what elements are new. I acknowledge that the paper contains some discussions here and there, but I feel that a more structured approach would significantly improve the exposition.
2. In its current form and presentation, I feel that the paper is only relevant to a very small audience of the ML/AI community.

**Questions:**

1. Why periodic restart in this work does not require a smooth visitation frequency assumption (compared to Fei et al., 2020)?
2. The authors use EXP3-P in Ada-PORTAL. Is this choice critical or could other choices be made instead? Are there reasons to prefer one over the others?
3. The authors stress the significance of \tau and W. For the first algorithm W and \tau, how exactly should the be selected? Could the authors provide clear guidelines?

**Limitations:**

None.

---

> ### Author Rebuttal · Authors · 2023-08-09
>
> We thank the reviewer for providing the helpful review! Below we address the reviewer’s comments, and we will revise the paper as suggested. If our response resolves your concerns to a satisfactory level, we kindly ask the reviewer to consider possibly raising the score of your evaluation.
>
> Q1: Provide an overview of the proof strategy.
>
> A1: Thanks for the comments. Please note that we have provided a proof sketch starting from line 451 in Appendix A. Based on the suggestions, we will further explain new elements as follows in the revision.
>
> In step 1, we decompose the average dynamic suboptimal gap into three terms, which can be divided into two parts: $T_1$ and $T_2$.
>
> In step 2, for part $T_1$ corresponding to model estimation errors, the proof contains two sub-steps. (i) First by Lemmas A.13 and A.15, we show model estimation errors can be bounded by the average of the truncated value functions of bonus terms plus a variation budgets term. In this sub-step, we provide a new nonstationary MLE guarantee as in Lemma A.10, which is novel as it characterizes the performance of MLE under nonstationary environment for the first time. We also develop the Nonstationary Step Back Lemma to handle distribution shift from the training data distribution to target data distribution. (ii) We then upper bound the average of the truncated value function as in Lemma A.18. Here, we adopt an auxiliary anchor representation for each block to deal with the challenge arising from time-varying representations when using standard elliptical potential based methods. This design of auxiliary anchor representation is new because of the unknown and nonstationary representations.
>
> In step 3, we bound part $T_2$ corresponding to performance difference bound. The analysis of step 3 is inspired by [4], which explores policy optimization under tabular MDPs, and we extend it to low-rank MDPs. Our novelty lies in adopting different techniques to bound Eq. (11) (please refer to A3), which removes the Assumption 1 in [4].
>
> Q2: The paper is only relevant to small audience of ML/AI community.
>
> A2: We will revise the paper as follows in order for the paper to be appreciated by a large set of audience in ML/AI community. (i) We will provide application examples to connect the nonstationary low-rank MDP model to realistic applications such as real-time online auctions, arm manipulation in robotics, autonomous driving, etc. In this way, we hope to attract a broad set of audience in these application areas. (ii) In order for the paper to be appreciated by a large set of audience who are interested in nonstationary RL, we will revise our description of the algorithm to help readers catch our main design ideas to handle nonstationarity and improve the sample efficiency. We will also explain our results on the sample complexity from the high level to make it easier for people to understand the performance guarantee. (iii) On the theory side, we will explain our proof techniques better and clearer to make them more accessible to people who will potentially use these techniques to study various other RL problems with nonstationarity, such as offline RL, multi-agent RL, RL with safety constraints, etc.
>
> Q3: Why this work does not require a smooth visitation assumption?
>
> A3:  Fei et al., (2020) [4] chose to make the following smooth visitation assumption in proving Lemma 5:
> $\max_{s_j}\sum_{s_{j+1}}|P^{\pi^{(j)}}(s_{j+1}|s_j)-P^{\pi^\prime}(s_{j+1}|s_j)| \leq C\\|\pi^{(j)}-\pi^\prime\\|_{\infty}$.
>
> Interestingly, we discovered that such an inequality can be proven to hold with equality as follows.
> \begin{align*}
>     & \quad \max_{s_j}\sum_{s_{j+1}}|P^{\pi^{(j)}}(s_{j+1}|s_j)-P^{\pi^\prime}(s_{j+1}|s_j)|\\\\
> &=\max_{s_j}\sum_{s_{j+1}}|\sum_{a}P(s_{j+1}|s_j,a)\pi^{(j)}(a|s_j)-\sum_{a}P(s_{j+1}|s_j,a)\pi^\prime(a|s_j)|\\\\
> &\leq \max_{s_j}\sum_{s_{j+1}}\sum_{a}P(s_{j+1}|s_j,a)|\pi^{(j)}(a|s_j)-\pi^\prime(a|s_j)|\\\\
> &=\max_{s_j}\sum_{a}\sum_{s_{j+1}}P(s_{j+1}|s_j,a)|\pi^{(j)}(a|s_j)-\pi^\prime(a|s_j)|\\\\
> &=\max_{s_j}\sum_{a}|\pi^{(j)}(a|s_j)-\pi^\prime(a|s_j)|=\\|\pi^{(j)}-\pi^\prime\\|_{\infty}\\\\
> \end{align*}
>
> Q4: Is this choice of EXP3-P critical or could be made instead? Reasons?
>
> A4: We believe other choices can be made instead. In Ada-PORTAL, the selection of the hyper-parameters such as $W$ and $\tau$ can be regarded as an adversarial bandit problem. In this paper, we choose EXP3-P because it is one of the most popular algorithms for adversarial bandits. Any other efficient algorithms for adversarial bandits, for example SAO in [5], can also be applied to serve the purpose. Given that SAO exhibits the same regret performance as EXP3-P (omitting the logarithmic term), there is no special preference on these two algorithms.
>
> Q5: The authors stress the significance of $\tau$ and $W$. For the first algorithm $W$ and $\tau$, how exactly should the be selected? Could the authors provide clear guidelines?
>
> A5: Thanks for the comments. Please note that in Appendix B, we explained how to select $W$ and $\tau$ in detail. More specifically, Theorem 4.4 provides an upper bound on the average suboptimality gap $\mathrm{Gap_{Ave}}(K)$ w.r.t. the values of $\tau$ and $W$. We can then select their values to minimize such an upper bound. The optimal values of $\tau$ and $W$ can be found in  Appendix B.
>
> [1] Cheung, W. C., et.al (2023). Nonstationary reinforcement learning: The blessing of (more) optimism. Management Science.
>
> [2] Cheung, W. C., et.al (2022). Hedging the drift: Learning to optimize under nonstationarity. Management Science, 68(3), 1696-1713.
>
> [3] Guo, X., et.al (2019). Learning mean-field games. NeurIPS 32.
>
> [4] Fei, Y.,  et.al (2020). Dynamic regret of policy optimization in non-stationary environments. NeurIPS 33.
>
> [5] Bubeck, Sébastien, and Aleksandrs (2012) Slivkins. The best of both worlds: Stochastic and adversarial bandits. COLT.

---

> > ### Comment · Reviewer_FzWn · 2023-08-21
> > **thank you for the response**
> >
> > I thank the authors for their detailed response. I will increase my score from 5 to 6, since all my concerns and questions were addressed. I encourage the authors to incorporate the various clarifications into the final version of this work for improved clarity, presentation and exposition.

---

### Official Review · Reviewer_niTs · 2023-07-26

**Soundness:** 3 good
**Presentation:** 2 fair
**Contribution:** 3 good
**Rating:** 7
**Confidence:** 3

**Summary:**

This paper studies reinforcement learning (RL) under non-stationary MDPs where the state transition and reward function changes across episodes. The paper focus on the low-rank MDP setting, in which the state transition function admits a low-rank decomposition. The authors propose a novel algorithm, namely PORTAL, along with theoretical results that characterize a bound for the average suboptimality gap of the learned policies over the episodes. Additionally, the paper also introduces an algorithm, namely Ada-PORTAL, for adaptive tuning of the hyperparameters of PORTAL.

**Strengths:**

* The paper is the first to study non-stationarity in low-rank MDPs.

* The proposed method is sound and results in a better suboptimality gap than previous related work (Wei & Luo, 2021).


**Weaknesses:**

* A few algorithmic decisions and assumptions require better explanation/justification to improve the clarity of the paper (see below).

* The method relies on an assumption that is not very realistic (reachability) and should be better discussed (see Limitations below).

* (Minor) The paper does not have an experimental section.


**Questions:**

Below, I have a few questions and constructive feedback for the authors:

1) In Section 3.2, I would suggest making it clearer what a “round” is. For instance, does each round have a fixed number of episodes?

2) The current version of Section 4.1 is a bit hard to follow. For instance, in Step 1, terms such as “bonus term” and “target policy” are used without being previously defined.

3) “However, in low-rank MDPs, the bonus term $b^k_k$ cannot serve as a point-wise uncertainty measure”. I suggest elaborating on why this is true.

4) In Step 1, why the exploration policy takes two uniformly chosen actions? This algorithmic decision is not discussed or explained.

5) I suggest writing comments in Algorithm 1 mentioning which lines correspond to Step 1, 2, and 3 for improving the clarity.

6) In Algorithm 1, why states are indexed two times (subscript and superscript) with the time step $h$, e.g., $s_h^{k, h}$? What the indexes mean was not defined.

7) What is the role of the coefficient $\tilde{\alpha_{k,W}}$? How is this selected?

8) “Compared with the previous work using periodic restart (Fei et al., 2020), whose choice of $\tau$ is based on a certain smooth visitation assumption, we remove such an assumption and hence our choice of $\tau$ is applicable to more general model classes.”
Why can you remove this assumption? How is the value of $\tau$ selected? Please elaborate on why it is necessary to reset the target policy every $\tau$ episode.


**Limitations:**

Assumption 4.3 is not very realistic: in most MDPs, it is not possible to reach any state of the MDP in a single step. For instance, in a maze, the probability of reaching the end of the maze from the initial state in a single step is zero. Is this assumption also made in related works, e.g., Wei & Luo (2021)?

---

> ### Author Rebuttal · Authors · 2023-08-09
>
> We thank the reviewer for providing helpful reviews! Below we address the reviewer’s comments, and we will revise the paper as suggested. If our response resolves your concerns to a satisfactory level, we kindly ask the reviewer to consider possibly raising the score of your evaluation.
>
> Q1: Indicate what a “round” is.
>
> A1: Yes, each "round" has a fixed number of episodes.
>
> Q2: Revise current version of Section 4.1. Define “bonus term” and “target policy”.
>
> A2: The term "bonus term" is the level of uncertainty introduced to the actual reward to account for the estimation error, and hence this term leads to a near optimistic estimation of the value function. The term "target policy", as defined in Sec. 3.2, represents the policy generated by the algorithm, which is employed to assess the algorithm's performance through a comparison with a series of optimal policies.
>
> Q3: Elaborate why $\hat{b}^k_h$ cannot serve as a point-wise uncertainty measure.
>
> A3: The main reason is because low-rank MDPs have unknown representations $\phi(s,a)$ and performance guarantee on the estimation error of such unknown representation $\phi(s,a)$ via MLE is given on the expected value over $(s,a)$ as in Lemma A.10. This further results in the following uncertainty bound on the expected value over $(s,a)$ (as in Lemma A.13), not on each individual $(s,a)$:
> $$
>   E_{(s_h,a_h) \sim (P^k, \pi)} [f_h^k(s_h,a_h)] \leq E_{(s_{h-1},a_{h-1}) \sim (P^k,\pi)}[\hat{b}^k_h(s_{h-1},a_{h-1})]+\Delta
> $$
>
> where $f_h^k$ is total variation distance between estimated models and true models, i.e. estimation errors, and $\Delta$ is a term related to the variation budget.
>
> Q4: In Step 1, why the exploration policy takes two uniformly chosen actions?
>
> A4: The main reason that exploration takes uniformly chosen actions is due to the fact that the representation is unknown in low-rank MDPs. So the agent needs to uniformly explore all actions and directions over the state-action space to get a comprehensive knowledge of the environment. Further, we note a single uniform step is not sufficient. Covariance matrix $U$ stores historical data information for future use of exploration. As explained above, the collected data is required to take at least one-step uniformly chosen actions. Subsequently, to effectively control estimation errors within target distribution, we use $U$ in the step back lemma (lemma A.12) to deal with distribution shifts, where importance sampling and one more uniform step is required. This leads the design of exploration policy with two-step uniformly chosen actions.
>
> Q5: Mention which lines correspond to Step 1, 2, and 3 in Alg. 1.
>
> A5: Step 1 corresponds to lines 5-8, Step 2 corresponds to line 9 (which calls Algorithm 2 E$^2$U), and Step 3 corresponds to lines 13-17.
>
> Q6: Define index of $s_h^{(k,h)}$ in Alg 1.
>
> A6: Subscript $h$ in ${s}_{h}^{(k,h)}$ indicates the data is collected at time step $h$ of each trajectory. Superscript $(k,h)$ indicates in which loop the data is collected (as in lines 3 and 4 of Algorithm 1, there are two loops regarding $k$ and $h$).
>
> Q7: What is the role of $\tilde{\alpha}_{k,W}$?
>
> A7: $\tilde\alpha_{k,W}$ quantifies data distribution shift from training data distribution to any target distribution. Consider Lemma A.13, where we bound estimation error of estimated model under target distribution (which is different from distribution of the training data used to estimate the model). $\tilde{\alpha}_{k,W}$ arises from applying the step back lemma to transmute target distribution back to distribution of training data.
>
> Specially, $\tilde\alpha_{k,W}$ is determined by the upper bound of distribution shift level given by $O((A+d^2)\log(kH|\Phi||\Psi|))$.
>
> Q8: Why can you remove assumption of [1]? How is $\tau$ selected? Elaborate on necessity to reset target policy.
>
> A8: For the reason of removing assumption, refer to A3 to Reviewer FzWn. Regarding how to select $\tau$, refer to A5 to Reviewer FzWn. We next elaborate the necessity to reset target policy every $\tau$ episode. Intuitively, if environment (i.e., MDP transition and/or reward) changes significantly during some episodes, then samples far in the past are not so informative to represent current transitions/reward and hence are not very useful for updating the current policy. Hence, periodic restart helps to stick with the most relevant samples and stablize the algorithm's performance against the nonstationary drift of the environment.
>
> Q9: The paper does not have an experimental section.
>
> A9: We will try our best to work out some experimental results.
>
> Q10: Assump. 4.3 is not very realistic. Is this assumption also made in related works, e.g., Wei and Luo (2021)?
>
> A10: Assumption 4.3 is introduced to deal with the nonstationarity. In order to improve the model estimation efficiency, here we need to apply historical data collected across different transition kernels for estimating the current model. To control such a model estimation error, both historical and current transition kernels should maintain non-zero transition probabilities (Assumption 4.3) to avoid singularity issues.
>
> Notably, Wei and Luo (2021) does not introduce an algorithm for nonstationary low-rank MDPs. Instead, for any types of MDPs, they require an off-the-shelf algorithm that performs effectively within nonstationary scenario with low variation as an input, and then uses a black-box approach to output an algorithm that performs efficiently under arbitrarily nonstationary environment without knowing variation budgets. In other words,  Wei and Luo (2021) would need our Algorithm 1 as an input in order to specialize to the nonstationary low-rank setting. This will make them also rely on our Assumption 4.3.
>
> [1] Jin, C., et. al (2020) Provably efficient reinforcement learning with linear function approximation. COLT.
>
> [2] Wei, C. Y., & Luo, H. (2021). Non-stationary reinforcement learning without prior knowledge: An optimal black-box approach. COLT.

---

> > ### Comment · Reviewer_niTs · 2023-08-11
> >
> > I thank the authors for carefully responding to all my questions/concerns. I am hence increasing my score appropriately.
> >
> > I would like to ask the authors to use the extra page to add the clarifications made in the rebuttal to the main body of the paper. This is particularly important for the clarity of the paper.

---

> > > ### Author Response · Authors · 2023-08-13
> > > **Thank you for your feedback!**
> > >
> > > We thank the reviewer very much for the further suggestions and for raising the score of your evaluation. We will add all the clarifications and discussions made in the rebuttal to the main body of the paper in the revised paper. Thanks again for your time and efforts during the review process!

---

### Author Rebuttal · Authors · 2023-08-09

Dear all reviewers,

We thank all the reviewers for providing the helpful review comments!

We further thank Reviewers niTs, FzWn and 5jBS for your positive evaluation of our paper. We have responded to your specific questions and will revise our paper based on your comments.

We also thank Reviewer ga5z for going through our proof, and want to clarify that the proof step you pointed out as the main concern is correct. Please see our response for detailed explanation.

We thank all the reviewers again for your efforts!

---

### Decision · Program_Chairs · 2023-09-21

**Decision:**

Accept (poster)

**Comment:**

This paper had strong support from a majority of its reviewers. The reviewers appreciated the soundness of the assumptions and the results, improvements in bounds and applicability (non-stationarity) over prior work and adequate placement of the surrounding literature.

Reviewer ga5z raised a concern regarding the proof, but did not respond to the authors' rebuttal. Additionally, we were unable to trace the origins of his concerns; so, we are setting aside the said review when evaluating the present work.

Therefore, we are happy to recommend the paper for acceptance.